# General Quantification of Covariate and Concept Shifts

**Hongbo Chen** [1]    **Li Charlie Xia** [1]

## Abstract

Generalization under distribution shift remains a core challenge in modern machine learning, yet existing learning bound theory is limited to narrow, idealized settings and is non-estimable from samples. In this paper, we bridge the gap between theory and practical applications. We first show that existing definition of concept shift breaks when the source and target supports mismatch. Leveraging entropic optimal transport, we propose a key notion: $\gamma^*$-concept shifts, and derive a general error bound unifying covariate and $\gamma^*$-concept shifts, which applies to broad loss functions, label spaces, and stochastic labeling. We further develop estimators for these shifts with concentration guarantees, and the DataShifts algorithm, which can quantify distribution shifts and estimate the error bound in most applications - a rigorous and general tool for analyzing learning error under distribution shift.

## 1. Introduction

With the growth of data and computing power, supervised learning has achieved remarkable success. Nonetheless, traditional supervised learning assumes that the training data (source domain) and the test or deployment data (target domain) share the same distribution. However, in many real-world applications, the test data distribution can differ substantially from the training distribution, and this discrepancy can significantly impact model performance in the target domain. To analyze such challenges, researchers theorized that the distributions between the source and target domains are shifted and developed methods to assess how learners trained in the source domain could perform on the target domain. Depending on whether the target domain data is accessible, the problems are further categorized into domain adaptation and domain generalization.

Theoretical results on distribution shift usually bound a model's target domain error by its source domain error plus a measure of the distribution shift. The shift is further dissected into X (covariate) and Y|X (concept) shifts (Moreno-Torres et al., 2012; Liu et al., 2021; Zhang et al., 2023). Early studies proposed using $\mathcal{H}$-divergence to measure X shift and derived an error bound for binary classification (Ben-David et al., 2006; 2010). Later works improved X shift using the maximum mean discrepancy (Long et al., 2015) and the Wasserstein distance (Shen et al., 2018; Courty et al., 2017). Further studies proposed more complex metrics for the X shift to obtain bounds for multiclass classification (Zhang et al., 2020; 2019). These results focused only on X shift and additionally relied on a joint-error term between the source and target domains. Lately, Zhao et al. (2019) improved this loose joint-error term and proposed a bound that explicitly considers both X and Y|X shifts for binary classification, and Zhang et al. (2023) extended the theory to multiclass classification.

In this paper, we focus on two remaining key problems in the existing theoretical frameworks, which significantly hinder researchers from analyzing X and Y|X shifts in applications:

- **Generalizability.** Existing theories rely on restrictive assumptions: they require deterministic labeling, omitting label noise and latent confounders commonly seen in practice; moreover, they only apply to classification with absolute error, excluding broader tasks such as regression and loss families.

- **Estimability.** Although existing theories provide a preliminary definition of Y|X shift, this definition and the accompanying error bounds—are *not* estimable. As a result, one cannot rigorously quantify Y|X shift on real data, nor assess its impact on model performance.

We aim to provide a general theoretical framework unifying X and Y|X shifts that widely applies to stochastic labeling, most supervised tasks, and broader loss functions. Moreover, these two shifts can be accurately estimated from samples, thereby offering a rigorous, plug-and-play tool for quantifying and analyzing distribution shifts in real applications.

Specifically, our key observation is that if X shift occurs, the

[1]Department of Statistics and Financial Mathematics, School of Mathematics, South China University of Technology, Guangzhou, China. First author: Hongbo Chen <hongboc616@gmail.com>. Correspondence to: Li Charlie Xia <lcxia@scut.edu.cn>.

*Proceedings of the $43^{rd}$ International Conference on Machine Learning*, Seoul, South Korea. PMLR 306, 2026. Copyright 2026 by the author(s).

supports of the source and target covariate distributions may not overlap. Such a support mismatch renders the existing theories' Y|X shift ill-defined, fundamentally causing the error bounds non-estimable and loose. Our key innovation is to employ the entropic optimal transport to give a general definition for X and Y|X shift. The $\gamma^*$-Y|X shift we propose, which depends on the entropic optimal transport coupling of X shift, stays well-defined even when supports mismatch, and applies to the stochastic labeling and general label space. Based on that, we derived a new error bound that considers both the X and the $\gamma^*$-Y|X shifts. Our new bound relies only on the Lipschitz continuity of the hypothesis $h$ and loss $\ell$, and is agnostic to specific hypothesis space, label space, or loss function, which naturally generalizes to binary and multiclass classification, regression and other tasks.

Since our $\gamma^*$-Y|X shift is well-defined regardless of support mismatch, X and Y|X shifts, and our new error bound becomes estimable from samples. Nonetheless, for X shift, the traditional plug-in estimator for entropic optimal transport tends to overestimate due to the curse of dimensionality. We thus further developed a debiased estimator that remains accurate in high dimensions. For our $\gamma^*$-Y|X shift, we also proposed an estimator. We proved both estimators' concentration inequalities to their true values. Leveraging these two estimators, we presented the DataShifts algorithm, enabling quantification of X and Y|X shifts on general labeled data. Finally, we apply our theoretical framework and DataShifts to three distinct tasks: Novozymes enzyme prediction, ColoredMNIST, and PACS, clearly validating the general effectiveness of our theoretical results.

**Contributions.** In summary, our major contributions are:

- We show that support mismatch makes the existing Y|X shift ill-defined—hence loose and non-estimable. We introduce the $\gamma^*$-Y|X shift as a well-defined concept shift, which possesses many desirable properties.

- We derive a general error bound unifying covariate and $\gamma^*$-concept shifts. Our bound covers broader learning scenarios, far beyond traditional binary classification.

- For the X and $\gamma^*$-Y|X shifts in our theory, we propose two estimators and prove their concentration inequalities, ensuring that these shifts can be rigorously estimated from finite samples.

- We integrate our theoretical results into the DataShifts algorithm, which can estimate X and $\gamma^*$-Y|X shifts from real data, supplying a rigorous and general tool for quantifying and analyzing distribution shifts.

The paper is organized as follows: Section 2 covers the preliminaries, Section 3 presents the population-level theoretical results, Section 4 focuses on the statistical results, and Section 5 presents experiments validating our theory.

## 2. Preliminary

### 2.1. Problem Setup

Our problem is to bound the error of models under distribution shift. Let $\mathcal{X}$ and $\mathcal{Y}$ be the covariate space and the label space, respectively. Let $\mathcal{D}_{XY}^S$ and $\mathcal{D}_{XY}^T$ be the joint distributions of covariates and labels on $\mathcal{X} \times \mathcal{Y}$ for the source and target domains, respectively. $\mathcal{D}_X^S, \mathcal{D}_X^T$ are their covariate marginals on $\mathcal{X}$. For any $x \in \mathcal{X}$, we let $\mathcal{D}_{Y|X=x}^S$ and $\mathcal{D}_{Y|X=x}^T$ be the conditional label distributions at $x$ in the source and target domain. Let $\mathcal{Y}'$ be the output space of the learner, $\ell : \mathcal{Y} \times \mathcal{Y}' \to \mathbb{R}$ the loss, and $\mathcal{H} \subseteq \{g : \mathcal{X} \to \mathcal{Y}'\}$ the hypothesis space. For a hypothesis $h \in \mathcal{H}$, the learning errors for source and target domains are:

$$\epsilon_S(h) = \mathbb{E}_{(x_S, y_S) \sim \mathcal{D}_{XY}^S} \left[ \ell(y_S, h(x_S)) \right]$$

$$\epsilon_T(h) = \mathbb{E}_{(x_T, y_T) \sim \mathcal{D}_{XY}^T} \left[ \ell(y_T, h(x_T)) \right]$$

We hope to bound $\epsilon_T(h)$ by $\epsilon_S(h)$ and a measure of distribution shift, consistent with existing theoretical results (Ben-David et al., 2006; Zhao et al., 2019). Notably, the space $\mathcal{X}$ can be the raw input space or a representation space output by an upstream learner (Ben-David et al., 2006). Our theory treats them in the same way, so it applies to both raw data and learned representations.

**Stochastic and deterministic labeling.** Above, we assume that the label $y$ follows the conditional distribution at point $x$: $y \sim \mathcal{D}_{Y|X=x}$, namely the stochastic labeling setting (Zhao et al., 2019). It enables our theory to accommodate latent confounders and label noise, which are common in practice. In contrast, existing theories oversimplify by using deterministic labeling: a labeling function $f : \mathcal{X} \to \mathcal{Y}$ with $y = f(x)$, a special case of stochastic labeling in which the conditional distribution collapses to a Dirac mass at $f(x)$, i.e., $\mathcal{D}_{Y|X=x} = \delta_{f(x)}$.

### 2.2. Existing Theory and Ill-Defined Y|X Shift

We first demonstrate that support mismatch leads to an ill-defined Y|X shift, a key flaw in existing theory.

**Definition 2.1** (Support). Let $\mu$ be a probability measure on the topological space $(\mathcal{X}, \tau)$. Its support is defined as:

$$\text{supp}(\mu) = \{ x \in \mathcal{X} \mid \forall U \in \tau, \ x \in U, \ \mu(U) > 0 \}$$

$\text{supp}(\mu)$ is the closure of every region where $\mu$ has positive measure, or equivalently, the complement of the union of all $\mu$-null open sets:

$$\text{supp}(\mu) = \mathcal{X} \setminus \left\{ \bigcup \{ U \in \tau \mid \mu(U) = 0 \} \right\}$$

**Lemma 2.2** (Ill-Defined Expectation of Conditional Probability). *For probability measure $P_X$, a $P_X$-measurable*

*function $P(A \mid X = x) : \mathcal{X} \to [0, 1]$ is the conditional probability of event $A$ given $X = x$. For probability measure $P'_X$ satisfying $\mathrm{supp}(P'_X) \setminus \mathrm{supp}(P_X) \neq \varnothing$, the expectation $\mathbb{E}_{P'_X}[P(A \mid X = x)]$ is arbitrary.*

**Remark** This problem arises because the conditional probability $P(A \mid X = x)$ is unique almost everywhere with respect to $P_X$ (unique $P_X$-a.e.), rather than to $P'_X$. When $\mathrm{supp}(P'_X) \setminus \mathrm{supp}(P_X) \neq \varnothing$, $P'_X$ assigns positive mass to some $P_X$-null sets. In this case, $\mathbb{E}_{P'_X}[P(A \mid X = x)]$ depends on the values of $P(A \mid X = x)$ on sets where it is not uniquely determined. Hence, support mismatch leads to the ill-defined expectation of conditional probability. This issue directly impacts the definition and computation of Y|X shift in existing theories, since the Y|X shift is typically formulated as an expectation over conditional labeling distributions or its collapsed labeling functions.

With deterministic labeling, Zhao et al. (2019) used $\mathcal{H}$-divergence (Ben-David et al., 2006) to derive an error bound for soft-label binary classification under X and Y|X shifts:

**Theorem 2.3** (Existing Learning Bound on Distribution Shift). *Let $\mathcal{D}_X^S, \mathcal{D}_X^T$ be the covariate distributions and $f_S, f_T : \mathcal{X} \to [0, 1]$ be the labeling functions for the source and target domain. Using absolute error loss $|\cdot|$, for any hypothesis space $\mathcal{H} \subseteq [0, 1]^{\mathcal{X}}$ define:*

$$\tilde{\mathcal{H}} = \{\mathrm{sgn}(|h(x) - h'(x)| - t) \mid h, h' \in \mathcal{H}, \ t \in [0, 1]\},$$

*then for any $h \in \mathcal{H}$,*

$$\epsilon_T(h) \leq \epsilon_S(h) + d_{\tilde{\mathcal{H}}}(\mathcal{D}_X^S, \mathcal{D}_X^T) + \min\Big\{\mathbb{E}_{x \sim \mathcal{D}_X^S}\big[|f_S(x)$$
$$- f_T(x)|\big], \mathbb{E}_{x \sim \mathcal{D}_X^T}\big[|f_S(x) - f_T(x)|\big]\Big\} \quad (1)$$

Here, $d_{\tilde{\mathcal{H}}}$ measures X shift, and the two expectations $\mathbb{E}_{x \sim \mathcal{D}_X^S}[|f_S(x) - f_T(x)|], \mathbb{E}_{x \sim \mathcal{D}_X^T}[|f_S(x) - f_T(x)|]$ are both Y|X shift but measured with the source or target covariate distribution, with the smaller one taken in the bound.

**Remark (Limitations of Theorem 2.3)** First, it is highly specialized, applying only to deterministic labeling, binary classification, and absolute loss. Moreover, when X shift causes support mismatch : $\mathrm{supp}(\mathcal{D}_X^T) \setminus \mathrm{supp}(\mathcal{D}_X^S) \neq \varnothing$, the conditional distribution $\mathcal{D}_{Y|X=x}^S$ (i.e., $f_S(x)$) is only unique $\mathcal{D}_X^S$-a.e. and arbitrary on the mismatched region, so the Y|X shift $\mathbb{E}_{x \sim \mathcal{D}_X^T}[|f_S(x) - f_T(x)|]$ is ill-defined. Similarly, if $\mathrm{supp}(\mathcal{D}_X^S) \setminus \mathrm{supp}(\mathcal{D}_X^T) \neq \varnothing$, the other Y|X shift term is also ill-defined. This additionally causes two problems:

- **Loose bound.** The ill-defined $Y|X$ shifts can take arbitrary values, making the bound in Eq. (1) loose.

- **Non-estimable.** When the real concept $\mathcal{D}_{Y|X=x}^S$ or $f_S(x)$ is unknown, we cannot sample it outside

$\mathrm{supp}(\mathcal{D}_X^S)$; hence, under support mismatch the expectations $\mathbb{E}_{x \sim \mathcal{D}_X^T}[|f_S(x) - f_T(x)|]$ is non-estimable, as well as $\mathbb{E}_{x \sim \mathcal{D}_X^S}[|f_S(x) - f_T(x)|]$.

### 2.3. Entropic Optimal Transport

We introduce entropic optimal transport, which we will employ to give the general definitions of X and Y|X shifts properly. It measures the distance between two probability distributions, augmenting conventional optimal transport with a relative-entropy regularizer. Given two probability distributions $\mathbb{P}$ and $\mathbb{Q}$ on the metric space $(\Omega, \rho)$ and a parameter $\beta \geq 0$, the order-1 entropic optimal transport is:

$$W_\beta(\mathbb{P}, \mathbb{Q}) = \inf_{\gamma \in \Gamma(\mathbb{P}, \mathbb{Q})} \left\{ \int \rho \, d\gamma + \beta \, H(\gamma \mid \mathbb{P} \otimes \mathbb{Q}) \right\} \quad (2)$$

where $H(\gamma \mid \mathbb{P} \otimes \mathbb{Q}) = \int \log\left(\frac{d\gamma(x_1, x_2)}{d\mathbb{P}(x_1) \, d\mathbb{Q}(x_2)}\right) d\gamma(x_1, x_2)$. Here, $\rho(x_1, x_2)$ is the cost of transporting mass between points, $\Gamma(\mathbb{P}, \mathbb{Q})$ is the set of joint distributions with marginals $\mathbb{P}$ and $\mathbb{Q}$, and $\gamma(x_1, x_2)$ is a transport coupling. Hence, $W_\beta(\mathbb{P}, \mathbb{Q})$ is the minimum total transport cost between $\mathbb{P}$ and $\mathbb{Q}$ under an entropy regularizer. When $\beta = 0$, this reduces to the Wasserstein-1 distance, denoted by $W_1(\mathbb{P}, \mathbb{Q})$. Compared with other distribution distances, (entropic) optimal transport has a well-established geometric meaning (Gangbo & McCann, 1996), and we will consistently use it to measure distribution shifts.

## 3. Theoretical Results

In this section, we focus on population-level theoretical results. We assume that covariate $X$ and label $Y$ are distributed on the metric spaces $(\mathcal{X}, \rho_{\mathcal{X}})$ and $(\mathcal{Y}, \rho_{\mathcal{Y}})$, respectively, and give the rigorous and general definitions of their distribution shifts and essential theorems below.

### 3.1. General X and Y|X Shifts

**Definition 3.1** (X Shift). Using entropic optimal transport, the X (covariate) shift is defined as:

$$S_{Cov} = W_\beta(\mathcal{D}_X^S, \mathcal{D}_X^T) = \inf_{\gamma \in \Gamma(\mathcal{D}_X^S, \mathcal{D}_X^T)} \left\{ \int \rho_{\mathcal{X}}(x_S, x_T) \right.$$
$$\left. d\gamma(x_S, x_T) + \beta \, H(\gamma \mid \mathcal{D}_X^S \otimes \mathcal{D}_X^T) \right\} \quad (3)$$

where the optimal coupling is denoted as $\gamma^*$, a joint distribution of $\mathcal{D}_X^S, \mathcal{D}_X^T$ that gives the minimum transport cost.

With realistic stochastic labeling, the label follows a conditional distribution given the covariate; we give a general and rigorous definition of Y|X shift below:

**Definition 3.2** ($\gamma^*$-Y|X Shift). Let $\gamma^*$ be the optimal transport coupling as in Definition 3.1, $S_{pair}(x_S, x_T) =$

$W_1\big(\mathcal{D}^S_{Y|X=x_S}, \mathcal{D}^T_{Y|X=x_T}\big)$, then the $\gamma^*$-Y|X shift is defined as the expectation of $S_{pair}(x_S, x_T)$ under $\gamma^*$:

$$S^{\gamma^*}_{Cpt} = \mathbb{E}_{(x_S,x_T)\sim\gamma^*}\big[S_{pair}(x_S, x_T)\big]$$

$$= \int W_1\big(\mathcal{D}^S_{Y|X=x_S}, \mathcal{D}^T_{Y|X=x_T}\big)\, d\gamma^*(x_S, x_T) \quad (4)$$

This definition is based on the optimal transport coupling $\gamma^*$ for X shift. $S_{pair}(x_S, x_T)$ denotes the paired conditional distribution shift between the source domain at $x_S$ and the target domain at $x_T$. Intuitively, since optimal transport coupling $\gamma^*$ places most of its mass on nearby pairs $(x_S, x_T)$, $S^{\gamma^*}_{Cpt}$ is evaluated mainly on such neighboring points.

With deterministic labeling, this definition reduces to:

**Corollary 3.3** (Consistency under Deterministic Labeling). *Assume deterministic labeling* $\mathcal{D}^S_{Y|X=x} = \delta_{f_S(x)}$, $\mathcal{D}^T_{Y|X=x} = \delta_{f_T(x)}$, *then:*

$$S^{\gamma^*}_{Cpt} = \mathbb{E}_{\gamma^*}\big[S_{pair}(x_S, x_T)\big] = \mathbb{E}_{\gamma^*}\big[\rho_{\mathcal{Y}}(f_S(x_S), f_T(x_T))\big]$$

The following three lemmas guarantee the good properties of $\gamma^*$-Y|X Shift.

**Lemma 3.4** (Support of $\gamma^*$). *For any* $\gamma \in \Gamma(\mathcal{D}^S_X, \mathcal{D}^T_X)$, $\mathrm{supp}(\gamma) \subseteq \mathrm{supp}(\mathcal{D}^S_X) \times \mathrm{supp}(\mathcal{D}^T_X)$.

**Lemma 3.5** (Uniqueness of $\gamma^*$). *If* $\beta > 0$, *then the entropic optimal transport coupling* $\gamma^*$ *in Definition 3.1 is unique.*

**Lemma 3.6** (Collapse of $\gamma^*$). *If* $\beta = 0$ *and* $\mathcal{D}^S_X = \mathcal{D}^T_X$, $\gamma^*$ *collapses to diagonal (identity) coupling* $\gamma^* = (\mathrm{Id}, \mathrm{Id})_{\#}\mathcal{D}^S_X$, *equivalently, for every measurable* $A \subseteq \mathcal{X} \times \mathcal{X}$, $\gamma^*(A) = \int \mathbf{1}_{\{(x,x)\in A\}}\, d\mathcal{D}^S_X(x)$, *where* $\mathbf{1}_{\{\cdot\}}$ *is the indicator function.*

Combining Lemmas 3.4 and 3.5, the rigor of $\gamma^*$-Y|X Shift is ensured:

**Theorem 3.7** (Well-Definedness of $\gamma^*$-Y|X Shift). *For* $\beta > 0$, *the* $\gamma^*$-Y|X *shift* $S^{\gamma^*}_{Cpt}$ *in Definition 3.2 is unique even when* $\mathrm{supp}(\mathcal{D}^S_X) \neq \mathrm{supp}(\mathcal{D}^T_X)$.

**Remark** Such $\gamma^*$-Y|X shift not only avoids the ill-definition encountered by existing theory, but also makes Y|X shift tight and estimable. On the other hand, combining Corollary 3.3 and Lemma 3.6, the relationship between $\gamma^*$-Y|X shift and existing Y|X shift is shown as follows:

**Proposition 3.8** (Relationship to existing Y|X shift). *Assume deterministic labeling* $\mathcal{D}^S_{Y|X=x} = \delta_{f_S(x)}$, $\mathcal{D}^T_{Y|X=x} = \delta_{f_T(x)}$ *and* $\mathcal{Y} \subset \mathbb{R}$ *with* $\rho_{\mathcal{Y}} = |\cdot|$, *if* $\beta = 0$ *and* $\mathcal{D}^S_X = \mathcal{D}^T_X$, *then:*

$$S^{\gamma^*}_{Cpt} = \mathbb{E}_{x\sim\mathcal{D}^S_X}\big[|f_S(x) - f_T(x)|\big]$$

$$= \mathbb{E}_{x\sim\mathcal{D}^T_X}\big[|f_S(x) - f_T(x)|\big]$$

**Remark** The above $\gamma^*$-Y|X shift, defined via the entropic optimal transport coupling $\gamma^*$, applies to general settings including stochastic labeling. When $\mathcal{D}^S_X = \mathcal{D}^T_X$, it recovers the existing Y|X shift; and when the supports of $\mathcal{D}^S_X$ and $\mathcal{D}^T_X$ are mismatched, it still remains rigorous.

### 3.2. General Learning Bound

We now give a new cross-domain learning error bound based on X and $\gamma^*$-Y|X shift. To be general, the output space of the learner is a metric space $(\mathcal{Y}', \rho'_{\mathcal{Y}})$, possibly different from the true label space $(\mathcal{Y}, \rho_{\mathcal{Y}})$. For the loss function $\ell : \mathcal{Y} \times \mathcal{Y}' \to \mathbb{R}$, we require the following basic assumption from them:

**Assumption 3.9** (Separately Lipschitz Continuity). For metric spaces $(\mathcal{Y}, \rho_{\mathcal{Y}})$ and $(\mathcal{Y}', \rho'_{\mathcal{Y}})$, a function $\ell : \mathcal{Y} \times \mathcal{Y}' \to \mathbb{R}$ satisfies separately $(L_\ell, L'_\ell)$-Lipschitz if there exist $L_\ell, L'_\ell \geq 0$ such that for any $y_1, y_2 \in \mathcal{Y}$ and $y'_1, y'_2 \in \mathcal{Y}'$, there is:

$$\big|\ell(y_1, y'_1) - \ell(y_2, y'_2)\big| \leq L_\ell\, \rho_{\mathcal{Y}}(y_1, y_2) + L'_\ell\, \rho'_{\mathcal{Y}}(y'_1, y'_2) \quad (5)$$

**Remark** This is a mild assumption: most loss functions are differentiable, which already implies continuity. And their stable optimization usually needs bounded gradients in a region, further ensuring Lipschitz continuity (Boyd & Vandenberghe, 2004). Note that this assumption also covers asymmetric losses. For instance, for binary classification with label space $[0, 1]$, output space is typically $[a, 1-a]$ ($a \in (0, 0.5)$) by Sigmoid function and $\rho_{\mathcal{Y}} = \rho'_{\mathcal{Y}} = |\cdot|$, the cross-entropy loss: $\ell_{CE}(y, \hat{y}) = -y\log\hat{y} - (1-y)\log(1-\hat{y})$ satisfies separately $(L_\ell, L'_\ell)$-Lipschitz with $L_\ell = \log\big(\frac{1-a}{a}\big)$ and $L'_\ell = \frac{1}{a}$.

**Corollary 3.10** (Composition Preserves Separate Lipschitzness). *Let* $h : \mathcal{X} \to \mathcal{Y}'$ *be* $L_h$-Lipschitz *and let* $\ell : \mathcal{Y} \times \mathcal{Y}' \to \mathbb{R}$ *be separately* $(L_\ell, L'_\ell)$-Lipschitz. *Then composite function* $\ell\big(y, h(x)\big) : \mathcal{Y} \times \mathcal{X} \to \mathbb{R}$ *is separately* $(L_\ell, L_h L'_\ell)$-Lipschitz.

The following two lemmas characterize the transport coupling of joint distributions $\mathcal{D}^S_{XY}$ and $\mathcal{D}^T_{XY}$:

**Lemma 3.11** (Separately Weak Duality). *Let* $\mathcal{G}(y, x) : \mathcal{Y} \times \mathcal{X} \to \mathbb{R}$ *is separately* $(L_{\mathcal{Y}}, L_{\mathcal{X}})$-Lipschitz, *for any coupling of joint distributions* $\gamma_{XY} \in \Gamma(\mathcal{D}^S_{XY}, \mathcal{D}^T_{XY})$, *there exists:*

$$\big|\mathbb{E}_{\mathcal{D}^S_{XY}}[\mathcal{G}]\big] - \mathbb{E}_{\mathcal{D}^T_{XY}}[\mathcal{G}]\big| \leq L_{\mathcal{X}} E_{\gamma_{XY}}[\rho_{\mathcal{X}}] + L_{\mathcal{Y}} E_{\gamma_{XY}}[\rho_{\mathcal{Y}}]$$
$$(6)$$

**Lemma 3.12** (Gluing Construction for Joint Coupling). *Let* $\gamma^*$ *be the optimal transport coupling of X shift in Definition 3.1,* $\gamma^*_{Y|(x_S,x_T)}$ *be the optimal transport coupling of* $S_{pair}(x_S, x_T)$ *in Definition 3.2, construct the joint distribution of couplings:* $\gamma_{XY}(dx_S, dx_T, dy_S, dy_T) = \gamma^*(dx_S, dx_T)\gamma^*_{Y|(x_S,x_T)}(dy_S, dy_T)$, *then* $\gamma_{XY}$ *is the coupling of joint distributions:* $\gamma_{XY} \in \Gamma(\mathcal{D}^S_{XY}, \mathcal{D}^T_{XY})$.

Combining Corollary 3.10, Lemmas 3.11 and 3.12, we obtain the general cross-domain error bound as follows:

**Theorem 3.13** (General Learning Bound). *Given the covariate space* $(\mathcal{X}, \rho_{\mathcal{X}})$, *the label space* $(\mathcal{Y}, \rho_{\mathcal{Y}})$, *and the output space* $(\mathcal{Y}', \rho'_{\mathcal{Y}})$, *the source and target distributions are* $(\mathcal{D}_X^S, \mathcal{D}_{Y|X=x}^S)$ *and* $(\mathcal{D}_X^T, \mathcal{D}_{Y|X=x}^T)$, *respectively. If the loss* $\ell : \mathcal{Y} \times \mathcal{Y}' \to \mathbb{R}$ *satisfies separately* $(L_\ell, L'_\ell)$-*Lipschitz, then for any hypothesis* $h : \mathcal{X} \to \mathcal{Y}'$ *that satisfies* $L_h$-*Lipschitz, the following bound holds:*

$$\epsilon_T(h) \ \leq \ \epsilon_S(h) \ + \ L_h\, L'_\ell\, S_{Cov} \ + \ L_\ell\, S_{Cpt}^{\gamma^*} \quad (7)$$

*where* $S_{Cov}$ *is the X shift in Definition 3.1 and* $S_{Cpt}^{\gamma^*}$ *is the* $\gamma^*$-*Y|X shift in Definition 3.2.*

**Remark**  This elegant bound unifies the covariate shift $S_{Cov}$ and the concept shift ($\gamma^*$-Y|X shift) $S_{Cpt}^{\gamma^*}$'s effect on target error via the Lipschitz factors $L_h\, L'_\ell$ and $L_\ell$. Notably, it depends only on the Lipschitz continuity of the hypothesis $h$ and the loss $\ell$, without any other specific restriction on the label space $\mathcal{Y}$ or loss $\ell$. It naturally covers binary classification or regression tasks when $\mathcal{Y}$ is one-dimensional, and multiclass classification or multi-label tasks when $\mathcal{Y}$ is multi-dimensional. Besides, since the bound holds under stochastic labeling, it applies to a wide range of supervised learning scenarios in practice. On the other hand, by using the well-defined $\gamma^*$-Y|X shift, our bound can be tighter than the existing bound, and more crucially, both the covariate and concept shifts in our theory can be robustly estimated from the finite samples. Additionally, since the entropic optimal transport $S_{Cov}$ increases with the hyperparameter $\beta$, we recommend choosing $\beta$ as a small non-zero value to balance the tightness and the rigor of the theory.

## 4. Statistical Results

In practice, true domain distributions are all unknown. We wish to estimate the shifts from domain samples and analyze how these shifts will influence model performance. In this section, we focus on sample-level theoretical results. Suppose we have i.i.d. samples $\{(X_i^{(S)}, Y_i^{(S)})\}$ and $\{(X_j^{(T)}, Y_j^{(T)})\}$ drawn from $\mathcal{D}_{XY}^S$ and $\mathcal{D}_{XY}^T$ with sizes $N_S$ and $N_T$, respectively. This section involves three parts: the estimation of X shift, the estimation of Y|X shift, and the DataShifts algorithm to estimate the overall bound.

### 4.1. Estimation of X Shift

By Definition 3.1, the X shift is the entropic optimal transport between $\mathcal{D}_X^S$ and $\mathcal{D}_X^T$. The traditional estimation method uses the entropic optimal transport of empirical distributions—known as the plug-in estimator.

**Traditional Plug-in Estimator**  Given i.i.d. samples $\{X_i^{(S)}\} \sim \mathcal{D}_X^S$, $\{X_j^{(T)}\} \sim \mathcal{D}_X^T$ with sample sizes $N_S, N_T$ respectively, set the empirical measures: $\widehat{\mathcal{D}_X^S} = \frac{1}{N_S} \sum_{i=1}^{N_S} \delta_{X_i^{(S)}}, \widehat{\mathcal{D}_X^T} = \frac{1}{N_T} \sum_{j=1}^{N_T} \delta_{X_j^{(T)}}$, their entropic optimal transport is:

$$W_\beta\big(\widehat{\mathcal{D}_X^S}, \widehat{\mathcal{D}_X^T}\big) = \min_{\hat{\gamma}} \langle C, \hat{\gamma} \rangle + \beta \sum_{i=1}^{N_S} \sum_{j=1}^{N_T} \hat{\gamma}_{ij} \log \hat{\gamma}_{ij}$$

$$+ \beta \log\big(N_S N_T\big) \quad \text{s.t.} \quad \hat{\gamma}\mathbf{1} = \tfrac{1}{N_S}\mathbf{1}, \ \hat{\gamma}^\top \mathbf{1} = \tfrac{1}{N_T}\mathbf{1} \quad (8)$$

where $C \in \mathbb{R}_+^{N_S \times N_T}$ is the cost matrix with $c_{ij} = \rho_{\mathcal{X}}\big(X_i^{(S)}, X_j^{(T)}\big)$, and $\hat{\gamma} \in \mathbb{R}_+^{N_S \times N_T}$ represents any discretized transport coupling satisfying the given linear constraints. Such an optimization problem can be solved efficiently at a large scale by the *Sinkhorn* algorithm (Cuturi, 2013; Genevay et al., 2016).

**Curse of Dimensionality.**  However, in modern applications, the covariate space $\mathcal{X}$ is often high-dimensional. Even when two distributions are very close, their samples' distance can be large. Such curse of dimensionality makes the plug-in estimator greatly overestimated (Verleysen & François, 2005; Panaretos & Zemel, 2019) (Fig.1(a)). When $\beta$ is small, entropic optimal transport behaves similarly to Wasserstein distance (Carlier et al., 2017; 2023), and the upward bias decays only in $O(N^{-1/d})$ order (Fournier & Guillin, 2015), which implies that increasing $N$ has a very limited debiasing effect when $d$ is high (Fig.1(b)).

To address the overestimation problem of the traditional plug-in estimator, we propose the following debiased estimator.

**Definition 4.1** (Debiased Estimator). Given i.i.d. samples $\{X_i^{(S)}\} \sim \mathcal{D}_X^S$, $\{X_j^{(T)}\} \sim \mathcal{D}_X^T$ with sample sizes $N_S, N_T$ respectively, split the samples in half to obtain four independent empirical measures: $\widehat{\mathcal{D}_X^S}{}' = \frac{2}{N_S} \sum_{i=1}^{N_S/2} \delta_{X_i^{(S)}}$, $\widehat{\mathcal{D}_X^S}{}'' = \frac{2}{N_S} \sum_{i=N_S/2+1}^{N_S} \delta_{X_i^{(S)}}, \widehat{\mathcal{D}_X^T}{}' = \frac{2}{N_T} \sum_{j=1}^{N_T/2} \delta_{X_j^{(T)}}$, $\widehat{\mathcal{D}_X^T}{}'' = \frac{2}{N_T} \sum_{j=N_T/2+1}^{N_T} \delta_{X_j^{(T)}}$. The debiased estimator is:

$$W_\beta^{deb}\big(\widehat{\mathcal{D}_X^S}, \widehat{\mathcal{D}_X^T}\big) = \Big| \tfrac{1}{2} W_\beta\big(\widehat{\mathcal{D}_X^S}{}', \widehat{\mathcal{D}_X^T}{}'\big)^2 + \tfrac{1}{2} W_\beta\big(\widehat{\mathcal{D}_X^S}{}'', \widehat{\mathcal{D}_X^T}{}''\big)^2$$

$$- \tfrac{1}{2} W_\beta\big(\widehat{\mathcal{D}_X^S}{}', \widehat{\mathcal{D}_X^S}{}''\big)^2 - \tfrac{1}{2} W_\beta\big(\widehat{\mathcal{D}_X^T}{}', \widehat{\mathcal{D}_X^T}{}''\big)^2 \Big|^{1/2} \quad (9)$$

**Remark**  This debiased estimator uses four plug-in estimators. The first two terms, $W_\beta\big(\widehat{\mathcal{D}_X^S}{}', \widehat{\mathcal{D}_X^T}{}'\big)$ and $W_\beta\big(\widehat{\mathcal{D}_X^S}{}'', \widehat{\mathcal{D}_X^T}{}''\big)$, estimate the distance between $\mathcal{D}_X^S$ and

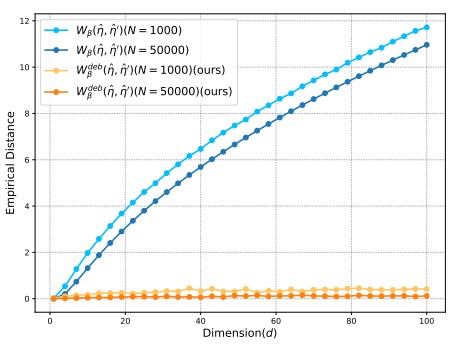

(a) Empirical distance vs. dimension ($d$)

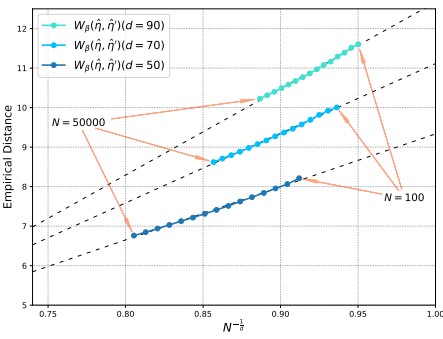

(b) Empirical distance vs. sample size ($N$)

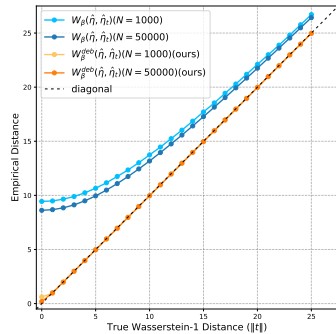

(c) Empirical distance vs. true $W_1$

*Figure 1.* (a)–(c) show the empirical distance from entropic optimal transport ($\beta = 0.001$) versus dimension $d$, sample size $N$, and true Wasserstein-1 distance. In (a)–(b), $\hat{\eta}$ and $\hat{\eta}'$ are independent empirical measures from high-dimensional standard normals, thus the true distance is zero. The traditional estimator $W_\beta(\hat{\eta}, \hat{\eta}')$ greatly overestimates as $d$ increases, as shown in (a), and even a much larger $N$ only brings a small improvement, as shown in (b). In (c), with $d = 70$, $\hat{\eta}_t$ is the empirical measure of shifted standard normal $\mathcal{N}(t, I)$, whose true Wasserstein-1 distance to the standard normal is $\|t\|$. Our debiased estimator remains accurate in every case.

$\mathcal{D}_X^T$, including the sample bias. And the last two terms, $W_\beta\left(\widehat{\mathcal{D}_X^S}', \widehat{\mathcal{D}_X^S}''\right)$ and $W_\beta\left(\widehat{\mathcal{D}_X^T}', \widehat{\mathcal{D}_X^T}''\right)$ estimate the distance arising from sample bias only. Subtracting the two parts reduces the sample bias, thus giving a better estimate of $W_\beta(\mathcal{D}_X^S, \mathcal{D}_X^T)$. Compared with the traditional plug-in estimator, our debiased estimator $W_\beta^{deb}$ reduces the overestimation and remains accurate regardless of the true distribution distance (see Fig.1(a) and 1(c)).

Moreover, when the covariate space is *Euclidean*, we derived a concentration inequality guaranteeing that the debiased estimator converges to the true distance:

**Theorem 4.2** (Concentration Inequality of Debiased Estimator). *Let $\mathcal{D}_X^S, \mathcal{D}_X^T$ be two distributions on $(\mathbb{R}^d, \|\cdot\|)$ with finite squared-exponential moments. For i.i.d. samples $\{X_i^{(S)}\} \sim \mathcal{D}_X^S$, $\{X_j^{(T)}\} \sim \mathcal{D}_X^T$ with sample sizes $N_S, N_T$ respectively, when $\beta = 0$, and for any $\varepsilon > 0$, there exists $N$ such that if $N_S, N_T > N$, then:*

$$\mathbb{P}\left(\left| W_\beta^{deb}\left(\widehat{\mathcal{D}_X^S}, \widehat{\mathcal{D}_X^T}\right) - W_\beta(\mathcal{D}_X^S, \mathcal{D}_X^T)\right| > \varepsilon\right) \leq$$

$$2\exp\left(-\frac{\lambda_S N_S V_\varepsilon \varepsilon^2}{32}\right) + 2\exp\left(-\frac{\lambda_T N_T V_\varepsilon \varepsilon^2}{32}\right) \quad (10)$$

*where $\lambda_S, \lambda_T > 0$ depend only on squared-exponential moments of $\mathcal{D}_X^S, \mathcal{D}_X^T$, respectively, and $V_\varepsilon \in [2 - \sqrt{3}, 2)$ depends only on $W_\beta(\mathcal{D}_X^S, \mathcal{D}_X^T)/\varepsilon$.*

**Remark** This theorem implies that the deviation probability of the debiased estimator decays exponentially with sample sizes, so with high probability, our debiased estimator can well approximate the true entropic optimal transport distance $W_\beta(\mathcal{D}_X^S, \mathcal{D}_X^T)$. Notably, it also shows the enlightening fact that our estimator's concentration depends not only on each distribution's scale characterized by $\lambda_S, \lambda_T$ but also on the true distance between two distributions.

### 4.2. Estimation of Y|X Shift

As above, we defined the $\gamma^*$-Y|X shift $S_{Cpt}^{\gamma^*}$ in Definition 3.2; we now estimate it from samples.

**Definition 4.3** (Estimator for $\gamma^*$-Y|X Shift). Given i.i.d. samples $\{(X_i^{(S)}, Y_i^{(S)})\} \sim \mathcal{D}_{XY}^S$, $\{(X_j^{(T)}, Y_j^{(T)})\} \sim \mathcal{D}_{XY}^T$ with sample sizes $N_S, N_T$ respectively, the estimator of $\hat{\gamma}^*$-Y|X shift is defined as:

$$\hat{S}_{Cpt} = \sum_{i=1}^{N_S} \sum_{j=1}^{N_T} \rho_{\mathcal{Y}}\left(Y_i^{(S)}, Y_j^{(T)}\right) \hat{\gamma}_{ij}^*, \quad (11)$$

where $\hat{\gamma}^* \in \mathbb{R}_+^{N_S \times N_T}$ represents the discrete optimal transport coupling for $W_\beta(\widehat{\mathcal{D}_X^S}, \widehat{\mathcal{D}_X^T})$.

**Remark** In Section 4.1, we use the debiased estimator $W_\beta^{deb}(\widehat{\mathcal{D}_X^S}, \widehat{\mathcal{D}_X^T})$ for X shift, whereas Definition 4.3 still estimates $\gamma^*$-Y|X shift via the optimal transport coupling from the plug-in estimator $W_\beta(\widehat{\mathcal{D}_X^S}, \widehat{\mathcal{D}_X^T})$. This is because the curse of dimensionality mainly affects the transport cost (i.e., inter-sample distances), rather than the transport coupling. Leveraging the stability of entropic optimal transport (Eckstein & Nutz, 2022), the following lemma establishes the convergence of the coupling for plug-in estimator:

**Lemma 4.4** (Stability of Entropic Optimal Transport Coupling). *Let $\mathcal{D}_X^S, \mathcal{D}_X^T$ be two distributions on $(\mathcal{X}, \rho_{\mathcal{X}})$ with finite squared-exponential moments. $\gamma^*$ and $\hat{\gamma}^*$ are the optimal transport couplings of $W_\beta(\mathcal{D}_X^S, \mathcal{D}_X^T)$ and $W_\beta(\widehat{\mathcal{D}_X^S}, \widehat{\mathcal{D}_X^T})$ respectively, then:*

$$W_1(\gamma^*, \hat{\gamma}^*) \leq \Lambda + 2\sqrt{\frac{2}{\beta \lambda_{\gamma^*}}}\sqrt{\Lambda}$$

$$\Lambda = W_1\left(\mathcal{D}_X^S, \widehat{\mathcal{D}_X^S}\right) + W_1\left(\mathcal{D}_X^T, \widehat{\mathcal{D}_X^T}\right) \quad (12)$$

where $W_1(\gamma^*, \hat{\gamma}^*)$ is computed on $\mathcal{X} \times \mathcal{X}$ with metric $\rho\big((x_1, x_2), (x_1', x_2')\big) := \rho_{\mathcal{X}}(x_1, x_1') + \rho_{\mathcal{X}}(x_2, x_2')$, $\lambda_{\gamma^*} > 0$ depend only on squared-exponential moments of $\mathcal{D}_X^S, \mathcal{D}_X^T$.

**Remark** This lemma shows that the Wasserstein-1 distance between the plug-in estimator coupling and the population optimal transport coupling, is bounded by the sum of the marginal Wasserstein-1 distance between each population and its empirical distribution, with the bound scaled by the parameter $\beta$. Such a uniform bound is guaranteed only when $\beta > 0$, which further highlights the necessity of using entropic optimal transport.

By Lemma 4.4, when covariate space $\mathcal{X}$ is *Euclidean* and label space $\mathcal{Y}$ is a bounded set in an Euclidean space, we derive a concentration inequality guaranteeing estimator in Definition 4.3 approximates true value:

**Theorem 4.5** (Concentration Inequality for Definition 4.3). *Let $\mathcal{D}_X^S, \mathcal{D}_X^T$ be two distributions on $(\mathbb{R}^d, \|\cdot\|)$ with finite squared-exponential moments. Let the label space $\mathcal{Y} \subset \mathbb{R}^{d'}$ be bounded by $M = \sup_{y,y' \in \mathcal{Y}} \|y - y'\|$, on which conditional distributions $\mathcal{D}_{Y|X=x_S}^S, \mathcal{D}_{Y|X=x_T}^T$ satisfy $L_{Y|X}$-Lipschitz respectively: $d_{\mathrm{TV}}\big(\mathcal{D}_{Y|X=x}, \mathcal{D}_{Y|X=x'}\big) \le L_{Y|X} \|x - x'\|$. For i.i.d. samples $\{(X_i^{(S)}, Y_i^{(S)})\} \sim \mathcal{D}_{XY}^S$, $\{(X_j^{(T)}, Y_j^{(T)})\} \sim \mathcal{D}_{XY}^T$ with sample sizes $N_S, N_T$, when $\beta > 0$, and for any $\varepsilon > 0$, there exists $N$ such that if $N_S, N_T > N$, then:*

$$\mathbb{P}\big(|\hat{S}_{Cpt} - S_{Cpt}^{\gamma^*} - \Delta| > \varepsilon\big) \le 2\exp\Big(-\frac{N_S N_T \Phi \varepsilon^2}{(N_S + N_T)M^2}\Big)$$

$$+ \exp\Big(-\frac{\lambda_S^{1/2} N_S \Phi \varepsilon^2}{4\lambda_T^{1/2} M^2}\Big) + \exp\Big(-\frac{\lambda_T^{1/2} N_T \Phi \varepsilon^2}{4\lambda_S^{1/2} M^2}\Big) \quad (13)$$

*where $d_{\mathrm{TV}}$ is total-variation distance, $\lambda_S, \lambda_T > 0$ depend only on the squared-exponential moments of $\mathcal{D}_X^S, \mathcal{D}_X^T$, $\Phi > 0$ depends on $L_{Y|X}, \lambda_S, \lambda_T, \beta$, the bias $\Delta$ is a constant.*

**Remark** This theorem implies that the deviation probability of the $\gamma^*$-Y|X Shift estimator $\hat{S}_{Cpt}$ decays exponentially with sample sizes, so with high probability, the estimator can well approximate the $S_{Cpt}^{\gamma^*} + \Delta$. The bias term $\Delta$ arises because Definition 4.3 uses $\rho_{\mathcal{Y}}\big(Y_i^{(S)}, Y_j^{(T)}\big)$ as a single-point estimate of $S_{pair}\big(X_i^{(S)}, X_j^{(T)}\big)$ in Definition 3.2. The following two propositions show that the bias $\Delta$ is controlled.

**Proposition 4.6** ($\Delta$ in Deterministic Labeling). *Assume deterministic labeling $\mathcal{D}_{Y|X=x}^S = \delta_{f_S(x)}, \mathcal{D}_{Y|X=x}^T = \delta_{f_T(x)}$, then the $\Delta$ in Theorem 4.5 satisfies:*

$$\Delta = 0 \quad (14)$$

**Remark** This proposition shows that under deterministic labeling, the estimator $\hat{S}_{Cpt}$ accurately estimates the Y|X

shift $S_{Cpt}^{\gamma^*}$ itself. For stochastic labeling, we show that $\Delta$ can be bounded by the irreducible error (James et al., 2013) (also known as the Bayes risk (Berger, 2013)) in traditional statistical learning.

**Proposition 4.7** ($\Delta$ in Stochastic Labeling). *When $(\mathcal{Y}, \rho_{\mathcal{Y}}) = (\mathbb{R}^{d'}, \|\cdot\|)$, the irreducible error of joint distribution $\mathcal{D}_{XY}$ under squared loss $\|\cdot\|^2$ is defined as:* $\mathrm{I}(\mathcal{D}_{XY}) = \inf_{g:\mathcal{X} \to \mathcal{Y}} \mathbb{E}_{(x,y) \sim \mathcal{D}_{XY}}\big[\|y - g(x)\|^2\big]$, *then the $\Delta$ in Theorem 4.5 satisfies:*

$$0 \le \Delta \le \sqrt{\mathrm{I}(\mathcal{D}_{XY}^S)} + \sqrt{\mathrm{I}(\mathcal{D}_{XY}^T)} \quad (15)$$

**Remark** The irreducible error is the fundamental error inherent to stochastic labeling that no model can overcome. When the problem is learnable, covariate and label are often well correlated, and the irreducible error is small relative to the overall label variability. In this case, Proposition 4.7 guarantees that the estimator $\hat{S}_{Cpt}$ does not substantially overestimate the Y|X shift $S_{Cpt}^{\gamma^*}$.

### 4.3. DataShifts Algorithm

**On the Lipschitz Constant of Learners.** The Lipschitz constant of the learner have been well studied, such as logistic regression (Roux et al., 2012), multi-layer perceptron (MLP) (Fazlyab et al., 2019), convolutional neural networks (CNN) (Virmaux & Scaman, 2018; Zou et al., 2019) and attention mechanism (Kim et al., 2021; Castin et al., 2023). Although the Lipschitz constants of modern large-scale neural networks such as ResNet-50 or Transformers remain difficult to analyze, researchers are more concerned with whether these models learn distribution-robust representations (Hendrycks et al., 2020), rather than distribution shift in the raw input space. In this setting, the covariate space $\mathcal{X}$ is the representation space, and the downstream model (the hypothesis $h$ in our theory) is often a simple learner—such as a linear classifier—whose Lipschitz constant is still easy to handle. We summarize the Lipschitz constants of various learners in Appendix D.

---

**Algorithm 1** DataShifts

---

**Input:** hyperparameter $\beta$ (default 0.01),
samples $\{(X_i^{(S)}, Y_i^{(S)})\}, \{(X_j^{(T)}, Y_j^{(T)})\}$,
Lipschitz constants $L_\ell, L_\ell', L_h$ (optional),
source domain empirical error $\hat{\epsilon}_S$ (optional)

**Do:**
Estimate X shift by 4.1 as $\hat{S}_{Cov}$
Estimate $\gamma^*$-Y|X shift by 4.3 as $\hat{S}_{Cpt}$
**if** $L_\ell, L_\ell', L_h$, and $\hat{\epsilon}_S$ are provided **then**
    Estimate bound: $B = \hat{\epsilon}_S + L_h L_\ell' \hat{S}_{Cov} + L_\ell \hat{S}_{Cpt}$
**end if**
**Return:** $\hat{S}_{Cov}, \hat{S}_{Cpt}$ and $B$ (optional)

---

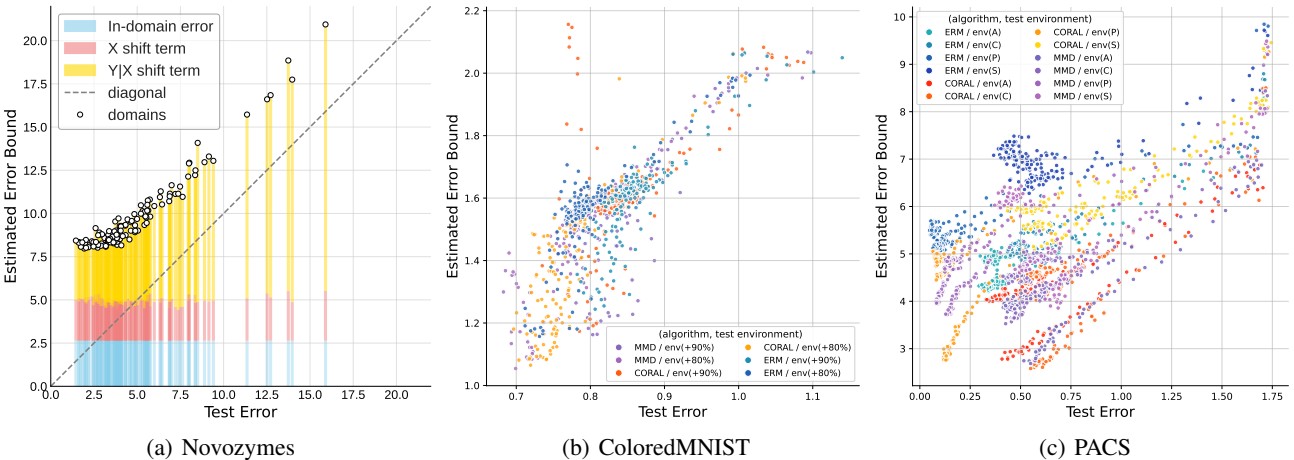

*Figure 2.* (a)–(c) show that the estimated error bounds track the test error well across three distinct tasks, corroborating the general effectiveness of our learning bound and estimators.

By leveraging the theoretical results above, we give the plug-and-play Algorithm 1 (DataShifts) for quantifying X and Y|X shifts from finite samples and estimating error bounds.

## 5. Experiments

In this section, we apply our estimable general theory to three distinct practical tasks: tabular regression, image binary classification, and image multi-class classification – to validate the general effectiveness of our bound and estimators. We also conduct experiments on synthetic tasks where existing theory apply, demonstrating our bound is tighter.

### 5.1. Novozymes Enzyme Stability Prediction

The Novozymes Enzyme Prediction Competition (Pultz et al., 2022) is a large-scale Kaggle contest. It is a tabular regression task, with 9,000 point-mutation samples spanning 180 enzyme families, aiming to predict transition temperatures for unseen families. Each enzyme family is treated as a separate domain; due to distribution shifts between them, thousands of participants found it difficult to develop any effective solution.

We select the 60 enzyme families with the smallest pretraining error as the source domain, and treat each of the remaining 120 enzyme families as a target domain. In this task, we focus on the distribution shift between the raw data of different enzyme families, taking the 20-dimensional input feature space as covariate space $\mathcal{X}$. We train a 3-layer MLP on the source domain; an analysis of its Lipschitz constant is provided in the Appendix D.4. We use the absolute loss (separately $(1, 1)$-Lipschitz) to evaluate the source domain error and the test error on each target domain, and apply our DataShifts algorithm($\beta = 0.2$) to estimate an error bound for each target domain.

We plot the test error and the estimated error bound on each target domain in Fig. 2(a). In this figure, the overall trend of the test error and the error bound lies just above the diagonal, indicating that our bound is tight and effectively captures the test error under distribution shift. Meanwhile, it directly shows the contributions of X and Y|X shifts on the error bound. The large Y|X shift across enzyme families is what drives the generalization failure in this contest.

### 5.2. ColoredMNIST and PACS

ColoredMNIST and PACS are two standard tasks in the DomainBed benchmark(Gulrajani & Lopez-Paz, 2020). ColoredMNIST is a binary classification task with 70,000 handwritten digit images exhibiting color shift, while PACS is a multi-class object recognition task with 9,991 images exhibiting style shift. For both tasks, we treat the model's representation space as the covariate space $\mathcal{X}$. Following the DomainBed setup, we use the simple CNN with a 128-dimensional representation for ColoredMNIST, and ResNet-50 with a 2048-dimensional representation for PACS. In this setting, the hypothesis $h$ in our theory corresponds to the model's final layer (a linear classifier), whose Lipschitz constant is analyzed in the Appendix D.3.

We evaluate three methods: ERM, CORAL, and MMD. Adhering to DomainBed, we select the best hyperparameters via training-domain validation over 20 random hyperparameter trials for each method, domain, and trial. For ColoredMNIST, we train each best-hyperparameter run for 5,000 steps and save checkpoints every 100 steps. For PACS, since the model converges earlier, we train for 1,000 steps and save checkpoints every 20 steps. At each checkpoint, we regard the mixture distribution over training domains as the source domain and the test domain as the target domain. We still use the absolute loss to measure source and target errors,

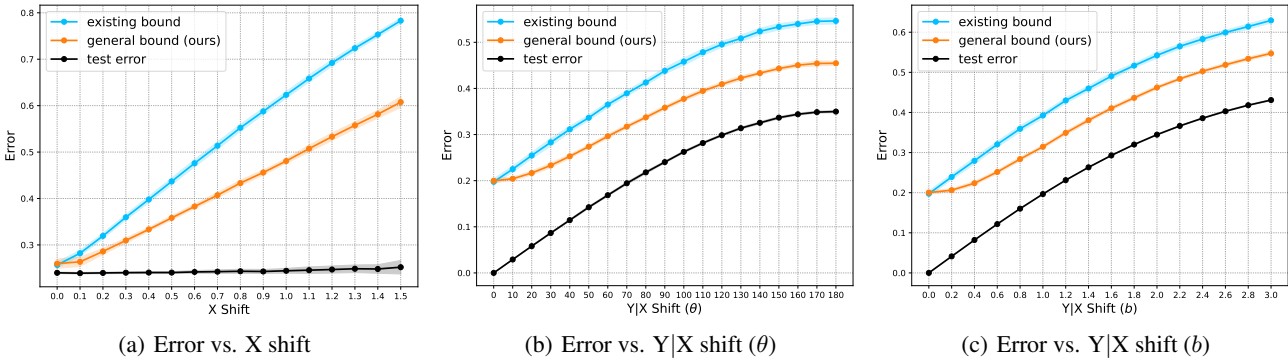

*Figure 3.* (a)–(c) respectively show the relationship between the estimated learning bounds and the X or Y|X shift in synthetic binary classification tasks. As these shifts increase, our bound becomes significantly tighter than the existing bound.

and run our DataShifts algorithm ($\beta = 0.2$) using each domain's representation-label pairs to estimate the test domain error bound at every checkpoint.

We plot the test error and the estimated error bound for both tasks in Fig. 2(b) and 2(c). In Fig. 2(b), the bound tracks the test error well across checkpoints. The MMD (purple) and CORAL (yellow) points lie closer to the lower-left region, indicating that these X shift-reducing methods indeed yield smaller bounds, and consequently lower error on ColoredM-NIST. In Fig. 2(c), although the bound becomes looser in magnitude when applied to PACS, a more complex image classification task, it still exhibits a consistent trend with the test error in each run.

### 5.3. Synthetic Binary Classification

We compare our theory with the existing bound in Zhao et al. (2019), which is shown to be tighter than previous learning bounds. As discussed in Remark 2.2, the existing learning bound only applies to soft-label binary classification with deterministic labeling and absolute loss. Since its estimation requires sampling the true concepts $f_S$ and $f_T$ outside the supports of the covariate distributions $\mathcal{D}_X^S$, $\mathcal{D}_X^T$ (oracle), it can only be estimated on synthetic tasks where the true concepts are known.

We construct a synthetic binary classification task using logistic regression. Let the covariate space be 10-dimensional, and the source inputs are sampled from uniform distribution: $\mathcal{D}_X^S = \mathrm{U}([0,1]^{10})$, with labels generated by $f_S(x) = \sigma(w_S^\top x + b_S)$, where $w_S = \mathbf{1}/\sqrt{10}$, $b_S = 0$ and $\sigma$ is the sigmoid function. The target inputs are generated by shifting $\mathcal{D}_X^S$ along a random direction, thereby controlling the X shift. And the target labels are generated by another logistic regression: $f_T(x) = \sigma(w_T^\top x + b_T)$, where $w_T$ is obtained by rotating $w_S$ by angle $\theta$ toward a random direction. $\theta$ and $b_T$ further control the Y|X shift. By varying the X shift, $\theta$, and $b_T$, we obtain a series of target domains. We

also use logistic regression as the learner and train it on the source domain. Its Lipschitz constant is analyzed in Appendix D.2. For each target domain, we estimate the test error, the existing and our bounds under the absolute loss.

We plot the learning bounds with respect to the X shift and the Y|X shifts ($\theta$ and $b_T$) in Fig. 3. In Fig. 3(a), since both the source concept and the learner are logistic regression models, the learner can fit the source concept well, and the X shift in the target domain has only a minor effect on the test error. As the X shift increases, our bound becomes tighter than the existing bound. In Figs. 3(b) and 3(c), as the Y|X shift increases, the existing bound becomes looser than ours. This verifies our point in Remark 2.2: the ill-defined Y|X shift in the existing bound is loose, whereas our $\gamma^*$-Y|X shift and the accompanying learning bound can be tighter.

## 6. Conclusion

In this paper, we focus on a general and estimable theoretical framework for learning under distribution shift. We first introduce a key notion, $\gamma^*$-Y|X shift, via entropic optimal transport, which addresses the ill-definedness in existing theory. Then we derive a general learning bound unifying X shift and $\gamma^*$-Y|X shift. We further develop concentration-guaranteed estimators for both shifts, and integrate our theory into the plug-and-play DataShifts algorithm, enabling researchers to quantify and analyze distribution shift in broad settings. Experiments on practical and synthetic tasks validate the general effectiveness and tightness of our theory. We believe our theoretical framework takes an important step toward learning under distribution shift and will spur further algorithmic advances.

## Acknowledgements

We also thank Dr. Jie Ren for suggestions and feedback on this work. This study was funded by the National Natural

Science Foundation of China (12571529) and Guangdong Basic and Applied Basic Research Foundation (2024A1515-010699) to LCX.

## Impact Statement

This paper presents work whose goal is to advance the field of Machine Learning. There are many potential societal consequences of our work, none which we feel must be specifically highlighted here.

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

## A. Related Work

Generalization theory under distribution shift is an important and long-standing problem. Researchers usually seek to measure distribution shift using some distribution divergence and derive corresponding learning bounds. Early theoretical works mainly focused on the X (covariate) shift. Ben-David et al. (2006; 2010) used the $\mathcal{H}$-divergence to measure X shift between the source and target domains for binary classification, thereby deriving learning bounds for domain adaptation. Subsequent studies characterized the X shift via maximum mean discrepancy (MMD), leading to new learning bounds and improved domain adaptation algorithms (Long et al., 2015). Redko et al. (2017); Courty et al. (2017); Shen et al. (2018) adopted optimal transport (OT) to characterize X shift and obtained similar bounds. Further studies proposed more complex metrics for the X shift to extend the theory to multiclass classification (Zhang et al., 2019; 2020; El Hamri et al., 2025).

Most of these theories focus only on the X shift and follow a similar proof strategy, resulting in a joint error term between the source and target domains, $\lambda = \min_h \epsilon_S(h) + \epsilon_T(h)$, which is widely recognized as loose and non-estimable. Our baseline, Zhao et al. (2019), showed that the Y|X (concept) shift is implicitly contained in this joint error term $\lambda$, and derived a tighter learning bound by explicitly introducing the Y|X shift instead. Zhang et al. (2023) extended this theory to multiclass classification. We follow this path of explicitly characterizing the Y|X shift, and point out that the existing definition of the Y|X shift becomes ill-defined when the supports of the covariate distributions are mismatched, which still makes the Y|X shift loose and non-estimable. Through entropic optimal transport, we introduce a well-defined $\gamma^*$-Y|X shift and the corresponding learning bound, which can be rigorously estimated from samples. Our theory can be tighter than existing bounds and further generalizes to a broader tasks, losses, and stochastic labeling.

*Table 1.* Comparison of learning bounds for distribution shift.

| Bound | Divergence | Terms | | | Properties | | |
|---|---|---|---|---|---|---|---|
| | | X shift | Y\|X shift | Remaining Term | Tight | Estimable | General |
| Ben-David et al. (2006) | $\mathcal{H}$-divergence | ✓ | | joint-error $\lambda$ | | | |
| Ben-David et al. (2010) | $\mathcal{H}$-divergence | ✓ | | joint-error $\lambda$ | | | |
| Long et al. (2015) | MMD | ✓ | | joint-error $\lambda$ | | | |
| Redko et al. (2017) | OT | ✓ | | joint-error $\lambda$ | | | |
| Courty et al. (2017) | OT | estimated XY shift | | joint-error $\lambda$, $kM\Phi$ | | | ✓ |
| Shen et al. (2018) | OT | ✓ | | joint-error $\lambda$ | | | |
| Zhao et al. (2019) | $\mathcal{H}$-divergence | ✓ | ✓(ill-defined) | | ✓ | | |
| Zhang et al. (2023) | $\mathcal{H}$-divergence | ✓ | ✓(ill-defined) | | ✓ | | ✓ |
| El Hamri et al. (2025) | hierarchical OT | ✓ | | joint-error $\lambda$ | | | |
| **Ours** | entropic OT | ✓ | ✓ | | ✓✓ | ✓ | ✓✓ |

✓✓ indicates a stronger degree than ✓.

Distribution shift theory often serves as a theoretical framework for domain adaptation and domain generalization, where many algorithms have been developed (Sugiyama et al., 2007; Glorot et al., 2011; Sun & Saenko, 2016; Ganin et al., 2016; Pei et al., 2018; Arjovsky et al., 2019; Krueger et al., 2021; Rame et al., 2022). However, the DomainBed benchmark (Gulrajani & Lopez-Paz, 2020) shows that many algorithms do not outperform standard empirical risk minimization. This calls for tighter and more general theoretical frameworks to explain the generalization failures of these algorithms. In addition, the proposed $\gamma^*$-Y|X shift takes a nested OT form, whose other mathematical properties have also been studied recently, such as the gradient flow of the Wasserstein-over-Wasserstein distance (Bonet et al., 2025).

## B. Technical Tools

In this section, we introduce several existing mathematical tools to prove the propositions in Appendix C.

### B.1. Conditional Probability

**Definition B.1** (Conditional Probability)**.** Let $(\Omega, \mathcal{F}, P)$ be a probability space and $X : \Omega \to \mathcal{X}$ a random variable with distribution $P_X$. For event $A \in \mathcal{F}$, a $P_X$-measurable function $P(A \mid X = x) : \mathcal{X} \to [0, 1]$ is the conditional probability of

$A$ given $X$ if:

$$\forall B \in \mathcal{B}_{\mathcal{X}}, \; P(A \cap \{X \in B\}) = \int_B P(A \mid X = x) \, dP_X(x)$$

where $\mathcal{B}_{\mathcal{X}}$ denotes the Borel $\sigma$-algebra on $\mathcal{X}$.

The conditional probability is unique almost everywhere with respect to $P_X$ (unique $P_X$-a.e.) (Kallenberg & Kallenberg, 1997). That is, for any $P_X$-null set $Z$ with $P_X(Z) = 0$, $\int_Z P(A \mid X = x) \, dP_X(x) = 0$, implying that $P(A \mid X = x)$ can be assigned arbitrarily on $Z$ without affecting its overall properties. Intuitively, it means discussing conditional probabilities on a null set is meaningless (Hájek, 2003).

## B.2. McDiarmid's Inequality

**Lemma B.2** (McDiarmid's Inequality). *Let $Z_1, \ldots, Z_n$ be independent random variables taking values in sets $\mathcal{Z}_1, \ldots, \mathcal{Z}_n$, and let $F : \mathcal{Z}_1 \times \cdots \times \mathcal{Z}_n \to \mathbb{R}$. Assume $F$ satisfies the bounded-differences property: there exist constants $c_1, \ldots, c_n \geq 0$ such that for every $i \in [n]$ and any two inputs $(z_1, \ldots, z_n)$ and $(z_1, \ldots, z_i', \ldots, z_n)$ differing only at the $i$-th coordinate,*

$$\left| F(z_1, \ldots, z_i, \ldots, z_n) - F(z_1, \ldots, z_i', \ldots, z_n) \right| \leq c_i.$$

*Then for any $\varepsilon > 0$,*

$$\mathbb{P}\Big( F(Z_1, \ldots, Z_n) - \mathbb{E}\big[F(Z_1, \ldots, Z_n)\big] \geq \varepsilon \Big) \leq \exp\left( -\frac{2\varepsilon^2}{\sum_{i=1}^n c_i^2} \right),$$

*and*

$$\mathbb{P}\Big( \big| F(Z_1, \ldots, Z_n) - \mathbb{E}[F(Z_1, \ldots, Z_n)] \big| \geq \varepsilon \Big) \leq 2 \exp\left( -\frac{2\varepsilon^2}{\sum_{i=1}^n c_i^2} \right).$$

## B.3. Concentration for Empirical $W_1$

**Lemma B.3** (Square-Exponential Moment Implies $T_1$ (Bolley et al., 2007)). *Let $(\mathcal{X}, \rho)$ be a Polish metric space and $\mu$ a Borel probability measure on $\mathcal{X}$. Assume $\mu$ admits a square-exponential moment: there exist $a > 0$ and $x_0 \in \mathcal{X}$ such that*

$$\int_{\mathcal{X}} \exp\big(a \, \rho(x, x_0)^2\big) \, d\mu(x) < \infty.$$

*Then $\mu$ satisfies a $T_1(\lambda)$ transport-entropy inequality for some $\lambda > 0$: for all probability measures $\nu$ on $\mathcal{X}$,*

$$W_1(\mu, \nu) \leq \sqrt{\frac{2}{\lambda} H(\nu \mid \mu)}.$$

**Lemma B.4** (Bolley–Guillin Concentration for Empirical $W_1$ (Bolley et al., 2007)). *Let $\mu$ be a probability measure on $(\mathbb{R}^d, \|\cdot\|)$ that satisfies the transport inequality $T_1(\lambda)$ for some $\lambda > 0$, namely for all probability measures $\nu$,*

$$W_1(\mu, \nu) \leq \sqrt{\frac{2}{\lambda} H(\nu \mid \mu)},$$

*where $H(\nu \mid \mu)$ denotes the relative entropy. Let $\hat{\mu} = \frac{1}{N} \sum_{i=1}^N \delta_{X_i}$ be the empirical measure of i.i.d. samples $X_1, \ldots, X_N \sim \mu$. Then for any $d' > d$ and any $\lambda_\mu \in (0, \lambda)$, there exists a constant $N_0$ depending only on $\lambda_\mu$, $d'$, and the squared-exponential moments of $\mu$ such that for any $\varepsilon > 0$ and any*

$$N \geq N_0 \max\big(\varepsilon^{-(d'+2)}, 1\big),$$

*we have*

$$\mathbb{P}\big(W_1(\mu, \hat{\mu}) > \varepsilon\big) \leq \exp\left( -\frac{\lambda_\mu}{2} N \varepsilon^2 \right).$$

# C. Full proofs

## C.1. Proof of Lemma 2.2

*Proof.* Since $\text{supp}(P_X') \setminus \text{supp}(P_X) \neq \varnothing$, pick $x_0 \in \text{supp}(P_X') \setminus \text{supp}(P_X)$. By Definition 2.1, $x_0 \notin \text{supp}(P_X)$ implies that there exists an open neighborhood $U \subseteq \mathcal{X}$ with $x_0 \in U$ such that

$$P_X(U) = 0.$$

On the other hand, $x_0 \in \text{supp}(P_X')$ implies $P_X'(V) > 0$ for every open neighborhood $V \ni x_0$, hence in particular $P_X'(U) > 0$. Let $Z := U$. Then $Z \in \mathcal{B}_\mathcal{X}$ and

$$P_X(Z) = 0, \qquad P_X'(Z) > 0.$$

Let $P(A \mid X = x) : \mathcal{X} \to [0,1]$ be a conditional probability in Definition B.1, i.e., for all $B \in \mathcal{B}_\mathcal{X}$,

$$P\big(A \cap \{X \in B\}\big) = \int_B P(A \mid X = x)\, dP_X(x).$$

For any constant $c \in [0,1]$, define a modified function

$$\widetilde{P}(A \mid X = x) := \begin{cases} P(A \mid X = x), & x \notin Z, \\ c, & x \in Z. \end{cases}$$

For any $B \in \mathcal{B}_\mathcal{X}$,

$$\begin{aligned}
\int_B \widetilde{P}(A \mid X = x)\, dP_X(x) &= \int_{B \setminus Z} P(A \mid X = x)\, dP_X(x) + \int_{B \cap Z} c\, dP_X(x) \\
&= \int_{B \setminus Z} P(A \mid X = x)\, dP_X(x) + c\, P_X(B \cap Z) \\
&= \int_{B \setminus Z} P(A \mid X = x)\, dP_X(x) \qquad (\text{since } P_X(Z) = 0) \\
&= \int_B P(A \mid X = x)\, dP_X(x) = P\big(A \cap \{X \in B\}\big),
\end{aligned}$$

so $\widetilde{P}(A \mid X = x)$ is also a valid conditional probability of $A$ given $X$ by Definition B.1, as another version of $P(A \mid X = x)$.

Now take expectation under $P_X'$:

$$\begin{aligned}
\mathbb{E}_{P_X'}\big[\widetilde{P}(A \mid X = x)\big] &= \int_\mathcal{X} \widetilde{P}(A \mid X = x)\, dP_X'(x) \\
&= \int_{\mathcal{X} \setminus Z} P(A \mid X = x)\, dP_X'(x) + \int_Z c\, dP_X'(x) \\
&= \int_{\mathcal{X} \setminus Z} P(A \mid X = x)\, dP_X'(x) + c\, P_X'(Z).
\end{aligned}$$

Since $P_X'(Z) > 0$ and $c \in [0,1]$ is arbitrary, the value of $\mathbb{E}_{P_X'}[P(A \mid X = x)]$ can be changed by choosing different $c$ while preserving the defining property of conditional probability under $P_X$. Hence $\mathbb{E}_{P_X'}[P(A \mid X = x)]$ is arbitrary. $\qquad\square$

**Remark** The above proof shows that support mismatch leads to arbitrariness. Broadly speaking, support mismatch commonly breaks the rigor of theoretical analyses. Some studies explicitly exclude it from analysis by extra assumptions, such as done in the absolute continuity in measure theory (Duncan, 1970), the positivity assumption in causal inference (Cole & Frangakis, 2009), or support overlap in importance sampling (Gelman & Meng, 1998).

## C.2. Proof of Corollary 3.3

*Proof.* By Definition 3.2,

$$S_{Cpt}^{\gamma^*} = \mathbb{E}_{(x_S, x_T) \sim \gamma^*} \left[ W_1 \left( \mathcal{D}_{Y|X=x_S}^S, \mathcal{D}_{Y|X=x_T}^T \right) \right].$$

Under deterministic labeling, $\mathcal{D}_{Y|X=x_S}^S = \delta_{f_S(x_S)}$ and $\mathcal{D}_{Y|X=x_T}^T = \delta_{f_T(x_T)}$, hence

$$S_{pair}(x_S, x_T) = W_1 \left( \delta_{f_S(x_S)}, \delta_{f_T(x_T)} \right).$$

For any $a, b \in \mathcal{Y}$, any coupling $\pi \in \Gamma(\delta_a, \delta_b)$ must satisfy $\pi(\{a\} \times \mathcal{Y}) = 1$ and $\pi(\mathcal{Y} \times \{b\}) = 1$, hence $\pi = \delta_{(a,b)}$ is the unique coupling. Therefore,

$$W_1(\delta_a, \delta_b) = \inf_{\pi \in \Gamma(\delta_a, \delta_b)} \int \rho_{\mathcal{Y}}(y_1, y_2) \, d\pi = \int \rho_{\mathcal{Y}}(y_1, y_2) \, d\delta_{(a,b)}(y_1, y_2) = \rho_{\mathcal{Y}}(a, b).$$

Applying this with $a = f_S(x_S)$ and $b = f_T(x_T)$ yields $S_{pair}(x_S, x_T) = \rho_{\mathcal{Y}}(f_S(x_S), f_T(x_T))$, and thus

$$S_{Cpt}^{\gamma^*} = \mathbb{E}_{(x_S, x_T) \sim \gamma^*} \left[ \rho_{\mathcal{Y}}(f_S(x_S), f_T(x_T)) \right],$$

as claimed. □

## C.3. Proof of Lemma 3.4

*Proof.* Any $\gamma \in \Gamma(\mathcal{D}_X^S, \mathcal{D}_X^T)$ is a coupling of $\mathcal{D}_X^S$ and $\mathcal{D}_X^T$, hence its first marginal is $\mathcal{D}_X^S$ and its second marginal is $\mathcal{D}_X^T$.

Take any $(x_S, x_T) \notin \mathrm{supp}(\mathcal{D}_X^S) \times \mathrm{supp}(\mathcal{D}_X^T)$. Then either $x_S \notin \mathrm{supp}(\mathcal{D}_X^S)$ or $x_T \notin \mathrm{supp}(\mathcal{D}_X^T)$.

If $x_S \notin \mathrm{supp}(\mathcal{D}_X^S)$, by Definition 2.1 there exists an open neighborhood $U \subseteq \mathcal{X}$ of $x_S$ such that $\mathcal{D}_X^S(U) = 0$. Since the first marginal of $\gamma$ is $\mathcal{D}_X^S$,

$$\gamma(U \times \mathcal{X}) = \mathcal{D}_X^S(U) = 0.$$

In particular, for any open neighborhood $V$ of $x_T$ we have $\gamma(U \times V) \le \gamma(U \times \mathcal{X}) = 0$, so $\gamma(U \times V) = 0$. Thus, taking the product open set $U \times V$ containing $(x_S, x_T)$, we conclude that $(x_S, x_T) \notin \mathrm{supp}(\gamma)$ by Definition 2.1 applied on the product topology of $\mathcal{X} \times \mathcal{X}$.

The case $x_T \notin \mathrm{supp}(\mathcal{D}_X^T)$ is symmetric: there exists an open $V \ni x_T$ with $\mathcal{D}_X^T(V) = 0$, and using the second marginal of $\gamma$ gives $\gamma(\mathcal{X} \times V) = 0$, hence $\gamma(U \times V) = 0$ for any open $U \ni x_S$, implying $(x_S, x_T) \notin \mathrm{supp}(\gamma)$.

Therefore every point outside $\mathrm{supp}(\mathcal{D}_X^S) \times \mathrm{supp}(\mathcal{D}_X^T)$ is outside $\mathrm{supp}(\gamma)$, i.e.,

$$\mathrm{supp}(\gamma) \subseteq \mathrm{supp}(\mathcal{D}_X^S) \times \mathrm{supp}(\mathcal{D}_X^T).$$

□

## C.4. Proof of Lemma 3.5

*Proof.* Let $\nu := \mathcal{D}_X^S \otimes \mathcal{D}_X^T$ and consider the entropic optimal transport objective

$$J(\gamma) := \int \rho_{\mathcal{X}}(x_S, x_T) \, d\gamma(x_S, x_T) + \beta \, H(\gamma \mid \nu), \qquad \gamma \in \Gamma(\mathcal{D}_X^S, \mathcal{D}_X^T).$$

We claim that $J$ is *strictly convex* over the convex set $\Gamma(\mathcal{D}_X^S, \mathcal{D}_X^T)$. Indeed, the transport cost term $\gamma \mapsto \int \rho_{\mathcal{X}} \, d\gamma$ is linear in $\gamma$. For the entropy term, write $r = \frac{d\gamma}{d\nu}$; then

$$H(\gamma \mid \nu) = \int r \log r \, d\nu.$$

For any $\gamma_1, \gamma_2 \in \Gamma(\mathcal{D}_X^S, \mathcal{D}_X^T)$ with $\gamma_1 \neq \gamma_2$, by Lemma 3.4,

$$\mathrm{supp}(\gamma_1) \subseteq \mathrm{supp}(\mathcal{D}_X^S) \times \mathrm{supp}(\mathcal{D}_X^T) = \mathrm{supp}(\nu).$$

$$\operatorname{supp}(\gamma_2) \subseteq \operatorname{supp}(\mathcal{D}_X^S) \times \operatorname{supp}(\mathcal{D}_X^T) = \operatorname{supp}(\nu).$$

Because $\varphi(t) = t \log t$ is strictly convex on $[0, \infty)$, for and any $\lambda \in (0, 1)$,

$$H\big(\lambda\gamma_1 + (1 - \lambda)\gamma_2 \mid \nu\big) < \lambda H(\gamma_1 \mid \nu) + (1 - \lambda)H(\gamma_2 \mid \nu),$$

hence (multiplying by $\beta > 0$) the whole objective satisfies

$$J\big(\lambda\gamma_1 + (1 - \lambda)\gamma_2\big) < \lambda J(\gamma_1) + (1 - \lambda)J(\gamma_2).$$

Now suppose, toward a contradiction, that there are two distinct optimal couplings $\gamma_1 \neq \gamma_2$ minimizing $J$ over $\Gamma(\mathcal{D}_X^S, \mathcal{D}_X^T)$. By convexity of $\Gamma(\mathcal{D}_X^S, \mathcal{D}_X^T)$, their mixture $\gamma_\lambda := \lambda\gamma_1 + (1 - \lambda)\gamma_2$ is feasible. Strict convexity then yields

$$J(\gamma_\lambda) < \lambda J(\gamma_1) + (1 - \lambda)J(\gamma_2) = \inf_{\gamma \in \Gamma} J(\gamma),$$

a contradiction. Therefore the optimizer $\gamma^*$ is unique. $\qquad\square$

## C.5. Proof of Lemma 3.6

*Proof.* When $\beta = 0$, Definition 3.1 reduces to the Wasserstein-1 problem

$$\inf_{\gamma \in \Gamma(\mathcal{D}_X^S, \mathcal{D}_X^T)} \int \rho_{\mathcal{X}}(x_S, x_T) \, d\gamma(x_S, x_T),$$

where $\rho_{\mathcal{X}}$ is a metric, hence $\rho_{\mathcal{X}} \geq 0$ and $\rho_{\mathcal{X}}(x_S, x_T) = 0$ if $x_S = x_T$.

Consider the diagonal (identity) coupling $\bar{\gamma} := (\mathrm{Id}, \mathrm{Id})_{\#}\mathcal{D}_X^S$, i.e.,

$$\bar{\gamma}(A) = \int \mathbf{1}_{\{(x,x) \in A\}} \, d\mathcal{D}_X^S(x).$$

It is immediate that $\bar{\gamma} \in \Gamma(\mathcal{D}_X^S, \mathcal{D}_X^S)$, and its transport cost is

$$\int \rho_{\mathcal{X}}(x_S, x_T) \, d\bar{\gamma}(x_S, x_T) = \int \rho_{\mathcal{X}}(x, x) \, d\mathcal{D}_X^S(x) = 0.$$

Therefore the optimal value is at most 0, hence equals 0.

Let $\gamma^* \in \Gamma(\mathcal{D}_X^S, \mathcal{D}_X^S)$ be any optimal coupling. Then

$$0 = \int \rho_{\mathcal{X}}(x_S, x_T) \, d\gamma^*(x_S, x_T).$$

Since the integrand is nonnegative, this implies $\rho_{\mathcal{X}}(x_S, x_T) = 0$ holds $\gamma^*$-almost surely, i.e., $(x_S, x_T) \in \zeta := \{(x, x) : x \in \mathcal{X}\}$ $\gamma^*$-a.s. Hence $\operatorname{supp}(\gamma^*) \subseteq \zeta$.

Finally, because $\gamma^*$ is supported on $\zeta$ and has first marginal $\mathcal{D}_X^S$, for every measurable $A \subseteq \mathcal{X} \times \mathcal{X}$ we have

$$\gamma^*(A) = \gamma^*(A \cap \zeta) = \int \mathbf{1}_{\{(x,x) \in A\}} \, d\mathcal{D}_X^S(x) = (\mathrm{Id}, \mathrm{Id})_{\#}\mathcal{D}_X^S(A),$$

which proves $\gamma^* = (\mathrm{Id}, \mathrm{Id})_{\#}\mathcal{D}_X^S$ and the stated equivalent form. $\qquad\square$

## C.6. Proof of Theorem 3.7

*Proof.* For $\beta > 0$, Lemma 3.5 ensures that the entropic optimal transport coupling $\gamma^*$ in Definition 3.1 is unique. Hence any potential ambiguity of $S_{Cpt}^{\gamma^*}$ can only come from the choice of versions of the conditional distributions.

Given any two versions of the source conditionals $\{\mathcal{D}_{Y|X=x}^S\}_{x \in \mathcal{X}}$ and $\{\widetilde{\mathcal{D}}_{Y|X=x}^S\}_{x \in \mathcal{X}}$, and any two versions of the target conditionals $\{\mathcal{D}_{Y|X=x}^T\}_{x \in \mathcal{X}}$ and $\{\widetilde{\mathcal{D}}_{Y|X=x}^T\}_{x \in \mathcal{X}}$, such that $\mathcal{D}_{Y|X=x}^S = \widetilde{\mathcal{D}}_{Y|X=x}^S$ holds $\mathcal{D}_X^S$-a.e., and $\mathcal{D}_{Y|X=x}^T = \widetilde{\mathcal{D}}_{Y|X=x}^T$

holds $\mathcal{D}_X^T$-a.e. Therefore, there exist measurable sets $E_S, E_T \subseteq \mathcal{X}$ with $\mathcal{D}_X^S(E_S) = 0$ and $\mathcal{D}_X^T(E_T) = 0$ such that the equalities hold for all $x \notin E_S$ and all $x \notin E_T$, respectively.

Define the $\gamma^*$-Y|X shift under the two choices as

$$S_{Cpt}^{\gamma^*} := \int W_1\big(\mathcal{D}_{Y|X=x_S}^S, \mathcal{D}_{Y|X=x_T}^T\big) \, d\gamma^*(x_S, x_T),$$

$$\widetilde{S}_{Cpt}^{\gamma^*} := \int W_1\big(\widetilde{\mathcal{D}}_{Y|X=x_S}^S, \widetilde{\mathcal{D}}_{Y|X=x_T}^T\big) \, d\gamma^*(x_S, x_T).$$

Let $E := (E_S \times \mathcal{X}) \cup (\mathcal{X} \times E_T)$. For any $(x_S, x_T) \in E^c$ we have $x_S \notin E_S$ and $x_T \notin E_T$, hence

$$\mathcal{D}_{Y|X=x_S}^S = \widetilde{\mathcal{D}}_{Y|X=x_S}^S, \qquad \mathcal{D}_{Y|X=x_T}^T = \widetilde{\mathcal{D}}_{Y|X=x_T}^T,$$

which implies

$$W_1\big(\mathcal{D}_{Y|X=x_S}^S, \mathcal{D}_{Y|X=x_T}^T\big) = W_1\big(\widetilde{\mathcal{D}}_{Y|X=x_S}^S, \widetilde{\mathcal{D}}_{Y|X=x_T}^T\big) \quad \text{for all } (x_S, x_T) \in E^c.$$

Consequently, the difference between the two versions satisfies

$$S_{Cpt}^{\gamma^*} - \widetilde{S}_{Cpt}^{\gamma^*} = \int_E \Big[ W_1\big(\mathcal{D}_{Y|X=x_S}^S, \mathcal{D}_{Y|X=x_T}^T\big) - W_1\big(\widetilde{\mathcal{D}}_{Y|X=x_S}^S, \widetilde{\mathcal{D}}_{Y|X=x_T}^T\big) \Big] \, d\gamma^*(x_S, x_T),$$

By Lemma 3.4, $\mathrm{supp}(\gamma^*) \subseteq \mathrm{supp}(\mathcal{D}_X^S) \times \mathrm{supp}(\mathcal{D}_X^T)$, so $\gamma^*$ never places mass outside the support product region. Furthermore, since $\gamma^* \in \Gamma(\mathcal{D}_X^S, \mathcal{D}_X^T)$, hence

$$\gamma^*(E_S \times \mathcal{X}) = \mathcal{D}_X^S(E_S) = 0, \qquad \gamma^*(\mathcal{X} \times E_T) = \mathcal{D}_X^T(E_T) = 0,$$

and therefore $\gamma^*(E) = 0$, and thus

$$S_{Cpt}^{\gamma^*} - \widetilde{S}_{Cpt}^{\gamma^*} = 0.$$

Therefore $S_{Cpt}^{\gamma^*}$ does not depend on the choice of versions of the conditional distributions. This proves that $S_{Cpt}^{\gamma^*}$ is unique, regardless of whether $\mathrm{supp}(\mathcal{D}_X^S) \neq \mathrm{supp}(\mathcal{D}_X^T)$. $\qquad \square$

## C.7. Proof of Proposition 3.8

*Proof.* Since $\beta = 0$ and $\mathcal{D}_X^S = \mathcal{D}_X^T$, Lemma 3.6 gives that the optimal coupling collapses to the diagonal:

$$\gamma^* = (\mathrm{Id}, \mathrm{Id})_\# D_X^S.$$

Under deterministic labeling and $\rho_{\mathcal{Y}} = |\cdot|$, Corollary 3.3 yields

$$S_{Cpt}^{\gamma^*} = \mathbb{E}_{(x_S, x_T) \sim \gamma^*} \big[ |f_S(x_S) - f_T(x_T)| \big].$$

Substituting $\gamma^* = (\mathrm{Id}, \mathrm{Id})_\# D_X^S$ implies $(x_S, x_T) = (x, x)$ with $x \sim D_X^S$, hence

$$S_{Cpt}^{\gamma^*} = \mathbb{E}_{x \sim D_X^S} \big[ |f_S(x) - f_T(x)| \big].$$

Finally, since $\mathcal{D}_X^S = \mathcal{D}_X^T$, we obtain

$$S_{Cpt}^{\gamma^*} = \mathbb{E}_{x \sim \mathcal{D}_X^S} \big[ |f_S(x) - f_T(x)| \big] = \mathbb{E}_{x \sim \mathcal{D}_X^T} \big[ |f_S(x) - f_T(x)| \big],$$

as claimed. $\qquad \square$

## C.8. Proof of Corollary 3.10

*Proof.* Take any $y_1, y_2 \in \mathcal{Y}$ and $x_1, x_2 \in \mathcal{X}$. By the separate $(L_\ell, L'_\ell)$-Lipschitzness of $\ell$,

$$\big| \ell(y_1, h(x_1)) - \ell(y_2, h(x_2)) \big| \leq L_\ell \, \rho_{\mathcal{Y}}(y_1, y_2) + L'_\ell \, \rho_{\mathcal{Y}'}\big(h(x_1), h(x_2)\big).$$

By the $L_h$-Lipschitzness of $h$, we further have $\rho_{\mathcal{Y}'}\big(h(x_1), h(x_2)\big) \leq L_h \, \rho_{\mathcal{X}}(x_1, x_2)$. Substituting this bound yields

$$\big| \ell(y_1, h(x_1)) - \ell(y_2, h(x_2)) \big| \leq L_\ell \, \rho_{\mathcal{Y}}(y_1, y_2) + (L_h L'_\ell) \, \rho_{\mathcal{X}}(x_1, x_2),$$

which is exactly the separate $(L_\ell, \, L_h L'_\ell)$-Lipschitz condition for the composite function $\ell\big(y, h(x)\big)$. $\qquad \square$

## C.9. Proof of Lemma 3.11

*Proof.* Write the two expectations explicitly:

$$\mathbb{E}_{\mathcal{D}_{XY}^S}[\mathcal{G}] = \int_{\mathcal{X}\times\mathcal{Y}} \mathcal{G}(y_S, x_S)\, d\mathcal{D}_{XY}^S(x_S, y_S), \qquad \mathbb{E}_{\mathcal{D}_{XY}^T}[\mathcal{G}] = \int_{\mathcal{X}\times\mathcal{Y}} \mathcal{G}(y_T, x_T)\, d\mathcal{D}_{XY}^T(x_T, y_T).$$

Let $\gamma_{XY} \in \Gamma(\mathcal{D}_{XY}^S, \mathcal{D}_{XY}^T)$ be any coupling on $(\mathcal{X}\times\mathcal{Y})\times(\mathcal{X}\times\mathcal{Y})$, i.e., its first marginal is $\mathcal{D}_{XY}^S$ and second marginal is $\mathcal{D}_{XY}^T$. Hence, for any integrable functions $\varphi, \psi$,

$$\int \varphi(x_S, y_S)\, d\gamma_{XY}(x_S, y_S, x_T, y_T) = \int \varphi(x_S, y_S)\, d\mathcal{D}_{XY}^S(x_S, y_S),$$

$$\int \psi(x_T, y_T)\, d\gamma_{XY}(x_S, y_S, x_T, y_T) = \int \psi(x_T, y_T)\, d\mathcal{D}_{XY}^T(x_T, y_T).$$

Applying these identities to $\varphi(x_S, y_S) = \mathcal{G}(y_S, x_S)$ and $\psi(x_T, y_T) = \mathcal{G}(y_T, x_T)$ gives

$$\mathbb{E}_{\mathcal{D}_{XY}^S}[\mathcal{G}] - \mathbb{E}_{\mathcal{D}_{XY}^T}[\mathcal{G}] = \int \mathcal{G}(y_S, x_S)\, d\mathcal{D}_{XY}^S(x_S, y_S) - \int \mathcal{G}(y_T, x_T)\, d\mathcal{D}_{XY}^T(x_T, y_T)$$

$$= \int \mathcal{G}(y_S, x_S)\, d\gamma_{XY}(x_S, y_S, x_T, y_T) - \int \mathcal{G}(y_T, x_T)\, d\gamma_{XY}(x_S, y_S, x_T, y_T)$$

$$= \int \Big(\mathcal{G}(y_S, x_S) - \mathcal{G}(y_T, x_T)\Big)\, d\gamma_{XY}(x_S, y_S, x_T, y_T).$$

Taking absolute values and using $|\int f\, d\mu| \leq \int |f|\, d\mu$ yields

$$\Big|\mathbb{E}_{\mathcal{D}_{XY}^S}[\mathcal{G}] - \mathbb{E}_{\mathcal{D}_{XY}^T}[\mathcal{G}]\Big| \leq \int \Big|\mathcal{G}(y_S, x_S) - \mathcal{G}(y_T, x_T)\Big|\, d\gamma_{XY}(x_S, y_S, x_T, y_T).$$

By the separate $(L_{\mathcal{Y}}, L_{\mathcal{X}})$-Lipschitz property of $\mathcal{G}$, we have

$$\Big|\mathcal{G}(y_S, x_S) - \mathcal{G}(y_T, x_T)\Big| \leq L_{\mathcal{Y}}\rho_{\mathcal{Y}}(y_S, y_T) + L_{\mathcal{X}}\rho_{\mathcal{X}}(x_S, x_T).$$

Substituting and splitting the integral,

$$\Big|\mathbb{E}_{\mathcal{D}_{XY}^S}[\mathcal{G}] - \mathbb{E}_{\mathcal{D}_{XY}^T}[\mathcal{G}]\Big| \leq \int \Big(L_{\mathcal{Y}}\rho_{\mathcal{Y}}(y_S, y_T) + L_{\mathcal{X}}\rho_{\mathcal{X}}(x_S, x_T)\Big)\, d\gamma_{XY}(x_S, y_S, x_T, y_T)$$

$$= L_{\mathcal{Y}} \int \rho_{\mathcal{Y}}(y_S, y_T)\, d\gamma_{XY}(x_S, y_S, x_T, y_T) + L_{\mathcal{X}} \int \rho_{\mathcal{X}}(x_S, x_T)\, d\gamma_{XY}(x_S, y_S, x_T, y_T)$$

$$= L_{\mathcal{Y}}\, \mathbb{E}_{\gamma_{XY}}\big[\rho_{\mathcal{Y}}(y_S, y_T)\big] + L_{\mathcal{X}}\, \mathbb{E}_{\gamma_{XY}}\big[\rho_{\mathcal{X}}(x_S, x_T)\big],$$

as claimed. $\qquad\square$

## C.10. Proof of Lemma 3.12

*Proof.* We show that the first marginal of $\gamma_{XY}$ equals $\mathcal{D}_{XY}^S$ and the second marginal equals $\mathcal{D}_{XY}^T$.

**First marginal.** Take any measurable $A \subseteq \mathcal{X}$ and $B \subseteq \mathcal{Y}$. By the definition of $\gamma_{XY}$,

$$\gamma_{XY}\big((A\times B)\times(\mathcal{X}\times\mathcal{Y})\big) = \int_{(x_S, x_T)\in\mathcal{X}\times\mathcal{X}} \gamma_{Y|(x_S, x_T)}^*\big((B\times\mathcal{Y})\big)\, \mathbf{1}_{\{x_S\in A\}}\, d\gamma^*(x_S, x_T).$$

For each fixed $(x_S, x_T)$, the coupling $\gamma_{Y|(x_S, x_T)}^* \in \Gamma(\mathcal{D}_{Y|X=x_S}^S, \mathcal{D}_{Y|X=x_T}^T)$ has first marginal $\mathcal{D}_{Y|X=x_S}^S$, hence

$$\gamma_{Y|(x_S, x_T)}^*(B\times\mathcal{Y}) = \mathcal{D}_{Y|X=x_S}^S(B).$$

Substituting gives

$$\gamma_{XY}\big((A\times B)\times(\mathcal{X}\times\mathcal{Y})\big) = \int \mathbf{1}_{\{x_S\in A\}}\, \mathcal{D}_{Y|X=x_S}^S(B)\, d\gamma^*(x_S, x_T).$$

Finally, since the first marginal of $\gamma^*$ is $\mathcal{D}_X^S$, integrating out $x_T$ yields

$$\gamma_{XY}\big((A\times B)\times(\mathcal{X}\times\mathcal{Y})\big) = \int_{x_S\in A} \mathcal{D}_{Y|X=x_S}^S(B)\, d\mathcal{D}_X^S(x_S) = \mathcal{D}_{XY}^S(A\times B),$$

**Second marginal.** Similarly, for measurable $A \subseteq \mathcal{X}$ and $B \subseteq \mathcal{Y}$,

$$\gamma_{XY}\big((\mathcal{X} \times \mathcal{Y}) \times (A \times B)\big) = \int \gamma^*_{Y|(x_S,x_T)}(\mathcal{Y} \times B)\,\mathbf{1}_{\{x_T \in A\}}\,d\gamma^*(x_S,x_T)$$

$$= \int \mathbf{1}_{\{x_T \in A\}}\,\mathcal{D}^T_{Y|X=x_T}(B)\,d\gamma^*(x_S,x_T)$$

$$= \int_{x_T \in A} \mathcal{D}^T_{Y|X=x_T}(B)\,d\mathcal{D}^T_X(x_T) = \mathcal{D}^T_{XY}(A \times B).$$

Therefore $\gamma_{XY} \in \Gamma(\mathcal{D}^S_{XY}, \mathcal{D}^T_{XY})$. $\qquad\square$

### C.11. Proof of Theorem 3.13

*Proof.* Define the composite function

$$\mathcal{G}(y,x) := \ell\big(y, h(x)\big), \qquad (y,x) \in \mathcal{Y} \times \mathcal{X}.$$

By Corollary 3.10, $\mathcal{G}$ is separately $(L_\ell,\ L_h L'_\ell)$-Lipschitz, i.e.,

$$\big|\mathcal{G}(y_1,x_1) - \mathcal{G}(y_2,x_2)\big| \le L_\ell\,\rho_{\mathcal{Y}}(y_1,y_2) + (L_h L'_\ell)\,\rho_{\mathcal{X}}(x_1,x_2).$$

Hence

$$\epsilon_T(h) = \epsilon_S(h) + \big(\epsilon_T(h) - \epsilon_S(h)\big)$$
$$\le \epsilon_S(h) + \big|\epsilon_T(h) - \epsilon_S(h)\big|$$
$$= \epsilon_S(h) + \big|\mathbb{E}_{\mathcal{D}^T_{XY}}[\mathcal{G}] - \mathbb{E}_{\mathcal{D}^S_{XY}}[\mathcal{G}]\big|.$$

**Step 1: construct a specific joint coupling.** Let $\gamma^* \in \Gamma(\mathcal{D}^S_X, \mathcal{D}^T_X)$ be the optimal coupling in Definition 3.1. For each $(x_S, x_T)$, let $\gamma^*_{Y|(x_S,x_T)} \in \Gamma(\mathcal{D}^S_{Y|X=x_S}, \mathcal{D}^T_{Y|X=x_T})$ be an optimal coupling of $S_{pair}(x_S, x_T)$ in Definition 3.2, define the joint distribution of couplings

$$\gamma_{XY}(dx_S, dx_T, dy_S, dy_T) := \gamma^*(dx_S, dx_T)\,\gamma^*_{Y|(x_S,x_T)}(dy_S, dy_T).$$

By Lemma 3.12, $\gamma_{XY} \in \Gamma(\mathcal{D}^S_{XY}, \mathcal{D}^T_{XY})$.

**Step 2: apply Lemma 3.11.** Apply Lemma 3.11 to $\mathcal{G}$ with the coupling $\gamma_{XY}$, we obtain:

$$\big|\mathbb{E}_{\mathcal{D}^S_{XY}}[\mathcal{G}] - \mathbb{E}_{\mathcal{D}^T_{XY}}[\mathcal{G}]\big| \le (L_h L'_\ell)\,\mathbb{E}_{\gamma_{XY}}\big[\rho_{\mathcal{X}}(x_S, x_T)\big] + L_\ell\,\mathbb{E}_{\gamma_{XY}}\big[\rho_{\mathcal{Y}}(y_S, y_T)\big].$$

It remains to bound the two expectations on the right-hand side.

**Step 3: bound the $\mathcal{X}$-term by $S_{Cov}$.** By the construction of $\gamma_{XY}$,

$$\mathbb{E}_{\gamma_{XY}}\big[\rho_{\mathcal{X}}(x_S, x_T)\big] = \int \rho_{\mathcal{X}}(x_S, x_T)\,\gamma^*(dx_S, dx_T)\,\gamma^*_{Y|(x_S,x_T)}(dy_S, dy_T)$$

$$= \int \rho_{\mathcal{X}}(x_S, x_T)\,\gamma^*(dx_S, dx_T) = \mathbb{E}_{\gamma^*}\big[\rho_{\mathcal{X}}(x_S, x_T)\big].$$

Since $S_{Cov} = W_\beta(\mathcal{D}^S_X, \mathcal{D}^T_X)$ and $\gamma^*$ is optimal,

$$S_{Cov} = \int \rho_{\mathcal{X}}(x_S, x_T)\,d\gamma^*(x_S, x_T) + \beta\,H\big(\gamma^* \mid \mathcal{D}^S_X \otimes \mathcal{D}^T_X\big) \ge \int \rho_{\mathcal{X}}(x_S, x_T)\,d\gamma^*(x_S, x_T),$$

because $\beta \ge 0$ and relative entropy is nonnegative. Therefore,

$$\mathbb{E}_{\gamma_{XY}}\big[\rho_{\mathcal{X}}(x_S, x_T)\big] = \mathbb{E}_{\gamma^*}\big[\rho_{\mathcal{X}}(x_S, x_T)\big] \le S_{Cov}.$$

**Step 4: identify the $\mathcal{Y}$-term with $S_{Cpt}^{\gamma^*}$.** Again by the construction of $\gamma_{XY}$,

$$\mathbb{E}_{\gamma_{XY}}\left[\rho_{\mathcal{Y}}(y_S, y_T)\right] = \int \rho_{\mathcal{Y}}(y_S, y_T)\, \gamma^*(dx_S, dx_T)\, \gamma^*_{Y|(x_S, x_T)}(dy_S, dy_T)$$

$$= \int_{\mathcal{X} \times \mathcal{X}} \left[\int_{\mathcal{Y} \times \mathcal{Y}} \rho_{\mathcal{Y}}(y_S, y_T)\, d\gamma^*_{Y|(x_S, x_T)}(y_S, y_T)\right] d\gamma^*(x_S, x_T)$$

$$= \int_{\mathcal{X} \times \mathcal{X}} W_1\big(\mathcal{D}^S_{Y|X=x_S}, \mathcal{D}^T_{Y|X=x_T}\big)\, d\gamma^*(x_S, x_T)$$

$$= \mathbb{E}_{(x_S, x_T) \sim \gamma^*}\left[S_{pair}(x_S, x_T)\right] = S_{Cpt}^{\gamma^*},$$

**Step 5: conclude the bound.** Combining Steps 2–4 yields

$$\left|\mathbb{E}_{\mathcal{D}^S_{XY}}[\mathcal{G}] - \mathbb{E}_{\mathcal{D}^T_{XY}}[\mathcal{G}]\right| \leq (L_h L'_\ell)\, S_{Cov} + L_\ell\, S_{Cpt}^{\gamma^*}.$$

Substituting this into the earlier inequality for $\epsilon_T(h)$ gives

$$\epsilon_T(h) \leq \epsilon_S(h) + (L_h L'_\ell)\, S_{Cov} + L_\ell\, S_{Cpt}^{\gamma^*},$$

which proves the theorem. $\qquad\square$

## C.12. Proof of Theorem 4.2

*Proof.* We work with $\beta = 0$, hence $W_\beta = W_1$. For brevity, denote

$$\mu := \mathcal{D}^S_X, \qquad \nu := \mathcal{D}^T_X, \qquad c := W_1(\mu, \nu).$$

Let $\hat{\mu}', \hat{\mu}''$ (resp. $\hat{\nu}', \hat{\nu}''$) be the two half-sample empirical measures constructed in Definition 4.1. They are independent within each domain because they are built from disjoint halves of i.i.d. samples.

**A useful inequality.** We will use the elementary fact: for any $a, b \geq 0$,

$$\left|\sqrt{a} - \sqrt{b}\right| \leq \sqrt{|a - b|}.$$

Indeed, if $a \geq b$ then $\sqrt{a} - \sqrt{b} = \frac{a-b}{\sqrt{a}+\sqrt{b}} \leq \frac{a-b}{\sqrt{a-b}} = \sqrt{a - b}$, and the case $b \geq a$ is symmetric.

**Step 1: rewrite the debiased estimator and introduce an auxiliary statistic.** Define

$$S := \tfrac{1}{2} W_1(\hat{\mu}', \hat{\nu}')^2 + \tfrac{1}{2} W_1(\hat{\mu}'', \hat{\nu}'')^2 - \tfrac{1}{2} W_1(\hat{\mu}', \hat{\mu}'')^2 - \tfrac{1}{2} W_1(\hat{\nu}', \hat{\nu}'')^2.$$

Then Definition 4.1 (with $\beta = 0$) gives

$$W_1^{deb}\big(\widehat{\mathcal{D}^S_X}, \widehat{\mathcal{D}^T_X}\big) = \sqrt{|S|}.$$

Introduce the auxiliary random variable

$$T := \left|S - c^2\right|.$$

**Step 2: reduce $\{|W_1^{deb} - c| > \varepsilon\}$ to an event on $T$.** First note the deterministic inequality

$$\left||S| - c^2\right| \leq |S - c^2| = T,$$

which holds because $c^2 \geq 0$. Hence, using the inequality in the first paragraph,

$$|W_1^{deb} - c| = \left|\sqrt{|S|} - \sqrt{c^2}\right| \leq \sqrt{\left||S| - c^2\right|} \leq \sqrt{T}.$$

Therefore, for any $\varepsilon > 0$,

$$\mathbb{P}\big(|W_1^{deb} - c| > \varepsilon\big) \leq \mathbb{P}\big(\sqrt{T} > \varepsilon\big) = \mathbb{P}\big(T > \varepsilon^2\big).$$

This bound is always valid and is particularly useful when $c$ is small.

When $c \geq \varepsilon$, we can obtain a complementary reduction by squaring the deviation event. Indeed, $|W_1^{deb} - c| > \varepsilon$ implies either $W_1^{deb} > c + \varepsilon$ or $W_1^{deb} < c - \varepsilon$. In both cases,

$$\left|(W_1^{deb})^2 - c^2\right| > (c + \varepsilon)^2 - c^2 = 2c\varepsilon + \varepsilon^2 \quad \text{or} \quad c^2 - (c - \varepsilon)^2 = 2c\varepsilon - \varepsilon^2,$$

hence in particular

$$\left|(W_1^{deb})^2 - c^2\right| > 2c\varepsilon - \varepsilon^2.$$

Since $(W_1^{deb})^2 = |S|$, we have

$$\{|W_1^{deb} - c| > \varepsilon\} \subseteq \{\left||S| - c^2\right| > 2c\varepsilon - \varepsilon^2\} \subseteq \{T > 2c\varepsilon - \varepsilon^2\}.$$

Combining both regimes, define the threshold

$$t(c, \varepsilon) := \begin{cases} \varepsilon^2, & c < \varepsilon, \\ 2c\varepsilon - \varepsilon^2, & c \geq \varepsilon, \end{cases}$$

so that for all $c \geq 0$,

$$\mathbb{P}\left(|W_1^{deb} - c| > \varepsilon\right) \leq \mathbb{P}\left(T > t(c, \varepsilon)\right).$$

**Step 3: bound $T$ by marginal empirical Wasserstein errors.** Start from the definition of $T$ and use the triangle inequality for $|\cdot|$:

$$T = \left|\tfrac{1}{2} W_1(\hat{\mu}', \hat{\nu}')^2 + \tfrac{1}{2} W_1(\hat{\mu}'', \hat{\nu}'')^2 - \tfrac{1}{2} W_1(\hat{\mu}', \hat{\mu}'')^2 - \tfrac{1}{2} W_1(\hat{\nu}', \hat{\nu}'')^2 - c^2\right|$$

$$\leq \tfrac{1}{2}\left|W_1(\hat{\mu}', \hat{\nu}')^2 - c^2\right| + \tfrac{1}{2}\left|W_1(\hat{\mu}'', \hat{\nu}'')^2 - c^2\right| + \tfrac{1}{2} W_1(\hat{\mu}', \hat{\mu}'')^2 + \tfrac{1}{2} W_1(\hat{\nu}', \hat{\nu}'')^2.$$

*Step 3.1: control the cross-domain square terms.* Let $u := W_1(\hat{\mu}', \hat{\nu}')$. Then

$$|u^2 - c^2| = |u - c|\,|u + c| \leq |u - c|\,(|u - c| + 2c) = 2c|u - c| + |u - c|^2.$$

Multiplying by $\tfrac{1}{2}$ gives

$$\tfrac{1}{2}|u^2 - c^2| \leq c|u - c| + \tfrac{1}{2}|u - c|^2.$$

Applying this with $u = W_1(\hat{\mu}', \hat{\nu}')$ and again with $u = W_1(\hat{\mu}'', \hat{\nu}'')$ yields

$$\tfrac{1}{2}\left|W_1(\hat{\mu}', \hat{\nu}')^2 - c^2\right| \leq c\left|W_1(\hat{\mu}', \hat{\nu}') - c\right| + \tfrac{1}{2}\left|W_1(\hat{\mu}', \hat{\nu}') - c\right|^2,$$

$$\tfrac{1}{2}\left|W_1(\hat{\mu}'', \hat{\nu}'')^2 - c^2\right| \leq c\left|W_1(\hat{\mu}'', \hat{\nu}'') - c\right| + \tfrac{1}{2}\left|W_1(\hat{\mu}'', \hat{\nu}'') - c\right|^2.$$

*Step 3.2: relate $|W_1(\hat{\mu}', \hat{\nu}') - c|$ to $W_1(\mu, \hat{\mu}')$ and $W_1(\nu, \hat{\nu}')$.* Using the triangle inequality for $W_1$,

$$W_1(\hat{\mu}', \hat{\nu}') \leq W_1(\hat{\mu}', \mu) + W_1(\mu, \nu) + W_1(\nu, \hat{\nu}') = W_1(\hat{\mu}', \mu) + c + W_1(\nu, \hat{\nu}'),$$

hence $W_1(\hat{\mu}', \hat{\nu}') - c \leq W_1(\hat{\mu}', \mu) + W_1(\nu, \hat{\nu}')$. Similarly,

$$c = W_1(\mu, \nu) \leq W_1(\mu, \hat{\mu}') + W_1(\hat{\mu}', \hat{\nu}') + W_1(\hat{\nu}', \nu),$$

so $c - W_1(\hat{\mu}', \hat{\nu}') \leq W_1(\mu, \hat{\mu}') + W_1(\nu, \hat{\nu}')$. Combining the two inequalities,

$$\left|W_1(\hat{\mu}', \hat{\nu}') - c\right| \leq W_1(\mu, \hat{\mu}') + W_1(\nu, \hat{\nu}').$$

The same argument gives

$$\left|W_1(\hat{\mu}'', \hat{\nu}'') - c\right| \leq W_1(\mu, \hat{\mu}'') + W_1(\nu, \hat{\nu}'').$$

*Step 3.3: relate $W_1(\hat{\mu}', \hat{\mu}'')$ and $W_1(\hat{\nu}', \hat{\nu}'')$ to the same marginal errors.* Again by the triangle inequality,

$$W_1(\hat{\mu}', \hat{\mu}'') \leq W_1(\hat{\mu}', \mu) + W_1(\mu, \hat{\mu}''), \qquad W_1(\hat{\nu}', \hat{\nu}'') \leq W_1(\hat{\nu}', \nu) + W_1(\nu, \hat{\nu}'').$$

*Step 3.4: assemble the bound.* Introduce the shorthand nonnegative random variables

$$u' := W_1(\mu, \hat{\mu}'), \quad u'' := W_1(\mu, \hat{\mu}''), \quad v' := W_1(\nu, \hat{\nu}'), \quad v'' := W_1(\nu, \hat{\nu}'').$$

Then Steps 3.1–3.3 imply

$$T \leq c(u' + v') + \tfrac{1}{2}(u' + v')^2 + c(u'' + v'') + \tfrac{1}{2}(u'' + v'')^2 + \tfrac{1}{2}(u' + u'')^2 + \tfrac{1}{2}(v' + v'')^2.$$

Now use $(a + b)^2 \leq 2a^2 + 2b^2$ repeatedly to simplify:

$$\tfrac{1}{2}(u' + v')^2 \leq u'^2 + v'^2, \qquad \tfrac{1}{2}(u'' + v'')^2 \leq u''^2 + v''^2, \qquad \tfrac{1}{2}(u' + u'')^2 \leq u'^2 + u''^2, \qquad \tfrac{1}{2}(v' + v'')^2 \leq v'^2 + v''^2.$$

Substituting yields the clean bound

$$T \leq c(u' + u'' + v' + v'') + 2\big(u'^2 + u''^2 + v'^2 + v''^2\big).$$

In particular, on the event $\{u' \leq \varepsilon_0,\ u'' \leq \varepsilon_0,\ v' \leq \varepsilon_0,\ v'' \leq \varepsilon_0\}$, we have

$$T \leq 4c\varepsilon_0 + 8\varepsilon_0^2.$$

Hence,

$$\{T > 4c\varepsilon_0 + 8\varepsilon_0^2\} \subseteq \{u' > \varepsilon_0\} \cup \{u'' > \varepsilon_0\} \cup \{v' > \varepsilon_0\} \cup \{v'' > \varepsilon_0\},$$

and by the union bound,

$$\mathbb{P}\big(T > 4c\varepsilon_0 + 8\varepsilon_0^2\big) \leq \mathbb{P}(u' > \varepsilon_0) + \mathbb{P}(u'' > \varepsilon_0) + \mathbb{P}(v' > \varepsilon_0) + \mathbb{P}(v'' > \varepsilon_0).$$

**Step 4: apply traditional concentration inequality.** By assumption, $\mu$ and $\nu$ have finite squared-exponential moments on $\mathbb{R}^d$. Lemma B.3 and B.4 implies that there exist constants $\lambda_S, \lambda_T > 0$, depending only on the squared-exponential moments of $\mu$ and $\nu$ respectively, such that for any $\varepsilon_0 > 0$ there exists $N$ with the following property: whenever $N_S/2 \geq N$ and $N_T/2 \geq N$,

$$\mathbb{P}\big(W_1(\mu, \hat{\mu}') > \varepsilon_0\big) \leq \exp\Big(-\tfrac{\lambda_S N_S \varepsilon_0^2}{4}\Big), \qquad \mathbb{P}\big(W_1(\mu, \hat{\mu}'') > \varepsilon_0\big) \leq \exp\Big(-\tfrac{\lambda_S N_S \varepsilon_0^2}{4}\Big),$$

and similarly

$$\mathbb{P}\big(W_1(\nu, \hat{\nu}') > \varepsilon_0\big) \leq \exp\Big(-\tfrac{\lambda_T N_T \varepsilon_0^2}{4}\Big), \qquad \mathbb{P}\big(W_1(\nu, \hat{\nu}'') > \varepsilon_0\big) \leq \exp\Big(-\tfrac{\lambda_T N_T \varepsilon_0^2}{4}\Big).$$

Therefore, for $N_S, N_T$ large enough,

$$\mathbb{P}\big(T > 4c\varepsilon_0 + 8\varepsilon_0^2\big) \leq 2\exp\Big(-\tfrac{\lambda_S N_S \varepsilon_0^2}{4}\Big) + 2\exp\Big(-\tfrac{\lambda_T N_T \varepsilon_0^2}{4}\Big).$$

**Step 5: choose $\varepsilon_0$ to match the deviation level $\varepsilon$.** Recall from Step 2 that

$$\mathbb{P}\big(|W_1^{deb} - c| > \varepsilon\big) \leq \mathbb{P}\big(T > t(c, \varepsilon)\big), \qquad t(c, \varepsilon) = \begin{cases} \varepsilon^2, & c < \varepsilon, \\ 2c\varepsilon - \varepsilon^2, & c \geq \varepsilon. \end{cases}$$

We now select $\varepsilon_0$ so that

$$4c\varepsilon_0 + 8\varepsilon_0^2 = t(c, \varepsilon).$$

Let $a := c/\varepsilon \geq 0$. Solving this quadratic in the two regimes yields the closed-form expression

$$\varepsilon_0^2 = \frac{V(a)\,\varepsilon^2}{8},$$

where

$$V(a) = \begin{cases} a^2 + 1 - a\sqrt{a^2 + 2}, & a < 1, \\ a^2 + 2a - 1 - a\sqrt{a^2 + 4a - 2}, & a \geq 1. \end{cases}$$

Define $V_\varepsilon := V(c/\varepsilon) = V(W_1(\mu, \nu)/\varepsilon)$. As shown in the "Range of $V(a)$" derivation, $V_\varepsilon \in [2 - \sqrt{3}, 2)$ and depends only on $c/\varepsilon$.

With this choice, $\{T > t(c, \varepsilon)\} = \{T > 4c\varepsilon_0 + 8\varepsilon_0^2\}$, hence

$$\mathbb{P}(|W_1^{deb} - c| > \varepsilon) \leq \mathbb{P}(T > 4c\varepsilon_0 + 8\varepsilon_0^2)$$

$$\leq 2\exp\left(-\frac{\lambda_S N_S \varepsilon_0^2}{4}\right) + 2\exp\left(-\frac{\lambda_T N_T \varepsilon_0^2}{4}\right)$$

$$= 2\exp\left(-\frac{\lambda_S N_S V_\varepsilon \varepsilon^2}{32}\right) + 2\exp\left(-\frac{\lambda_T N_T V_\varepsilon \varepsilon^2}{32}\right).$$

Finally, note $c = W_1(\mu, \nu) = W_\beta(\mathcal{D}_X^S, \mathcal{D}_X^T)$ when $\beta = 0$, and $W_1^{deb} = W_\beta^{deb}$ when $\beta = 0$, so the above inequality is exactly (10). This completes the proof.

**Range of $V(a)$.** Recall

$$V(a) = \begin{cases} V_1(a) := a^2 + 1 - a\sqrt{a^2 + 2}, & 0 \leq a < 1, \\ V_2(a) := a^2 + 2a - 1 - a\sqrt{a^2 + 4a - 2}, & a \geq 1. \end{cases}$$

First, the two branches agree at $a = 1$:

$$V_1(1) = 2 - \sqrt{3}, \qquad V_2(1) = 2 - \sqrt{3}.$$

**(i) Monotonicity on $[0, 1]$.** Differentiate $V_1$ on $(0, 1)$:

$$V_1'(a) = 2a - \sqrt{a^2 + 2} - \frac{a^2}{\sqrt{a^2 + 2}} = 2a - \frac{2(a^2 + 1)}{\sqrt{a^2 + 2}}.$$

For $a \geq 0$, we have $a\sqrt{a^2 + 2} \leq a^2 + 1$ since

$$(a^2 + 1)^2 - a^2(a^2 + 2) = 1 > 0.$$

Thus $\frac{a^2 + 1}{\sqrt{a^2 + 2}} \geq a$, which implies $V_1'(a) \leq 0$ on $(0, 1)$. Hence $V_1$ is decreasing on $[0, 1]$, so

$$V_1(a) \in [V_1(1), V_1(0)] = [2 - \sqrt{3}, 1].$$

**(ii) Monotonicity on $[1, \infty)$.** Let $s(a) := \sqrt{a^2 + 4a - 2}$. Then for $a > 1$,

$$V_2'(a) = 2a + 2 - s(a) - \frac{a(a + 2)}{s(a)}.$$

Since $s(a) > 0$, the inequality $V_2'(a) \geq 0$ is equivalent to

$$(a + 1)s(a) \geq a^2 + 3a - 1.$$

Squaring both sides (both sides are nonnegative for $a \geq 1$) gives

$$(a + 1)^2(a^2 + 4a - 2) \geq (a^2 + 3a - 1)^2,$$

and the difference factors as

$$(a + 1)^2(a^2 + 4a - 2) - (a^2 + 3a - 1)^2 = 3(2a - 1) \geq 0 \qquad (a \geq 1).$$

Therefore $V_2'(a) \geq 0$ for all $a \geq 1$, i.e., $V_2$ is increasing on $[1, \infty)$. In particular,

$$V_2(a) \geq V_2(1) = 2 - \sqrt{3} \qquad (a \geq 1).$$

**(iii) Upper bound $V(a) < 2$ and $\lim_{a \to \infty} V(a) = 2$.** For $a \geq 1$, write $s(a) = \sqrt{(a+2)^2 - 6}$. Then

$$(a + 2 - s(a))(a + 2 + s(a)) = (a+2)^2 - s(a)^2 = 6, \quad \text{so} \quad a + 2 - s(a) = \frac{6}{a + 2 + s(a)}.$$

Using $V_2(a) = a(a+2) - 1 - a\, s(a)$, we get

$$2 - V_2(a) = 3 - a\,(a + 2 - s(a)) = 3 - \frac{6a}{a + 2 + s(a)}.$$

Since $s(a) > a - 2$ for $a \geq 1$ (indeed $s(a) = \sqrt{a^2 + 4a - 2} > a$), we have $a + 2 + s(a) > 2a$, hence $\frac{6a}{a+2+s(a)} < 3$, which implies $2 - V_2(a) > 0$ and thus $V_2(a) < 2$ for every finite $a$. Moreover, as $a \to \infty$, one has $a + 2 + s(a) \sim 2a$, hence $\frac{6a}{a+2+s(a)} \to 3$ and therefore $V_2(a) \to 2$.

**Conclusion.** Combining (i)–(iii), the global minimum is attained at $a = 1$ with

$$\min_{a \geq 0} V(a) = 2 - \sqrt{3},$$

and the supremum equals 2 but is not attained:

$$V(a) \in [\,2 - \sqrt{3},\ 2\,), \qquad a \geq 0.$$

$\square$

### C.13. Proof of Lemma 4.4

*Proof.* We follow (Eckstein & Nutz, 2022). Recall that the entropically regularized OT problem is

$$S_{\text{ent}}^{\varepsilon}(\mu, \nu, c) := \inf_{\pi \in \Pi(\mu,\nu)} \int c\, d\pi + \varepsilon\, \text{KL}(\pi \,\|\, \mu \otimes \nu),$$

and for fixed $\varepsilon > 0$ one may assume $\varepsilon = 1$ by dividing the objective by $\varepsilon$ and using the cost $c/\varepsilon$.

**Step 1 (Reduction to $\varepsilon = 1$ and identification of $L = 1/\beta$).** Let $c(x_1, x_2) := \rho_{\mathcal{X}}(x_1, x_2)$. The $\beta$-entropic OT objective

$$\int c\, d\pi + \beta\, \text{KL}(\pi \,\|\, \mu \otimes \nu)$$

has the *same optimizer* as

$$\int \frac{1}{\beta} c\, d\pi + \text{KL}(\pi \,\|\, \mu \otimes \nu),$$

since the two objectives differ only by a multiplicative factor $\beta$. Hence the optimal coupling for $W_\beta(\mu, \nu)$ equals the optimizer of $S_{\text{ent}}^1(\mu, \nu, c_\beta)$ with $c_\beta := c/\beta$.

Consider the product space $\mathcal{X} \times \mathcal{X}$ with metric

$$\rho\big((x_1, x_2), (x_1', x_2')\big) := \rho_{\mathcal{X}}(x_1, x_1') + \rho_{\mathcal{X}}(x_2, x_2').$$

By the triangle inequality,

$$\big|c(x_1, x_2) - c(x_1', x_2')\big| = \big|\rho_{\mathcal{X}}(x_1, x_2) - \rho_{\mathcal{X}}(x_1', x_2')\big| \leq \rho_{\mathcal{X}}(x_1, x_1') + \rho_{\mathcal{X}}(x_2, x_2') = \rho((x_1, x_2), (x_1', x_2')),$$

so $c$ is 1-Lipschitz w.r.t. $\rho$, and therefore $c_\beta = c/\beta$ satisfies the Lipschitz condition (AL) of Eckstein & Nutz (2022) with constant

$$L = \text{Lip}(c_\beta) = \frac{1}{\beta}.$$

**Step 2 (Apply Theorem 3.11).** Let $\mu := \mathcal{D}_X^S$, $\nu := \mathcal{D}_X^T$, and similarly $\hat{\mu} := \widehat{\mathcal{D}_X^S}$, $\hat{\nu} := \widehat{\mathcal{D}_X^T}$. Let $\gamma^*$ and $\hat{\gamma}^*$ be the optimizers of $S_{\text{ent}}^1(\mu, \nu, c_\beta)$ and $S_{\text{ent}}^1(\hat{\mu}, \hat{\nu}, c_\beta)$, respectively.

Assume $\mu$ and $\nu$ have finite squared-exponential moments. Then by Lemma 3.10(ii) in Eckstein & Nutz (2022), the pair of marginals satisfies the inequality (I$_1$) with some finite constant $C_1 > 0$ depending only on these squared-exponential moments.

Now apply Theorem 3.11 of Eckstein & Nutz (2022) with $N = 2$ and $p = q = 1$. Since $N^{1/q - 1/p} = 1$, we obtain

$$W_1(\gamma^*, \hat{\gamma}^*) \leq \Delta + C_1 (2L\Delta)^{1/2}, \qquad \Delta := W_1\big((\mu, \nu); (\hat{\mu}, \hat{\nu})\big),$$

.

**Step 3 (Upper bound $\Delta$ by $\Lambda$).** Let $\pi_S$ be an optimal coupling between $\mu$ and $\hat{\mu}$, and $\pi_T$ an optimal coupling between $\nu$ and $\hat{\nu}$. Then $\pi_S \otimes \pi_T$ is a coupling between $(\mu, \nu)$ and $(\hat{\mu}, \hat{\nu})$ on $\mathcal{X} \times \mathcal{X}$, and its expected $\rho$-cost equals

$$\int \rho_{\mathcal{X}}(x, x') \, d\pi_S(x, x') + \int \rho_{\mathcal{X}}(y, y') \, d\pi_T(y, y') = W_1(\mu, \hat{\mu}) + W_1(\nu, \hat{\nu}) =: \Lambda.$$

Taking the infimum over all couplings yields $\Delta \leq \Lambda$. Therefore,

$$W_1(\gamma^*, \hat{\gamma}^*) \leq \Lambda + C_1 \sqrt{2L\Lambda} = \Lambda + C_1 \sqrt{\frac{2}{\beta}} \Lambda.$$

**Step 4 (Definition of $\lambda_{\gamma^*}$ and the factor $2\sqrt{2}$).** Define

$$\lambda_{\gamma^*} := \frac{4}{C_1^2} \quad \Longleftrightarrow \quad C_1 = \frac{2}{\sqrt{\lambda_{\gamma^*}}}.$$

Then

$$C_1 \sqrt{\frac{2}{\beta}} \Lambda = \frac{2}{\sqrt{\lambda_{\gamma^*}}} \sqrt{\frac{2}{\beta}} \sqrt{\Lambda} = 2\sqrt{\frac{2}{\beta \lambda_{\gamma^*}}} \sqrt{\Lambda},$$

which yields the claimed bound. Finally, $\lambda_{\gamma^*} > 0$ depends only on the squared-exponential moments of $\mu$ and $\nu$ because $C_1$ does via Lemma 3.10(ii) (Eckstein & Nutz, 2022). $\qquad\square$

## C.14. Proof of Theorem 4.5

*Proof.* Let

$$\mu := \mathcal{D}_X^S, \qquad \nu := \mathcal{D}_X^T, \qquad \hat{\mu} := \widehat{\mathcal{D}_X^S} = \frac{1}{N_S} \sum_{i=1}^{N_S} \delta_{X_i^{(S)}}, \qquad \hat{\nu} := \widehat{\mathcal{D}_X^T} = \frac{1}{N_T} \sum_{j=1}^{N_T} \delta_{X_j^{(T)}}.$$

Let $\gamma^*$ be the population entropic OT coupling between $\mu$ and $\nu$ (with $\beta > 0$), and let $\hat{\gamma}^* = (\hat{\gamma}_{ij}^*) \in \mathbb{R}_+^{N_S \times N_T}$ be the discrete optimal coupling solving $W_\beta(\hat{\mu}, \hat{\nu})$. It satisfies the marginal constraints

$$\sum_{j=1}^{N_T} \hat{\gamma}_{ij}^* = \frac{1}{N_S} \quad (\forall i), \qquad \sum_{i=1}^{N_S} \hat{\gamma}_{ij}^* = \frac{1}{N_T} \quad (\forall j).$$

Associate to $\hat{\gamma}^*$ a probability measure on $\mathbb{R}^d \times \mathbb{R}^d$:

$$\bar{\gamma}^* := \sum_{i=1}^{N_S} \sum_{j=1}^{N_T} \hat{\gamma}_{ij}^* \, \delta_{(X_i^{(S)}, X_j^{(T)})} \in \Gamma(\hat{\mu}, \hat{\nu}).$$

Throughout, Wasserstein–1 distances on $\mathbb{R}^d$ use the Euclidean norm $\| \cdot \|$, and on $\mathbb{R}^d \times \mathbb{R}^d$ we use the metric

$$\rho\big((x_S, x_T), (x_S', x_T')\big) := \|x_S - x_S'\| + \|x_T - x_T'\|.$$

**Step 1: define the population quantity we will concentrate around and the bias $\Delta$.** Define the function $f : \mathbb{R}^d \times \mathbb{R}^d \to \mathbb{R}_+$ by

$$f(x_S, x_T) := \mathbb{E}_{y_S \sim \mathcal{D}^S_{Y|X=x_S}, \, y_T \sim \mathcal{D}^T_{Y|X=x_T}} \left[ \, \|y_S - y_T\| \, \right].$$

Since $\mathcal{Y} \subset \mathbb{R}^{d'}$ is bounded with $M = \sup_{y,y' \in \mathcal{Y}} \|y - y'\|$, we have $0 \le f(x_S, x_T) \le M$ for all $(x_S, x_T)$.

Recall that

$$S_{pair}(x_S, x_T) = W_1\big(\mathcal{D}^S_{Y|X=x_S}, \, \mathcal{D}^T_{Y|X=x_T}\big), \qquad S^{\gamma^*}_{Cpt} = \mathbb{E}_{(x_S,x_T) \sim \gamma^*}\big[S_{pair}(x_S, x_T)\big].$$

Because $f(x_S, x_T)$ is the transport cost under the *independent* coupling $\mathcal{D}^S_{Y|x_S} \otimes \mathcal{D}^T_{Y|x_T}$, while $S_{pair}(x_S, x_T)$ is the infimum over all couplings, we have

$$f(x_S, x_T) - S_{pair}(x_S, x_T) \ge 0.$$

Define the (constant) bias term

$$\Delta := \mathbb{E}_{(x_S,x_T) \sim \gamma^*}\Big[f(x_S, x_T) - S_{pair}(x_S, x_T)\Big].$$

Then

$$S^{\gamma^*}_{Cpt} + \Delta = \mathbb{E}_{\gamma^*}\big[f(x_S, x_T)\big].$$

Hence it suffices to prove concentration of $\hat{S}_{Cpt}$ around $\mathbb{E}_{\gamma^*}[f]$.

**Step 2: decompose the error into a $Y$-noise term and a coupling-stability term.** By Definition 4.3,

$$\hat{S}_{Cpt} = \sum_{i=1}^{N_S} \sum_{j=1}^{N_T} \|Y_i^{(S)} - Y_j^{(T)}\| \, \hat{\gamma}^*_{ij}.$$

Add and subtract $\mathbb{E}_{\bar{\gamma}^*}[f]$:

$$|\hat{S}_{Cpt} - \mathbb{E}_{\gamma^*}[f]| \le \underbrace{\big|\hat{S}_{Cpt} - \mathbb{E}_{\bar{\gamma}^*}[f]\big|}_{=:A} + \underbrace{\big|\mathbb{E}_{\bar{\gamma}^*}[f] - \mathbb{E}_{\gamma^*}[f]\big|}_{=:B}.$$

We will bound $A$ by McDiarmid's inequality and $B$ by Lipschitzness of $f$ and stability of entropic OT.

**Step 3: concentration of $A$ via McDiarmid inequality B.2.** Condition on all covariates $\{X_i^{(S)}\}_{i=1}^{N_S}$ and $\{X_j^{(T)}\}_{j=1}^{N_T}$. Under this conditioning, $\hat{\gamma}^*$ is fixed (it depends only on the covariates), and the labels $\{Y_i^{(S)}\}$ are independent with $Y_i^{(S)} \sim \mathcal{D}^S_{Y|X=X_i^{(S)}}$, and similarly $\{Y_j^{(T)}\}$ are independent with $Y_j^{(T)} \sim \mathcal{D}^T_{Y|X=X_j^{(T)}}$.

Define the function of all labels

$$F\big(\{Y_i^{(S)}\}_{i=1}^{N_S}, \{Y_j^{(T)}\}_{j=1}^{N_T}\big) := \sum_{i=1}^{N_S} \sum_{j=1}^{N_T} \|Y_i^{(S)} - Y_j^{(T)}\| \, \hat{\gamma}^*_{ij}.$$

Then $A = \big|F - \mathbb{E}[F \mid \{X\}]\big|$.

*Bounded differences for changing one source label.* Fix an index $i$ and replace $Y_i^{(S)}$ by another value $Y_i^{(S)'}$ in $\mathcal{Y}$, keeping all other labels the same. Then, by the triangle inequality,

$$\left|F(\dots, Y_i^{(S)}, \dots) - F(\dots, Y_i^{(S)'}, \dots)\right|$$

$$= \left|\sum_{j=1}^{N_T} \Big(\|Y_i^{(S)} - Y_j^{(T)}\| - \|Y_i^{(S)'} - Y_j^{(T)}\|\Big)\hat{\gamma}^*_{ij}\right|$$

$$\le \sum_{j=1}^{N_T} \Big|\|Y_i^{(S)} - Y_j^{(T)}\| - \|Y_i^{(S)'} - Y_j^{(T)}\|\Big|\hat{\gamma}^*_{ij}$$

$$\le \sum_{j=1}^{N_T} \|Y_i^{(S)} - Y_i^{(S)'}\| \, \hat{\gamma}^*_{ij} \; \le \; M \sum_{j=1}^{N_T} \hat{\gamma}^*_{ij} \; = \; \frac{M}{N_S}.$$

*Bounded differences for changing one target label.* Similarly, replacing one $Y_j^{(T)}$ by $Y_j^{(T)'}$ changes $F$ by at most

$$\frac{M}{N_T}.$$

*Apply McDiarmid.* By McDiarmid nequality B.2, for any $\varepsilon_1 > 0$,

$$\mathbb{P}\Big(|F - \mathbb{E}[F \mid \{X\}]| > \varepsilon_1 \;\Big|\; \{X\}\Big) \leq 2\exp\Big(-\frac{2\varepsilon_1^2}{\sum_{i=1}^{N_S}(M/N_S)^2 + \sum_{j=1}^{N_T}(M/N_T)^2}\Big)$$

$$= 2\exp\Big(-\frac{2\varepsilon_1^2}{M^2\big(\frac{1}{N_S} + \frac{1}{N_T}\big)}\Big)$$

$$= 2\exp\Big(-\frac{2N_S N_T\, \varepsilon_1^2}{(N_S + N_T)\, M^2}\Big).$$

Taking expectation over $\{X\}$ gives the unconditional bound

$$\mathbb{P}(A > \varepsilon_1) \leq 2\exp\Big(-\frac{2N_S N_T\, \varepsilon_1^2}{(N_S + N_T)\, M^2}\Big).$$

*Compute the conditional mean* $\mathbb{E}[F \mid \{X\}]$ *and identify* $\mathbb{E}_{\bar{\gamma}^*}[f]$. Since $Y_i^{(S)}$ and $Y_j^{(T)}$ are independent given $\{X\}$, we have

$$\mathbb{E}\big[\|Y_i^{(S)} - Y_j^{(T)}\| \mid \{X\}\big] = f\big(X_i^{(S)}, X_j^{(T)}\big),$$

hence

$$\mathbb{E}[F \mid \{X\}] = \sum_{i=1}^{N_S}\sum_{j=1}^{N_T} f\big(X_i^{(S)}, X_j^{(T)}\big)\, \hat{\gamma}_{ij}^* = \mathbb{E}_{(x_S, x_T)\sim\bar{\gamma}^*}\big[f(x_S, x_T)\big] = \mathbb{E}_{\bar{\gamma}^*}[f].$$

Therefore $A = |F - \mathbb{E}[F \mid \{X\}]|$ is exactly the deviation term we bounded.

### Step 4: bound $B$ using Lipschitzness of $f$ and stability of entropic OT.

**Step 4.1: show $f$ is $(ML_{Y|X})$-Lipschitz on $\mathbb{R}^d \times \mathbb{R}^d$.** We prove Lipschitzness in the first coordinate; the second is symmetric. Fix $x_T$ and let $x_S, x_S' \in \mathbb{R}^d$. Write $\pi_{x_S}^S := \mathcal{D}_{Y|X=x_S}^S$ and $\pi_{x_T}^T := \mathcal{D}_{Y|X=x_T}^T$. Then

$$f(x_S, x_T) - f(x_S', x_T)$$

$$= \int_{\mathcal{Y}}\int_{\mathcal{Y}} \|y_S - y_T\|\, \big(\mathrm{d}\pi_{x_S}^S(y_S) - \mathrm{d}\pi_{x_S'}^S(y_S)\big)\, \mathrm{d}\pi_{x_T}^T(y_T).$$

Taking absolute values and using Fubini,

$$\big|f(x_S, x_T) - f(x_S', x_T)\big| \leq \int_{\mathcal{Y}}\Big|\int_{\mathcal{Y}} \|y_S - y_T\|\, \big(\mathrm{d}\pi_{x_S}^S(y_S) - \mathrm{d}\pi_{x_S'}^S(y_S)\big)\Big|\, \mathrm{d}\pi_{x_T}^T(y_T).$$

For each fixed $y_T$, define

$$g_{y_T}(y_S) := \frac{2}{M}\|y_S - y_T\| - 1.$$

Since $\|y_S - y_T\| \in [0, M]$ for $y_S, y_T \in \mathcal{Y}$, we have $g_{y_T} \in [-1, 1]$ and thus $\|g_{y_T}\|_\infty \leq 1$. Also note that

$$\|y_S - y_T\| = \frac{M}{2}\big(g_{y_T}(y_S) + 1\big),$$

and $\int 1\, (\mathrm{d}\pi_{x_S}^S - \mathrm{d}\pi_{x_S'}^S) = 0$, hence

$$\Big|\int_{\mathcal{Y}} \|y_S - y_T\|\, \big(\mathrm{d}\pi_{x_S}^S - \mathrm{d}\pi_{x_S'}^S\big)\Big| = \frac{M}{2}\Big|\int_{\mathcal{Y}} g_{y_T}(y_S)\, \big(\mathrm{d}\pi_{x_S}^S - \mathrm{d}\pi_{x_S'}^S\big)\Big|$$

$$\leq M\, d_{\mathrm{TV}}(\pi_{x_S}^S, \pi_{x_S'}^S),$$

where we used the dual representation

$$d_{\mathrm{TV}}(P,Q) = \frac{1}{2} \sup_{\|g\|_\infty \le 1} \left| \int g \, \mathrm{d}(P-Q) \right|.$$

Plugging this into the previous bound and using that $\pi_{x_T}^T$ has total mass 1 gives

$$\left| f(x_S, x_T) - f(x'_S, x_T) \right| \le M \, d_{\mathrm{TV}}\big(\mathcal{D}^S_{Y|X=x_S}, \mathcal{D}^S_{Y|X=x'_S}\big) \le M L_{Y|X} \|x_S - x'_S\|.$$

Similarly,

$$\left| f(x_S, x_T) - f(x_S, x'_T) \right| \le M L_{Y|X} \|x_T - x'_T\|.$$

Combining both, for all $(x_S, x_T), (x'_S, x'_T)$,

$$\left| f(x_S, x_T) - f(x'_S, x'_T) \right| \le M L_{Y|X} \big( \|x_S - x'_S\| + \|x_T - x'_T\| \big) = M L_{Y|X} \, \rho\big((x_S, x_T), (x'_S, x'_T)\big).$$

Thus $f$ is $(M L_{Y|X})$-Lipschitz on $(\mathbb{R}^d \times \mathbb{R}^d, \rho)$.

**Step 4.2: convert $B$ to a Wasserstein distance on couplings.** By the Kantorovich–Rubinstein duality for $W_1$ on $(\mathbb{R}^d \times \mathbb{R}^d, \rho)$, for any $K$-Lipschitz function $\varphi$,

$$\left| \mathbb{E}_\pi[\varphi] - \mathbb{E}_{\pi'}[\varphi] \right| \le K \, W_1(\pi, \pi').$$

Apply this with $\varphi = f$ and $K = M L_{Y|X}$, $\pi = \bar{\gamma}^*$, $\pi' = \gamma^*$:

$$B = \left| \mathbb{E}_{\bar{\gamma}^*}[f] - \mathbb{E}_{\gamma^*}[f] \right| \le M L_{Y|X} \, W_1(\gamma^*, \bar{\gamma}^*).$$

**Step 4.3: apply entropic OT stability and then concentration of marginals.** By Lemma 4.4,

$$W_1(\gamma^*, \bar{\gamma}^*) \le \Lambda + 2 \sqrt{\frac{2}{\beta \lambda_{\gamma^*}}} \sqrt{\Lambda}, \qquad \Lambda := W_1(\mu, \hat{\mu}) + W_1(\nu, \hat{\nu}).$$

Therefore,

$$B \le M L_{Y|X} \left( \Lambda + 2 \sqrt{\frac{2}{\beta \lambda_{\gamma^*}}} \sqrt{\Lambda} \right).$$

Fix $\varepsilon_2 > 0$. If $\Lambda \le 2\varepsilon_2$, then

$$\sqrt{\Lambda} \le \sqrt{2\varepsilon_2}, \qquad \Lambda + 2 \sqrt{\frac{2}{\beta \lambda_{\gamma^*}}} \sqrt{\Lambda} \le 2\varepsilon_2 + 4 \sqrt{\frac{\varepsilon_2}{\beta \lambda_{\gamma^*}}},$$

and hence

$$B \le 2 M L_{Y|X} \varepsilon_2 + 4 M L_{Y|X} \sqrt{\frac{\varepsilon_2}{\beta \lambda_{\gamma^*}}}.$$

Consequently,

$$\left\{ B > 2 M L_{Y|X} \varepsilon_2 + 4 M L_{Y|X} \sqrt{\frac{\varepsilon_2}{\beta \lambda_{\gamma^*}}} \right\} \subseteq \{ \Lambda > 2\varepsilon_2 \}.$$

By the union bound,

$$\mathbb{P}(\Lambda > 2\varepsilon_2) \le \mathbb{P}\big( W_1(\mu, \hat{\mu}) > \varepsilon_2 \big) + \mathbb{P}\big( W_1(\nu, \hat{\nu}) > \varepsilon_2 \big).$$

Since $\mu, \nu$ have finite squared-exponential moments on $\mathbb{R}^d$, by Lemma B.3 and B.4, there exist constants $\lambda_S, \lambda_T > 0$ depending only on these moments such that for any $\varepsilon_2 > 0$ there exists $N$ with, whenever $N_S, N_T > N$,

$$\mathbb{P}\big( W_1(\mu, \hat{\mu}) > \varepsilon_2 \big) \le \exp\Big( -\frac{\lambda_S N_S \varepsilon_2^2}{2} \Big), \qquad \mathbb{P}\big( W_1(\nu, \hat{\nu}) > \varepsilon_2 \big) \le \exp\Big( -\frac{\lambda_T N_T \varepsilon_2^2}{2} \Big).$$

Thus we conclude

$$\mathbb{P}\Big( B > 2 M L_{Y|X} \varepsilon_2 + 4 M L_{Y|X} \sqrt{\frac{\varepsilon_2}{\beta \lambda_{\gamma^*}}} \Big) \le \exp\Big( -\frac{\lambda_S N_S \varepsilon_2^2}{2} \Big) + \exp\Big( -\frac{\lambda_T N_T \varepsilon_2^2}{2} \Big).$$

**Step 5: combine the two parts.** For any $\varepsilon_1, \varepsilon_2 > 0$, by the union bound,

$$\mathbb{P}\Big(|\hat{S}_{Cpt} - \mathbb{E}_{\gamma^*}[f]| > \varepsilon_1 + 2ML_{Y|X}\varepsilon_2 + 4ML_{Y|X}\sqrt{\tfrac{\varepsilon_2}{\beta\lambda_{\gamma^*}}}\Big) \leq \mathbb{P}(A > \varepsilon_1)$$

$$+ \mathbb{P}\Big(B > 2ML_{Y|X}\varepsilon_2 + 4ML_{Y|X}\sqrt{\tfrac{\varepsilon_2}{\beta\lambda_{\gamma^*}}}\Big)$$

$$\leq 2\exp\Big(-\frac{2N_S N_T \varepsilon_1^2}{(N_S + N_T)M^2}\Big)$$

$$+ \exp\Big(-\frac{\lambda_S N_S \varepsilon_2^2}{2}\Big) + \exp\Big(-\frac{\lambda_T N_T \varepsilon_2^2}{2}\Big).$$

Recalling $\mathbb{E}_{\gamma^*}[f] = S_{Cpt}^{\gamma^*} + \Delta$, the left-hand side is exactly $\mathbb{P}(|\hat{S}_{Cpt} - S_{Cpt}^{\gamma^*} - \Delta| > \cdots)$.

**Step 6: parameter tying and the explicit $\Phi$.** Introduce a single auxiliary parameter $\varepsilon_0 > 0$ and set

$$\varepsilon_1 := M\varepsilon_0, \qquad \varepsilon_2 := \lambda_S^{-1/4}\lambda_T^{-1/4}\,\varepsilon_0.$$

Then the three exponential terms become

$$2\exp\Big(-\frac{2N_S N_T \varepsilon_1^2}{(N_S + N_T)M^2}\Big) = 2\exp\Big(-\frac{2N_S N_T \varepsilon_0^2}{N_S + N_T}\Big),$$

$$\exp\Big(-\frac{\lambda_S N_S \varepsilon_2^2}{2}\Big) = \exp\Big(-\frac{\lambda_S^{1/2} N_S \varepsilon_0^2}{2\lambda_T^{1/2}}\Big),$$

$$\exp\Big(-\frac{\lambda_T N_T \varepsilon_2^2}{2}\Big) = \exp\Big(-\frac{\lambda_T^{1/2} N_T \varepsilon_0^2}{2\lambda_S^{1/2}}\Big).$$

The deviation level on the left-hand side becomes

$$\varepsilon_1 + 2ML_{Y|X}\varepsilon_2 + 4ML_{Y|X}\sqrt{\tfrac{\varepsilon_2}{\beta\lambda_{\gamma^*}}} = M\varepsilon_0 + 2ML_{Y|X}\lambda_S^{-1/4}\lambda_T^{-1/4}\varepsilon_0 + 4ML_{Y|X}\beta^{-1/2}\lambda_{\gamma^*}^{-1/2}\lambda_S^{-1/8}\lambda_T^{-1/8}\varepsilon_0^{1/2}.$$

Define the constants

$$c_2 := 1 + 2L_{Y|X}\lambda_S^{-1/4}\lambda_T^{-1/4}, \qquad c_1 := 16L_{Y|X}^2\beta^{-1}\lambda_{\gamma^*}^{-1}\lambda_S^{-1/4}\lambda_T^{-1/4}.$$

Then the deviation level can be written compactly as

$$M\Big(c_2\varepsilon_0 + \sqrt{c_1\varepsilon_0}\Big).$$

Now choose $\varepsilon_0$ so that

$$M\Big(c_2\varepsilon_0 + \sqrt{c_1\varepsilon_0}\Big) = \varepsilon.$$

Let $u := \sqrt{\varepsilon_0}$. The equation becomes $c_2 u^2 + \sqrt{c_1}\,u = \varepsilon/M$, whose positive solution is

$$u = \frac{\sqrt{c_1 + 4c_2(\varepsilon/M)} - \sqrt{c_1}}{2c_2}, \qquad \varepsilon_0 = u^2.$$

It is convenient to express $\varepsilon_0^2$ in the form $\varepsilon_0^2 = \frac{\Phi\varepsilon^2}{2M^2}$. Let

$$a := \frac{c_1 M}{c_2\varepsilon}\quad (>0), \qquad \Lambda(a) := a^2 + 4a + 2 - (a+2)\sqrt{a^2 + 4a}, \qquad \Phi := \frac{\Lambda(a)}{c_2^2}.$$

A direct algebraic simplification (expanding $(a + 2 - \sqrt{a^2 + 4a})^2$) shows

$$\varepsilon_0^2 = \frac{\Phi\varepsilon^2}{2M^2}.$$

Substituting this $\varepsilon_0^2$ into the exponential terms yields

$$2\exp\left(-\frac{2N_S N_T \varepsilon_0^2}{N_S + N_T}\right) = 2\exp\left(-\frac{N_S N_T \, \Phi \, \varepsilon^2}{(N_S + N_T) \, M^2}\right),$$

$$\exp\left(-\frac{\lambda_S^{1/2} N_S \varepsilon_0^2}{2\lambda_T^{1/2}}\right) = \exp\left(-\frac{\lambda_S^{1/2} N_S \, \Phi \, \varepsilon^2}{4\lambda_T^{1/2} M^2}\right),$$

$$\exp\left(-\frac{\lambda_T^{1/2} N_T \varepsilon_0^2}{2\lambda_S^{1/2}}\right) = \exp\left(-\frac{\lambda_T^{1/2} N_T \, \Phi \, \varepsilon^2}{4\lambda_S^{1/2} M^2}\right).$$

This is exactly the claimed bound, completing the proof. $\qquad\square$

### C.15. Proof of Proposition 4.6

*Proof.* Recall the notation used in the proof of Theorem 4.5: for $(x_S, x_T) \in \mathcal{X} \times \mathcal{X}$ define

$$f(x_S, x_T) := \mathbb{E}_{y_S \sim \mathcal{D}^S_{Y|X=x_S}, \, y_T \sim \mathcal{D}^T_{Y|X=x_T}}\big[\rho_{\mathcal{Y}}(y_S, y_T)\big], \qquad S_{pair}(x_S, x_T) := W_1\big(\mathcal{D}^S_{Y|X=x_S}, \mathcal{D}^T_{Y|X=x_T}\big),$$

and the bias is

$$\Delta = \mathbb{E}_{(x_S, x_T) \sim \gamma^*}\big[f(x_S, x_T) - S_{pair}(x_S, x_T)\big].$$

Under deterministic labeling, $\mathcal{D}^S_{Y|X=x_S} = \delta_{f_S(x_S)}$ and $\mathcal{D}^T_{Y|X=x_T} = \delta_{f_T(x_T)}$. Hence the random variables $y_S, y_T$ are almost surely equal to $f_S(x_S), f_T(x_T)$, and thus

$$f(x_S, x_T) = \mathbb{E}\big[\rho_{\mathcal{Y}}(y_S, y_T)\big] = \rho_{\mathcal{Y}}\big(f_S(x_S), f_T(x_T)\big).$$

On the other hand, the Wasserstein-1 distance between two Dirac measures is exactly the ground metric:

$$S_{pair}(x_S, x_T) = W_1\big(\delta_{f_S(x_S)}, \delta_{f_T(x_T)}\big) = \rho_{\mathcal{Y}}\big(f_S(x_S), f_T(x_T)\big).$$

Therefore $f(x_S, x_T) - S_{pair}(x_S, x_T) = 0$ for all $(x_S, x_T)$, and so

$$\Delta = \mathbb{E}_{\gamma^*}\big[f(x_S, x_T) - S_{pair}(x_S, x_T)\big] = 0.$$

This concludes the proof. $\qquad\square$

### C.16. Proof of Proposition 4.7

*Proof.* We follow the notation used in the proof of Theorem 4.5. For $(x_S, x_T) \in \mathcal{X} \times \mathcal{X}$, define

$$f(x_S, x_T) := \mathbb{E}_{y_S \sim \mathcal{D}^S_{Y|X=x_S}, \, y_T \sim \mathcal{D}^T_{Y|X=x_T}}\big[\|y_S - y_T\|\big], \qquad S_{pair}(x_S, x_T) := W_1\big(\mathcal{D}^S_{Y|X=x_S}, \mathcal{D}^T_{Y|X=x_T}\big),$$

and recall that

$$\Delta = \mathbb{E}_{(x_S, x_T) \sim \gamma^*}\big[f(x_S, x_T) - S_{pair}(x_S, x_T)\big].$$

**Lower bound $\Delta \geq 0$.** For each fixed $(x_S, x_T)$, the product measure $\mathcal{D}^S_{Y|X=x_S} \otimes \mathcal{D}^T_{Y|X=x_T}$ belongs to $\Gamma\big(\mathcal{D}^S_{Y|X=x_S}, \mathcal{D}^T_{Y|X=x_T}\big)$. Since $S_{pair}(x_S, x_T)$ is the infimum of $\mathbb{E}[\|y_S - y_T\|]$ over all such couplings,

$$S_{pair}(x_S, x_T) \leq \mathbb{E}_{\mathcal{D}^S_{Y|X=x_S} \otimes \mathcal{D}^T_{Y|X=x_T}}\big[\|y_S - y_T\|\big] = f(x_S, x_T).$$

Thus $f(x_S, x_T) - S_{pair}(x_S, x_T) \geq 0$ pointwise, hence $\Delta \geq 0$.

**An upper bound for each pair** $(x_S, x_T)$. Fix $(x_S, x_T)$ and write $P := \mathcal{D}^S_{Y|X=x_S}$ and $Q := \mathcal{D}^T_{Y|X=x_T}$. Let their means be

$$m_S(x_S) := \mathbb{E}_{y \sim P}[y], \qquad m_T(x_T) := \mathbb{E}_{y \sim Q}[y],$$

For independent draws $y_S \sim P$ and $y_T \sim Q$, the triangle inequality yields

$$\|y_S - y_T\| \le \|y_S - m_S(x_S)\| + \|m_S(x_S) - m_T(x_T)\| + \|m_T(x_T) - y_T\|.$$

Taking expectation gives

$$f(x_S, x_T) \le \mathbb{E}_{y \sim P}\big[\|y - m_S(x_S)\|\big] + \|m_S(x_S) - m_T(x_T)\| + \mathbb{E}_{y \sim Q}\big[\|y - m_T(x_T)\|\big].$$

Next we lower bound $S_{pair}(x_S, x_T) = W_1(P, Q)$ by the distance between the means. Let $\pi \in \Gamma(P, Q)$ be any coupling, and denote $m_P := \mathbb{E}_{y \sim P}[y]$, $m_Q := \mathbb{E}_{y \sim Q}[y]$. Since $\|\cdot\|$ is convex, Jensen's inequality yields

$$\mathbb{E}_{(y,y') \sim \pi}\big[\|y - y'\|\big] \ge \Big\|\mathbb{E}_{(y,y') \sim \pi}[y - y']\Big\|.$$

By the marginal constraints of $\pi$, we have $\mathbb{E}_{(y,y') \sim \pi}[y] = m_P$ and $\mathbb{E}_{(y,y') \sim \pi}[y'] = m_Q$, hence

$$\Big\|\mathbb{E}_{(y,y') \sim \pi}[y - y']\Big\| = \|m_P - m_Q\|.$$

Therefore, for every $\pi \in \Gamma(P, Q)$,

$$\mathbb{E}_{(y,y') \sim \pi}\big[\|y - y'\|\big] \ge \|m_P - m_Q\|.$$

Taking the infimum over $\pi \in \Gamma(P, Q)$ gives

$$W_1(P, Q) = \inf_{\pi \in \Gamma(P,Q)} \mathbb{E}_{(y,y') \sim \pi}\big[\|y - y'\|\big] \ge \|m_P - m_Q\|.$$

Combining the two displays and canceling the middle term yields

$$f(x_S, x_T) - S_{pair}(x_S, x_T) \le \mathbb{E}_{y \sim P}\big[\|y - m_S(x_S)\|\big] + \mathbb{E}_{y \sim Q}\big[\|y - m_T(x_T)\|\big].$$

**Integrate over $\gamma^*$ and relate to irreducible errors.** Taking expectation over $(x_S, x_T) \sim \gamma^*$ and using that the marginals of $\gamma^*$ are $\mathcal{D}^S_X$ and $\mathcal{D}^T_X$,

$$\Delta \le \mathbb{E}_{x \sim \mathcal{D}^S_X}\Big[\mathbb{E}\big[\|Y^{(S)} - m_S(x)\| \mid X^{(S)} = x\big]\Big] + \mathbb{E}_{x \sim \mathcal{D}^T_X}\Big[\mathbb{E}\big[\|Y^{(T)} - m_T(x)\| \mid X^{(T)} = x\big]\Big].$$

For any random variable $Z \ge 0$, $\mathbb{E}[Z] \le \sqrt{\mathbb{E}[Z^2]}$, hence

$$\mathbb{E}\big[\|Y^{(S)} - m_S(x)\| \mid X^{(S)} = x\big] \le \sqrt{\mathbb{E}\big[\|Y^{(S)} - m_S(x)\|^2 \mid X^{(S)} = x\big]},$$

and similarly for the target term. Thus

$$\Delta \le \mathbb{E}_{x \sim \mathcal{D}^S_X}\Big[\sqrt{\mathbb{E}\big[\|Y^{(S)} - m_S(x)\|^2 \mid X^{(S)} = x\big]}\Big] + \mathbb{E}_{x \sim \mathcal{D}^T_X}\Big[\sqrt{\mathbb{E}\big[\|Y^{(T)} - m_T(x)\|^2 \mid X^{(T)} = x\big]}\Big].$$

Since $\sqrt{\cdot}$ is concave on $\mathbb{R}_+$, Jensen's inequality gives

$$\mathbb{E}_{x \sim \mathcal{D}^S_X}\Big[\sqrt{\mathbb{E}\big[\|Y^{(S)} - m_S(x)\|^2 \mid X^{(S)} = x\big]}\Big] \le \sqrt{\mathbb{E}_{(x,y) \sim \mathcal{D}^S_{XY}}\big[\|y - m_S(x)\|^2\big]},$$

$$\mathbb{E}_{x \sim \mathcal{D}^T_X}\Big[\sqrt{\mathbb{E}\big[\|Y^{(T)} - m_T(x)\|^2 \mid X^{(T)} = x\big]}\Big] \le \sqrt{\mathbb{E}_{(x,y) \sim \mathcal{D}^T_{XY}}\big[\|y - m_T(x)\|^2\big]}.$$

Finally, under squared loss on $\mathbb{R}^{d'}$, the minimizer of $\mathbb{E}[\|Y - g(x)\|^2 \mid X = x]$ is $g(x) = \mathbb{E}[Y \mid X = x]$. Therefore the functions $m_S(x) = \mathbb{E}[Y^{(S)} \mid X^{(S)} = x]$ and $m_T(x) = \mathbb{E}[Y^{(T)} \mid X^{(T)} = x]$ achieve the infima of $I(\mathcal{D}^S_{XY})$ and $I(\mathcal{D}^T_{XY})$, i.e.,

$$\mathbb{E}_{(x,y) \sim \mathcal{D}^S_{XY}}\big[\|y - m_S(x)\|^2\big] = I(\mathcal{D}^S_{XY}), \qquad \mathbb{E}_{(x,y) \sim \mathcal{D}^T_{XY}}\big[\|y - m_T(x)\|^2\big] = I(\mathcal{D}^T_{XY}).$$

Putting the above bounds together yields

$$\Delta \le \sqrt{I(\mathcal{D}^S_{XY})} + \sqrt{I(\mathcal{D}^T_{XY})},$$

which completes the proof. $\qquad\square$

# D. Analysis of Lipschitz Constants: Results and Proofs

## D.1. Lipschitz constant of the sigmoid function

**Proposition D.1** (Lipschitz constant of the sigmoid). *Let $\sigma : \mathbb{R} \to \mathbb{R}$ be the sigmoid function*

$$\sigma(x) \;=\; \frac{1}{1 + e^{-x}}.$$

*Then $\sigma$ is globally $\frac{1}{4}$-Lipschitz on $\mathbb{R}$.*

*Proof.* We first compute the derivative:

$$\sigma'(x) = \frac{d}{dx}\left(1 + e^{-x}\right)^{-1} = \frac{e^{-x}}{\left(1 + e^{-x}\right)^2}.$$

Using $\sigma(x) = \frac{1}{1+e^{-x}}$ and $1 - \sigma(x) = \frac{e^{-x}}{1+e^{-x}}$, we can rewrite

$$\sigma'(x) = \sigma(x)\big(1 - \sigma(x)\big).$$

Let $u = \sigma(x) \in (0, 1)$. Then $\sigma'(x) = u(1 - u)$, and for all $u \in [0, 1]$,

$$u(1 - u) = u - u^2 \;\leq\; \max_{t \in [0,1]} (t - t^2) = \frac{1}{4},$$

where the maximum is attained at $t = \frac{1}{2}$. Hence $0 < \sigma'(x) \leq \frac{1}{4}$ for all $x$.

Now fix any $x, y \in \mathbb{R}$. Since $\sigma$ is differentiable on $\mathbb{R}$ (hence continuous), by the mean value theorem, there exists $c$ between $x$ and $y$ such that

$$\sigma(x) - \sigma(y) = \sigma'(c)\,(x - y).$$

Taking absolute values and using $\sigma'(c) \leq \frac{1}{4}$ yields

$$|\sigma(x) - \sigma(y)| = |\sigma'(c)|\,|x - y| \leq \frac{1}{4}|x - y|,$$

so $\sigma$ is $\frac{1}{4}$-Lipschitz. $\qquad\square$

## D.2. Lipschitz constant of the logistic regression

**Proposition D.2** (Lipschitz constant of logistic regression under $\ell_p/\ell_q$). *Fix $w \in \mathbb{R}^d$ and $b \in \mathbb{R}$. Define*

$$h_{w,b}(x) \;=\; \sigma\big(w^\top x + b\big), \qquad \sigma(t) = \frac{1}{1 + e^{-t}}.$$

*Let $p \in [1, \infty]$ and let $q \in [1, \infty]$ be its Hölder conjugate, i.e., $\frac{1}{p} + \frac{1}{q} = 1$. Then $h_{w,b}$ is globally $\frac{\|w\|_q}{4}$-Lipschitz with respect to $\|\cdot\|_p$, namely for all $x, x' \in \mathbb{R}^d$,*

$$\big|h_{w,b}(x) - h_{w,b}(x')\big| \;\leq\; \frac{\|w\|_q}{4}\,\|x - x'\|_p.$$

*In particular, under $\|\cdot\|_2$, the Lipschitz constant equals $\|w\|_2/4$.*

*Proof.* For any $x, x' \in \mathbb{R}^d$, apply the mean value theorem to $\sigma$ to obtain some $\xi$ between $w^\top x + b$ and $w^\top x' + b$ such that

$$h_{w,b}(x) - h_{w,b}(x') = \sigma'(\xi)\,w^\top (x - x').$$

Thus

$$\big|h_{w,b}(x) - h_{w,b}(x')\big| \leq |\sigma'(\xi)|\,\big|w^\top (x - x')\big|.$$

By Proposition D.1, we have $|\sigma'(\xi)| \leq \frac{1}{4}$. By Hölder's inequality with conjugate exponents $(p, q)$,

$$\left| w^\top (x - x') \right| \leq \|w\|_q \, \|x - x'\|_p.$$

Combining the two inequalities yields

$$\left| h_{w,b}(x) - h_{w,b}(x') \right| \leq \frac{1}{4} \, \|w\|_q \, \|x - x'\|_p,$$

which proves the claim. $\qquad\square$

### D.3. Lipschitz constant of the linear multi-class classifier

**Proposition D.3** (Lipschitz constant of the linear classifier under $\|\cdot\|_2$)**.** *Let $W \in \mathbb{R}^{N \times d}$ and $b \in \mathbb{R}^N$. Define*

$$f(x) = \mathrm{softmax}(Wx + b) \in \Delta^{N-1}, \qquad x \in \mathbb{R}^d.$$

*Here $\Delta^{N-1} := \{p \in \mathbb{R}^N : p_i \geq 0, \, \forall i, \, \sum_{i=1}^N p_i = 1\}$ denotes the probability simplex. Let $w_i^\top$ denote the $i$-th row of $W$ and define*

$$D(W) := \max_{i,j \in [N]} \|w_i - w_j\|_2.$$

*Then $f$ is Lipschitz from $(\mathbb{R}^d, \|\cdot\|_2)$ to $(\mathbb{R}^N, \|\cdot\|_2)$ and its optimal Lipschitz constant equals*

$$\sup_{x \in \mathbb{R}^d} \|\nabla f(x)\|_2 = \frac{1}{2\sqrt{2}} \, D(W),$$

*where $\|\cdot\|_2$ denotes the matrix operator norm induced by the vector $\|\cdot\|_2$ norm.*

*Proof.* For $z \in \mathbb{R}^N$, write $p = \mathrm{softmax}(z) \in \Delta^{N-1}$. The Jacobian of softmax is

$$J(p) = \nabla_z \mathrm{softmax}(z) = \mathrm{Diag}(p) - pp^\top,$$

where $\mathrm{Diag}(p)$ is the diagonal matrix with diagonal entries $p_1, \ldots, p_N$. By the chain rule, for $p = \mathrm{softmax}(Wx + b)$,

$$\nabla f(x) = J(p) \, W.$$

Therefore,

$$\sup_{x \in \mathbb{R}^d} \|\nabla f(x)\|_2 = \sup_{p \in \Delta^{N-1}} \|J(p)W\|_2. \qquad (16)$$

Fix $p \in \Delta^{N-1}$ and a unit vector $u \in \mathbb{R}^d$ with $\|u\|_2 = 1$. Let

$$v := Wu \in \mathbb{R}^N, \qquad \mu := p^\top v.$$

and $\mu$ represents the mean of $v$ under the probability distribution $p$. We have

$$J(p)v = \left( \mathrm{Diag}(p) - pp^\top \right)v = \mathrm{Diag}(p)\left(v - \mu\mathbf{1}\right),$$

hence

$$\|J(p)Wu\|_2^2 = \|J(p)v\|_2^2 = \sum_{i=1}^N p_i^2 \, (v_i - \mu)^2. \qquad (17)$$

Now fix $v \in \mathbb{R}^N$ and consider the quantity

$$\Phi(v) := \sup_{p \in \Delta^{N-1}} \sum_{i=1}^N p_i^2 \, (v_i - p^\top v)^2.$$

Indeed, for any $p \in \Delta^{N-1}$,

$$\sum_{i=1}^{N} p_i^2 (v_i - \mu)^2 \leq \left( \max_i p_i \right) \sum_{i=1}^{N} p_i (v_i - \mu)^2 = \left( \max_i p_i \right) \mathrm{Var}_p(v),$$

where $\mathrm{Var}_p(v) := \sum_i p_i (v_i - \mu)^2$, which represents the variance of $v$ under the probability distribution $p$. If $p$ is supported on two values $m := \min_i v_i$ and $M := \max_i v_i$ with masses $\alpha$ and $1 - \alpha$, let $r := M - m$, then

$$\sum_{i=1}^{N} p_i^2 (v_i - \mu)^2 = 2\alpha^2 (1 - \alpha)^2 r^2 \leq \frac{r^2}{8},$$

with equality at $\alpha = \frac{1}{2}$. Since any mass placed at intermediate values of $v$ cannot increase the range-based extremum, we obtain $\Phi(v) = r^2/8$, and the optimal $p$ is supported on indices attaining $M$ and $m$.

Combining (17), for any unit $u$,

$$\sup_{p \in \Delta^{N-1}} \|J(p)Wu\|_2 = \frac{1}{2\sqrt{2}} \left( \max_i (Wu)_i - \min_j (Wu)_j \right). \tag{18}$$

Finally, maximize the range over $u$. Writing $(Wu)_i = w_i^\top u$, we have

$$\max_i (Wu)_i - \min_j (Wu)_j = \max_{i,j} (w_i - w_j)^\top u.$$

Thus,

$$\sup_{\|u\|_2 = 1} \left( \max_i (Wu)_i - \min_j (Wu)_j \right) = \max_{i,j} \sup_{\|u\|_2 = 1} (w_i - w_j)^\top u = \max_{i,j} \|w_i - w_j\|_2 = D(W).$$

Plugging this into (18) and then into (16) yields

$$
\begin{aligned}
&\sup_{x \in \mathbb{R}^d} \|\nabla f(x)\|_2 \\
&= \sup_{p \in \Delta^{N-1}} \|J(p)W\|_2 \\
&= \sup_{p \in \Delta^{N-1}} \sup_{\|u\|_2 = 1} \|J(p)Wu\|_2 \\
&= \sup_{\|u\|_2 = 1} \sup_{p \in \Delta^{N-1}} \|J(p)Wu\|_2 \\
&= \frac{1}{2\sqrt{2}} D(W)
\end{aligned}
$$

This completes the proof. $\qquad\qquad\qquad\qquad\qquad\qquad\qquad\qquad\qquad\qquad\qquad\qquad\qquad\qquad\quad\square$

**Proposition D.4** (Lipschitz constant of the linear classifier under $\| \cdot \|_2 \to \| \cdot \|_1$). *Let $W \in \mathbb{R}^{N \times d}$ and $b \in \mathbb{R}^N$. Define*

$$f(x) = \mathrm{softmax}(Wx + b) \in \Delta^{N-1}, \qquad x \in \mathbb{R}^d.$$

*Here $\Delta^{N-1} := \{p \in \mathbb{R}^N : p_i \geq 0, \forall i, \sum_{i=1}^N p_i = 1\}$ denotes the probability simplex. Let $w_i^\top$ denote the $i$-th row of $W$ and define*

$$D(W) := \max_{i,j \in [N]} \|w_i - w_j\|_2.$$

*Then $f$ is Lipschitz from $(\mathbb{R}^d, \| \cdot \|_2)$ to $(\mathbb{R}^N, \| \cdot \|_1)$, and its optimal Lipschitz constant equals*

$$\sup_{x \in \mathbb{R}^d} \|\nabla f(x)\|_{2 \to 1} = \frac{1}{2} D(W),$$

*where $\| \cdot \|_{2 \to 1}$ denotes the operator norm induced by $\| \cdot \|_2$ on the input and $\| \cdot \|_1$ on the output.*

*Proof.* For $z \in \mathbb{R}^N$, write $p = \mathrm{softmax}(z) \in \Delta^{N-1}$ and recall

$$J(p) = \nabla_z \mathrm{softmax}(z) = \mathrm{Diag}(p) - pp^\top.$$

By the chain rule, for $p = \mathrm{softmax}(Wx + b)$ we have

$$\nabla f(x) = J(p)\, W.$$

Hence

$$\sup_{x \in \mathbb{R}^d} \|\nabla f(x)\|_{2 \to 1} = \sup_{p \in \Delta^{N-1}} \|J(p)W\|_{2 \to 1}. \tag{19}$$

Fix $p \in \Delta^{N-1}$ and $v \in \mathbb{R}^N$. Let $\mu := p^\top v$. Using

$$J(p)v = (\mathrm{Diag}(p) - pp^\top)v = \mathrm{Diag}(p)(v - \mu\mathbf{1}),$$

we obtain

$$\|J(p)v\|_1 = \sum_{i=1}^N p_i\, |v_i - \mu|. \tag{20}$$

The right-hand side is the mean absolute deviation of $v$ under the probability distribution $p$.

Fix $v \in \mathbb{R}^N$ and define $m := \min_i v_i$, $M := \max_i v_i$, and $r := M - m$. We claim that

$$\sup_{p \in \Delta^{N-1}} \sum_{i=1}^N p_i\, |v_i - p^\top v| = \frac{r}{2}. \tag{21}$$

*Upper bound.* Let $\mu = p^\top v$. By Cauchy–Schwarz,

$$\sum_{i=1}^N p_i |v_i - \mu| \leq \Big(\sum_{i=1}^N p_i\Big)^{1/2} \Big(\sum_{i=1}^N p_i(v_i - \mu)^2\Big)^{1/2} = \sqrt{\mathrm{Var}_p(v)}.$$

By Popoviciu's inequality for bounded random variables, $\mathrm{Var}_p(v) \leq r^2/4$, hence

$$\sum_{i=1}^N p_i|v_i - \mu| \leq \frac{r}{2}.$$

*Lower bound (attainability).* Let $i_{\max} \in \arg\max_i v_i$ and $i_{\min} \in \arg\min_i v_i$, and take

$$p = \frac{1}{2}(e_{i_{\max}} + e_{i_{\min}}).$$

Here $e_i \in \mathbb{R}^N$ denotes the $i$-th standard basis vector, i.e., $(e_i)_j = \mathbf{1}\{j = i\}$. Then $\mu = (M + m)/2$, and

$$\sum_{i=1}^N p_i|v_i - \mu| = \frac{1}{2}\Big|M - \frac{M+m}{2}\Big| + \frac{1}{2}\Big|m - \frac{M+m}{2}\Big| = \frac{r}{2}.$$

This proves (21).

Combining (20) and (21), for any $u \in \mathbb{R}^d$ with $\|u\|_2 = 1$,

$$\sup_{p \in \Delta^{N-1}} \|J(p)Wu\|_1 = \frac{1}{2}\Big(\max_i(Wu)_i - \min_j(Wu)_j\Big). \tag{22}$$

Writing $(Wu)_i = w_i^\top u$, we have

$$\max_i(Wu)_i - \min_j(Wu)_j = \max_{i,j}(w_i - w_j)^\top u.$$

Therefore,

$$\sup_{\|u\|_2=1} \Big( \max_i(Wu)_i - \min_j(Wu)_j \Big) = \max_{i,j} \sup_{\|u\|_2=1} (w_i - w_j)^\top u = \max_{i,j} \|w_i - w_j\|_2 = D(W).$$

From (19) and (22),

$$
\begin{aligned}
&\sup_{x\in\mathbb{R}^d} \|\nabla f(x)\|_{2\to 1} \\
&= \sup_{p\in\Delta^{N-1}} \sup_{\|u\|_2=1} \|J(p)Wu\|_1 \\
&= \sup_{\|u\|_2=1} \sup_{p\in\Delta^{N-1}} \|J(p)Wu\|_1 \\
&= \frac{1}{2} \sup_{\|u\|_2=1} \Big( \max_i(Wu)_i - \min_j(Wu)_j \Big) \\
&= \frac{1}{2} D(W).
\end{aligned}
$$

This completes the proof. $\qquad\qquad\square$

## D.4. Lipschitz constant of MLPs

Based on the previous theory on the Lipschitz constant of MLPs by Fazlyab et al. (2020), we obtain the following direct result:

**Proposition D.5** (SDP-certified Lipschitz constant for MLPs). *Consider the $\ell$-hidden-layer MLP*

$$x_0 = x \in \mathbb{R}^{n_0}, \qquad x_k = \phi(W_{k-1}x_{k-1} + b_{k-1}) \in \mathbb{R}^{n_k} \ (k=1,\ldots,\ell),$$

$$f(x) = W_\ell x_\ell + b_\ell \in \mathbb{R}^m,$$

*where $\phi(z) = [\varphi(z_1),\ldots,\varphi(z_{n_k})]^\top$ acts componentwise and the scalar activation $\varphi$ is slope-restricted on $[0,\beta]$ (i.e., $0 \le \frac{\varphi(u)-\varphi(v)}{u-v} \le \beta$ for all $u \ne v$).*

*Let $N := n_0 + n_1 + \cdots + n_\ell$ and $n := n_1 + \cdots + n_\ell$. Define the matrices $A \in \mathbb{R}^{n\times N}$ and $B \in \mathbb{R}^{n\times N}$ by*

$$
A = \begin{bmatrix} W_0 & 0 & \cdots & 0 & 0 \\ 0 & W_1 & \cdots & 0 & 0 \\ \vdots & \vdots & \ddots & \vdots & \vdots \\ 0 & 0 & \cdots & W_{\ell-1} & 0 \end{bmatrix}, \qquad
B = \begin{bmatrix} 0 & I_{n_1} & 0 & \cdots & 0 \\ 0 & 0 & I_{n_2} & \cdots & 0 \\ \vdots & \vdots & \vdots & \ddots & \vdots \\ 0 & 0 & 0 & \cdots & I_{n_\ell} \end{bmatrix}.
$$

*Let the decision variable $T_n \in \mathbb{S}^n$ be block-diagonal across layers:*

$$T_n = \mathrm{blkdiag}(T_{n_1},\ldots,T_{n_\ell}), \qquad T_{n_k} = \mathrm{diag}(t_{k,1},\ldots,t_{k,n_k}), \quad t_{k,i} \ge 0.$$

*Consider the SDP (with variable $H \ge 0$):*

$$
\begin{aligned}
\min_{\{t_{k,i}\},\,H} \quad & H \\
\text{s.t.} \quad & M \succeq 0, \\
& M = -\Big( M_L(H) + M_A(T_n) \Big),
\end{aligned}
$$

$$
M_L(H) = \begin{bmatrix} -HI_{n_0} & 0 & \cdots & 0 \\ 0 & 0 & \cdots & 0 \\ \vdots & \vdots & \ddots & \vdots \\ 0 & 0 & \cdots & W_\ell^\top W_\ell \end{bmatrix} \in \mathbb{S}^N,
$$

$$
M_A(T_n) = \begin{bmatrix} A \\ B \end{bmatrix}^\top \begin{bmatrix} 0 & \beta T_n \\ \beta T_n & -2T_n \end{bmatrix} \begin{bmatrix} A \\ B \end{bmatrix} \in \mathbb{S}^N.
$$

*Let $H^\star$ be the optimal value and define $L_{\mathrm{SDP}} := \sqrt{H^\star}$. Then $L_{\mathrm{SDP}}$ is a global $\|\cdot\|_2$-Lipschitz constant of $f$, i.e.,*

$$\|f(x) - f(y)\|_2 \le L_{\mathrm{SDP}}\|x - y\|_2, \qquad \forall x, y \in \mathbb{R}^{n_0}.$$

