# OpenReview forum: "General Quantification of Covariate and Concept Shifts"
_ICML.cc/2026/Conference — ICML 2026 regular_

### Official Review · Reviewer_74cM · 2026-02-23

**Soundness:** 3
**Presentation:** 3
**Significance:** 3
**Originality:** 3
**Overall Recommendation:** 4
**Confidence:** 2

**Summary:**

This work reinvestigates the problem of decomposing distribution shift into covariate shift and concept shift. It aims to solve two problems left in previous works.
1. the ill-definedness of concept shift when the support over two domains is different.
2. deterministic labeling for concept shift.

This work proposes using optimal transport to describe these shifts. Specifically, the authors use entropic optimal transport to describe covariate shifts. They then define the $\gamma^{*}$-concept shift by leveraging the optimal transport coupling in the covariate shift definition. Because the coupling pairs source points with target points rather than evaluating source conditionals at target locations, the support mismatch issue goes away.

This work then provides generic learning bounds, which improves upon the bounds in previous work. On the estimation side, the authors propose a debiased estimator for the covariate shift and a coupling-based estimator for the concept shift term, both with concentration guarantees, and then pack them into the DataShifts algorithm.

**Compliance With Llm Reviewing Policy:**

Affirmed.

**Ethical Review Flag:**

Flag this paper for an ethics review.

**Key Questions For Authors:**

1. I would suggest authors compare Theorem 2.3 and Theorem 3.13 in a table to better illustrate the improvements.
2. The handling of stochastic labeling is highlighted as a key contribution, but all three experiments appear to have approximately deterministic labels. It may be more convincing to have synthetic data to demonstrate how the estimator and bound behave under controlled label noise.

**Limitations:**

This work acknowledges the Lipschitz constant issue in practice.

**Strengths And Weaknesses:**

## Strength
1. Although this work is not the first to consider covariate and concept shift simultaneously, it spots the ill-definedness of concept shift caused by mismatch in the support, which seems a novel insight to me.
2. The idea of using the OT coupling to redefine concept shift is the main novelty (at least to me). Rather than evaluating the source conditional at target locations (which fails under support mismatch), the coupling sidesteps this by pairing source and target points.
3. For the presentation, the comparison between classical results in Theorem 2.3 and results from this work, Theorem 3.13, helps to understand the improvement.

## Weakness
1. Following my previous point in (S2), while the idea of using OT coupling for concept shift is novel to me, I'm not sure if this is already common in the field, as the idea is a natural one.
2. While the authors assert the bound can help for model selection or adaptation strategy, they don't demonstrate it. Specifically, it doesn't show a case where the decomposition into covariate vs. concept shift actually informs a decision (e.g., "concept shift dominates, so use method X instead of Y").
3. The bound depends on $L_h$. For models beyond simple models, like logistic regression, computing a tight Lipschitz constant is difficult. If the constants obtained from models are loose, the looseness propagates to the bound.
4. The choice of $\beta$ is not well-characterized. The algorithm uses $\beta = 0.01$ as a default, but the authors use $\beta = 0.2$ in experiments. Since $\beta$ affects both the covariate shift value and the coupling (and therefore the concept shift value), some sensitivity analysis would strengthen the empirical claims.

---

> ### Author Rebuttal · Authors · 2026-03-31
>
> Thank you for the careful reading and positive comments. We are glad you value the novelty of our theory.
>
> ## W1&Q1
>
> Thank you for the suggestion. We summarize our baseline (Theorem 2.3) and other OT-based bounds in the table below:
>
> |$\quad\quad\quad\quad$Bound|Divergence|$\quad\quad$X shift|Y\|X shift|Remaining Term|Tight|Estimable|General|
> |:--:|:--:|:--:|:--:|:--:|:--:|:--:|:--:|
> |Ben-David et al. (2006)|$\mathcal{H}$-divergence|✓||joint-error λ||||
> |Redko et al. (2017)|OT|✓||joint-error λ||||
> |Courty et al. (2017)|OT|estimated XY shift ||joint-error λ, kMΦ|||✓|
> |Zhao et al. (2019) **(ICML oral, our baseline)**|$\mathcal{H}$-divergence|✓|✓(ill-defined)||✓|||
> |Zhang et al. (2023)|$\mathcal{H}$-divergence|✓|✓(ill-defined)||✓||✓|
> |Hamri et al. (2025) **(from Reviewer VJ2J)**|hierarchical OT|✓||joint-error λ||||
> |**Ours**|entropic OT|✓|✓||✓✓|✓|✓✓|
>
> Other OT-based works mostly study X shift, with Y|X shift hidden in the joint-error λ. Our baseline Zhao et al. (2019) made clear that such λ is loose and non-estimable, and thus proposed an explicit Y|X shift and a tighter bound. We follow this line and further show that prior Y|X shift is loose and non-estimable as well when support mismatch. To the best of our knowledge, we are the first to use OT to improve explicit Y|X shift, yielding a tighter, fully estimable, and general theory.
>
> ## W2
>
> We may not have asserted the bound can help for model selection or adaptation, but we do hope to explore the algorithmic application of this theory as future work, since this paper focuses on a general, quantifiable theory of distribution shift. It can support post hoc diagnosis of why models fail to generalize, as in our Novozymes experiment.
>
> ## W3
>
> Yes, a large $L_h$ can make the bound loose, and we can additionally show the X and Y|X shifts still stay informative. On PACS, the feature dimension is 2048, so $L_h$ is large and the bound is loose: over all points in Fig. 1(c), the Pearson correlation between the bound and test error is 56.9%. Since $L_h$ measures the model’s largest response to input change, we also replace it by $\frac14 L_h$ and $\frac18 L_h$ to reflect a more average response. The resulting scaled bounds raise the correlation to 86.7% and 91.4%. This suggests that the looseness mainly comes from $L_h$, while our X and Y|X shifts still retain much information about error.
>
> ## W4
>
> Thank you for your suggestion. For $\beta$, we test the sensitivity of our X and Y|X shift. We take the "-90%" domain of ColoredMNIST as target, use ERM, train with the best hyperparameters from DomainBed. At the last checkpoint, we sample 10,000 feature-label pairs from both source and target domains. Running DataShifts with different $\beta$ gives:
>
> |$\quad\beta$|0.001|0.002|0.005|0.01|0.02|0.05|0.1|0.2|
> |:--:|:--:|:--:|:--:|:--:|:--:|:--:|:--:|:--:|
> |$\hat S_{Cov}$|0.5074|0.5073|0.5071|0.5066|0.5048|0.5028|0.4985|0.4927|
> |$\hat S_{Cpt}$|1.4459|1.4461|1.4471|1.4489|1.4507|1.4582|1.4686|1.4903|
> |time (s)|15.27|14.13|12.75|11.64|10.66|9.12|8.05|6.75|
>
> Across magnitude changes in $\beta$, the coefficients of variation of $\hat S_{Cov}$ and $\hat S_{Cpt}$ are 0.983% and 0.998%, so both are stable for small $\beta$. Since entropic OT gets faster as $\beta$ grows, we use $\beta=0.2$ in the paper to balance speed and accuracy.
>
> ## Q2
>
> Thank you for your suggestion. We test our $\gamma^*$-Y|X shift estimator on the following stochastic-labeling synthetic data. In both source and target, covariates are drawn from a 10-dim standard normal distributions. At each $x$, scalar labels are sampled from $\mathcal{N}(\|x\|_2,\sigma^2)$ in the source and $\mathcal{N}(\|x\|_2+1.0,\sigma^2)$ in the target. Thus, the true Y|X shift is 1.0, and standard deviation $\sigma$ controls the noise. By Proposition 4.7, the irreducible error in both domains is $\sigma^2$, and the overestimate $\Delta$ of our estimator should be bounded by $2\sigma$. We vary $\sigma$, sample 10,000 points from each domain each time, and run DataShifts with $\beta=0.2$, giving:
>
> |$\quad\quad\quad\quad\quad\quad\quad\sigma$|0.01|0.1|0.3|0.5|1.0|2.0|5.0|10.0|
> |:--:|:--:|:--:|:--:|:--:|:--:|:--:|:--:|:--:|
> |irreducible error bound ($1.0+2\sigma$)|1.02|1.2|1.6|2.0|3.0|5.0|11.0|21.0|
> |$\hat S_{Cpt}$|1.0027|1.0338|1.0712|1.1540|1.4750|2.4693|5.6697|11.4051|
>
> When noise $\sigma$ is small, our estimator $\hat S_{Cpt}$ stays close to the true Y|X shift (1.0). Even when the noise std in both domains reaches half of the true Y|X shift ($\sigma=0.5$), it still does not notably overestimate. As $\sigma$ grows far beyond 1.0, Y|X shift becomes less identifiable, and overestimation appears, but it stays below the bound set by irreducible error, as proved in Proposition 4.7.
>
> **Thanks again for your support. We will add these tests and notes in the revision to further strengthen our theory. If these replies address your concerns, we would greatly appreciate a score increase and welcome further comments.**

---

> > ### Author Rebuttal · Reviewer_74cM · 2026-04-03
> >
> > Thank you for your response.
> >
> > I think those tables will be good additions to the paper. I maintain my positive score.

---

> > > ### Author Response · Authors · 2026-04-04
> > >
> > > Thank you very much for your careful feedback and support for our work. We are delighted that you appreciate this research. We wish you all the best!

---

### Official Review · Reviewer_VJ2J · 2026-03-09

**Soundness:** 2
**Presentation:** 2
**Significance:** 1
**Originality:** 2
**Overall Recommendation:** 3
**Confidence:** 3

**Summary:**

This paper investigates the difference in expected losses between source and target distributions when both concept and covariate shifts occur. The authors propose metrics to measure concept and covariate shifts using the entropic regularized Wasserstein distance, which is capable of measuring distributional shift even when the supports of the source and target distributions are disjoint. They demonstrate that the sum of these metrics characterizes the difference in expected losses between the source and target distributions under a Lipschitz continuity assumption on the loss function and the hypothesis class. Furthermore, they develop estimators for the proposed concept and covariate shift metrics and establish their convergence rates. The experimental results show a correlation between the proposed bound and the test error.

**Compliance With Llm Reviewing Policy:**

Affirmed.

**Final Justification:**

I have increased the score to 3, as the authors’ response clarifies the distinction from existing measures. However, the benefits of the proposed bounds remain unclear. Although the authors claim improved tightness in their responses and experimental discussion, this claim is not sufficiently validated, either theoretically or empirically, in comparison to existing bounds.

Moreover, I disagree with the authors’ argument that the proposed metric captures conditional shift more adequately than WoW. The proof of Theorem 3.13 appears to remain valid even when $\gamma^*$ is replaced with any conditional coupling, as the underlying logic is preserved. This would also allow for the use of an optimal coupling, as in WoW.

**Key Questions For Authors:**

1. Could the authors explicitly highlight the theoretical advantages or practical benefits that their derived bounds offer over existing results, particularly those established by Courty et al. (2017) and Hamri et al. (2025)?
2. What are the specific methodological advantages of the proposed estimators and sample complexity bounds (e.g., Theorem 4.5) compared to the related findings by Hamri et al. under similar assumptions?
3. What are the primary insights derived from the experimental results, and how do they substantiate the benefits of the proposed theoretical and methodological contributions?

**Limitations:**

yes

**Strengths And Weaknesses:**

Addressing distributional shift is a crucial problem in machine learning and is highly relevant to the scope of the conference. Disjoint supports between source and target distributions are a common issue in real-world transfer learning applications. Thus, theoretical analyses of distributional shift under disjoint supports are well motivated. The theoretical results presented in this paper appear to be sound, although I have not fully verified the proofs in the appendices.


However, the contributions of this paper are unclear. The authors state their main contribution is developing methods to quantify distributional shift under disjoint supports, yet many existing works have already provided theoretical analyses for this exact setting. For example, existing results based on optimal transport costs, such as those by Redko et al. (2017) and Courty et al. (2017), already provide non-trivial bounds on distributional shift under disjoint supports.


The main result of this paper is Theorem 3.13; however, its novelty is unclear. Theorem 3.13 is derived using a deliberately chosen coupling measure alongside a Lipschitz continuity assumption on the loss function and the hypothesis class. This approach is highly similar to the analysis given by Courty et al. (2017). While the resulting bounds differ due to the different assumptions employed, the specific benefits over Courty et al. (2017)'s bound need to be adequately justified. Moreover, the proposed measure for concept shift can be interpreted as the hierarchical optimal transport cost (Hamri et al., 2025) or the Wasserstein over Wasserstein distance (Bonet et al., 2025). Hamri et al. and Bonet et al. (and references therein) have already introduced the idea of characterizing distributional shift using such measures. In particular, Hamri et al. provide a bound on the difference between source and target risks using the hierarchical optimal transport cost. Therefore, a clear benefit over Hamri et al.'s results should be demonstrated.

- M.E. Hamri et al. Theoretical guarantees for domain adaptation with hierarchical optimal transport. Machine Learning, 2025.
- Bonet et al. Flowing Datasets with Wasserstein over Wasserstein Gradient Flows. ICML, 2025.


The estimation of the covariate (X) shift is equivalent to estimating the entropic regularized optimal transport cost. The estimation of (entropic regularized) optimal transport cost is a well-studied problem, and numerous existing works provide theoretical analyses of its estimation, including:

- Genevay et al. Sample Complexity of Sinkhorn Divergences. AISTATS, 2019.
- Mena et al. Statistical Bounds for Entropic Optimal Transport: Sample Complexity and the Central Limit Theorem. NeurIPS, 2019.
- Rigollet et al. On the sample complexity of entropic optimal transport. Ann. Statist., 2025.

The novelty of the proposed estimator and Theorem 4.2 relative to these prior results should be clarified.


The estimation of the concept (Y|X) shift is fundamentally the same task as estimating the hierarchical optimal transport cost (Hamri et al., 2025) or the Wasserstein over Wasserstein distance (Bonet et al., 2025). Hamri et al. established the sample complexity of the hierarchical optimal transport cost under the same assumption of finite squared-exponential moments. The benefit of the proposed estimator and Theorem 4.5 over Hamri et al.'s results should be demonstrated.


The experimental results are not convincing. For the experiments in Section 5.1, the authors claim that the estimated error bound is tight; however, this assessment is subjective. Furthermore, the objective of the experiments in Section 5.2 is unclear and fails to demonstrate the benefits of the proposed estimator. Moreover, Figures 1(b) and 1(c) are misleading; the vertical and horizontal axes should be plotted on the same scale and include a diagonal reference line, as shown in Figure 1(a).

---

> ### Author Rebuttal · Authors · 2026-03-31
>
> Thanks for the comments. Reviewer VJ2J sees our work as similar to Hamri et al. (2025). We disagree: our work is **fundamentally different and better.**
>
> # 1. Clear gains over prior OT bounds
>
> > Could the authors explicitly highlight the theoretical advantages...particularly Courty et al. (2017) and Hamri et al. (2025)?
>
> Yes. These gains are already implied by our baseline, Zhao et al. (2019) (Theorem 2.3). We restate them below:
>
> |$\quad\quad\quad$Bound|Divergence|$\quad\quad$X shift|Y\|X shift|Remaining Term|Tight|Estimable|General|
> |:--:|:--:|:--:|:--:|:--:|:--:|:--:|:--:|
> |Ben-David et al. (2006)|$\mathcal{H}$-divergence|✓||joint-error λ||||
> |Redko et al. (2017)|OT|✓||joint-error λ||||
> |Courty et al. (2017)|OT|estimated XY shift ||joint-error λ, kMΦ|||✓|
> |Zhao et al. (2019) **(ICML oral, baseline)**|$\mathcal{H}$-divergence|✓|✓(ill-defined)||✓|||
> |Zhang et al. (2023)|$\mathcal{H}$-divergence|✓|✓(ill-defined)||✓||✓|
> |Hamri et al. (2025) **(raised by Reviewer VJ2J)**|hierarchical OT|✓||joint-error λ||||
> |**Ours**|entropic OT|✓|✓||✓✓|✓|✓✓|
>
> - All bounds with joint error $\lambda=\min\limits_{h}\epsilon_S(h)+\epsilon_T(h)$, including Hamri et al., follow Ben-David et al. (2006): bound via $h^*$, leaving λ.
> - Zhao et al. (2019) already showed that λ hides Y|X shift, is "very conservative and loose," and is "intractable to compute." It thus argued for bounds "free of the pessimistic λ term" and introduced an **explicit** Y|X shift tighter than λ. Many later works support this view.
> - By contrast, **Hamri et al. uses hierarchical OT only for X shift, still leaving Y|X shift in λ**. The paper itself calls λ "ideal" and "non-estimable." So bounds with λ, including Hamri et al., are loose and non-estimable.
> - Our theory keeps this gain and further improves Y|X shift via OT. Unlike prior OT bounds, it does not follow the naive proof route and has no λ. **It is tighter, fully estimable, and general.**
>
> Moreover, Hamri et al. notes that, since hierarchical OT bounds standard OT, its final bound is even looser than Redko et al. (2017). **Since both its X-shift term and λ are loose, it may not be a SOTA theory for distribution shift.**
>
> # 2. "hierarchical OT" is unrelated
>
> VJ2J’s claim is incorrect: our idea is wholly different from Hamri et al. We use the transport coupling γ\* to link X and Y|X shifts, handle support mismatch in Y|X shift, and derive a unified bound for X and Y|X. By contrast, Hamri et al. split source and target covariates into k clusters each **(conditional X)**, then use Wasserstein distances between clusters as costs between the two cluster sets; this is their hierarchical OT for unsupervised DA. **The only common point is a nested OT form, i.e., merely a formal similarity. Their definitions, meanings, properties, and goals are all different—hierarchical OT cannot capture Y|X shift.**
>
> In our γ\*-Y|X shift, γ\* is exogenous: it comes from X shift and ensures that the Y|X gap $S_{pair}$ is compared only on nearby X pairs $(x_S,x_T)$. In the hierarchical OT view, if one instead minimizes over γ with Y|X gap as cost, the term becomes $S_{hot}=\min\limits_{\gamma}\int S_{pair}d\gamma$. Then γ is endogenous and no longer enforces proximity in X. **So the Y|X gap no longer respects the geometry of X**: if $D_X^S=D_X^T=U(0,1)$ but the concepts are 1-x and x, then still $S_{hot}=0$. **Thus, Y|X shift must be defined via the X-shift coupling γ\*, not hierarchical OT.**
>
> # 3. Statistical results in Hamri et al. is incorrect
>
> > What are the specific advantages of the proposed estimators and sample complexity bounds (Theorem 4.5) compared to Hamri et al.?
>
> A direct comparison is not possible: our Theorem4.5 is for Y|X shift, while Hamri et al. studies X shift. More importantly, its sample complexity result is wrong. Theorem 13 in Hamri et al. implies: let HW be hierarchical OT, ζ' a constant; with probability at least 1−δ,
> $$
> |HW_1(\varphi_S,\varphi_T)-HW_1(\hat{\varphi}_S,\hat{\varphi}_T)|\leq HW_1(\varphi_S,\hat{\varphi}_S)+HW_1(\varphi_T,\hat{\varphi}_T)\leq2\sqrt{\frac{2\log\left(\frac{1}{\delta}\right)}{\zeta^{\prime}k}}
> $$
> This is odd: **each domain has N samples, split into k clusters, with c samples per cluster, yet the concentration of hierarchical OT from 2N samples depends only on k, not on N or c!**
>
> This proof has two clear errors:
>
> - The bound treats each cluster as one sample and directly uses Theorem 11 in Hamri et al. But Theorem 11 **requires i.i.d. samples**. In hierarchical OT, clusters come from labels or clustering, so they are **not i.i.d.**
> - Even ignoring this, since the events $HW_1(\varphi_S,\hat{\varphi}_S)$ and $HW_1(\varphi_T,\hat{\varphi}_T)$ are independent, the bound should hold with probability at least 1-2δ, not 1-δ. **This basic stats error also gives a wrong concentration rate and sample complexity.**
>
> Worse, this error spreads to many other results in Hamri et al. In fact, nested OT-type statistical proofs are hard and cannot be obtained by misusing a standard result.

---

> > ### Author Rebuttal · Reviewer_VJ2J · 2026-04-02
> >
> > Thank you for the detailed response. I appreciate the clarification regarding Hamari et al. However, the concerns related to Courty et al. and Bonet et al. remain unaddressed. I would appreciate it if the authors could explicitly clarify how their approach compares to and improves upon these prior works.

---

> > > ### Author Response · Authors · 2026-04-07
> > >
> > > **Thank you very much for acknowledging that our work is different from Hamari et al. We are glad to further clarify our advantages over both Courty et al. (2017) and Bonet et al. (2025).**
> > > # Our gains over Courty et al.
> > > - **1. Like Hamari et al., Courty et al.'s bound also contains the joint error λ, making it loose and non-estimable.** As shown in the third row of our rebuttal table, Courty et al.'s bound includes the joint error λ ($err_S(f^* )+err_T(f^*)$ in Theorem 3.1), which is widely recognized as loose and non-estimable. **In contrast, our bound uses the explicit Y|X shift instead of this defective λ term, making it tighter and estimable.**
> > > - **2. Courty et al. measures an "estimated" joint (XY) shift, depending on an extra hypothesis $f$ rather than ground truth.** It uses the output of $f$ to supply target labels (Eq.4), yielding an estimated joint shift without real target label information. If $f$ fits the target labels poorly, **this can introduce large error. By contrast, our theory studies real X and Y|X shifts, without relying on an extra hypothesis $f$ or its induced error.**
> > > - **3. Courty et al.'s statistical result also has issues.** Their sample complexity proof for Eq. 20 directly relies on Bolley et al. (2007), proved for Euclidean cost. But Courty et al. uses OT on joint distributions, where the cost is the sum of metrics on X and Y (below Eq. 20), not a Euclidean cost. Hamari et al. also note this in their Remark 12. **In contrast, our statistical result is rigorous: it avoids such joint cost and does not generalize beyond Bolley et al.'s standard result.**
> > >
> > > Overall, unlike Courty et al., our theory studies real X and Y|X shifts rather than an "estimated" joint shift, and yields a tighter, estimable bound, with rigorous statistical results.
> > > # Comparison with  Bonet et al.
> > > **Bonet et al. is on a different topic**: they study the mathematical properties of Wasserstein over Wasserstein (WoW, i.e., hierarchical OT), including tangent spaces, gradients, continuity equations, and gradient flows on the space of distributions over distributions, and proposes a computable WoW gradient flow. **Unlike our paper, it gives no learning bound for distribution shift, nor any estimator, concentration result or sample complexity.** It only reports gradient flow experiments on domain adaptation datasets, and notes that domain adaptation "is not the goal of the paper".
> > >
> > > Your concern seems to arise from a formal similarity, as **WoW (Bonet et al., 2025), i.e., hierarchical OT (Hamari et al., 2025),** follows the form:
> > > $$
> > > \inf_{\gamma\in\Pi(\mu_A,\nu_A)}\int\mathrm{OT}(\mu_{B|a_1},\nu_{B|a_2})\mathrm{d}\gamma(\mu_{B|a_1},\nu_{B|a_2})
> > > $$
> > >
> > > where the coupling $\gamma$ **depends only on the cost** $\mathrm{OT}(\mu_{B|a_1},\nu_{B|a_2})$. By contrast, **our γ\*-Y|X shift** follows the form:
> > > $$
> > > \int\mathrm{OT}(\mu_{B|a_1},\nu_{B|a_2})\mathrm{d}\gamma^* (\mu_{B|a_1},\nu_{B|a_2})\quad\text{where }\gamma^*\sim\mathrm{OT}(\mu_A,\nu_A)
> > > $$
> > >
> > > where the optimal coupling $\gamma^*$ **aligns the two marginals** $\mu_A,\nu_A$.
> > >
> > > **This difference in the role of the coupling is crucial.** We emphasize:
> > > ## WoW (hierarchical OT) can fail to capture conditional (Y|X or X|Y) shift, while our marginal-aligned form is necessary.
> > >
> > > - **A clear X|Y shift example:** Let $P(X),Q(X)$ be two different covariate distributions, and consider class-balanced binary classification: source $D^S_{X|y=0}=P(X),D^S_{X|y=1}=Q(X)$, target $D^T_{X|y=0}=Q(X),D^T_{X|y=1}=P(X)$. **This has clear X|Y shift.** Yet $\mathrm{OT}(D^S_{X|y=0},D^T_{X|y=1})=0$ and $\mathrm{OT}(D^S_{X|y=1},D^T_{X|y=0})=0$. **So if WoW is used for X|Y shift, its coupling γ matches the wrong classes due to these smaller costs, and the resulting value is 0. By contrast, our marginal-aligned form first aligns the marginal—class Y, so X|Y is compared within the same class and the correct X|Y shift is recovered.** We also gave a similar Y|X shift example in Part 2 of the earlier rebuttal.
> > >
> > > - **To compare conditionals (B|a), one must first match similar marginals a.** In our form, $\mathrm{OT}(\mu_A,\nu_A)$ yields the coupling γ\* aligning the two marginals, so $\mu_{B|a_1},\nu_{B|a_2}$ are compared only along nearby pairs $a_1,a_2$. In WoW, the coupling γ depends only on the cost, ignoring the similarity between $a_1,a_2$. **Thus, as the examples show, WoW can fail to capture conditional (Y|X or X|Y) shift, while using γ\* to first align the marginals, as we do, is necessary.**
> > >
> > > **Overall, our bound improves on Courty et al. in many respects. Bonet et al. studies a different topic and provides no bounds or statistical results. Formally, our γ\*-Y|X shift captures underexplored Y|X shift, whereas WoW (Bonet et al.) or hierarchical OT (Hamari et al.) isn't suitable.** We still thank you for pointing out these recent papers, and are glad to cite and distinguish them in Related Works. If these replies address your concerns, we would appreciate a score increase and welcome further comments!

---

### Official Review · Reviewer_BPza · 2026-03-10

**Soundness:** 4
**Presentation:** 3
**Significance:** 4
**Originality:** 4
**Overall Recommendation:** 5
**Confidence:** 4

**Summary:**

Distribution shifts are of myriad kinds, and this work really appears to identify the ill definition and non estimability of concept shift when the supports of the source and target distributions are non overlapping. The work reframe the problem wrt  entropic optimal transport that is robust to support misalignmet. It derives a unified learning bound that would apply over randomized labeling and generic losses. The presented algorithm follows a debiased estimator to quantify the shift from a finite sample size.

**Compliance With Llm Reviewing Policy:**

Affirmed.

**Key Questions For Authors:**

I wonder if the Lipshiftz-ness of the non-final layers  will hold if the the functional approximate presents discontinuities, but perhaps I'm nitpicking.

Could you comment on the reason for the losseness of the bound on the pacs data?

**Limitations:**

The apper mentions the difficulty of estimating for large transformer/s, and admit that the shift estimate may itself have estimation bias.

**Strengths And Weaknesses:**

I could not follow all proofs due to revie load, but the work proves and provides concentration bounds for shift estimation and before that, for the well definition of the new framing of shift.
That soundness is followed by experimental presentation over regression and classification tasks to show that the experimental error lies close to the developed estimates/

The work is well presented, and demarkates itself from the limitations of prior art. The additional proofs give  information on Lipshits constants. The distinction between central and ancillary proofs is essential for a long, dense pape r.

Having worked  etxensively with other types of distribution shift, I understand the significance of the framing to understand and quantify the shift in deployment  regimes, especially with little support overlap.

The originality of the work is quite elevated, from extensive reading of sota.

---

> ### Author Rebuttal · Authors · 2026-03-31
>
> **Thank you very much for recognizing the value of this work—our $\gamma^*$-Y|X shift solves the support mismatch in prior Y|X shifts, yields a general bound unifying X and Y|X shifts, and further provides estimators and concentration results for these shifts, making them deployable and quantifiable across a wide range of tasks.** Having devoted long-term effort to this goal, your strong support means a great deal to us.
>
> > I wonder if the Lipshiftz-ness of the non-final layers will hold if the the functional approximate presents discontinuities, but perhaps I'm nitpicking.
>
> We believe this depends on the network type.
>
> - For a simple feedforward network, since each linear and activation layer is Lipschitz continuous, their composition is also Lipschitz continuous. Still, the Lipschitz constant of a deep network may become extremely large due to repeated multiplication, so the network may **approximately present discontinuities**.
> - For more complex models, such as Transformers, Lipschitzness may not be guaranteed. As Castin et al. (2023) point out, for sequence length $n$, the Lipschitz constant of an attention layer has a lower bound of order $\sqrt{n}$. This means that under the common theoretical assumption of unbounded sequence length ($\sqrt{n}\to\infty$), even a single attention layer may fail to be Lipschitz, and thus the whole Transformer may not be Lipschitz either.
>
> However, regardless of whether the non-final layers are continuous, as long as the last layer is continuous (often a linear classifier, including in Transformers), one can naturally apply our theory by taking the representation space of the preceding layers as the covariate space.
>
> > Could you comment on the reason for the losseness of the bound on the pacs data?
>
> Yes, this follows your last question. We stress that **the main looseness comes from the hypothesis Lipschitz constant $L_h$, while the X and Y|X shifts in our framework still remain informative.** On PACS, the representation dimension is 2048; even using only the last layer, it’s still a complex function on a 2048-dim input. Thus $L_h$ is large and the bound is loose: over all points in Fig. 1(c), the Pearson correlation between bound and test error is 56.9%. But since $L_h$ measures the model’s largest response to input change, we also replace it by $\frac14L_h$ and $\frac18L_h$ to reflect a more average response. The resulting scaled bounds raise the correlation to 86.7% and 91.4%. This suggests that the looseness mainly comes from $L_h$, while our X and Y|X shifts still retain much information about error.
>
> **We will further develop the theory. Thank you again for your strong support of this work!**
>
> [1]. Castin, Valérie, Pierre Ablin, and Gabriel Peyré. "How smooth is attention?." *arXiv preprint arXiv:2312.14820* (2023).

---

> > ### Author Rebuttal · Reviewer_BPza · 2026-04-06
> >
> > My questions were about curiosities. The authors have agreed that both represent a limitation. We both see the edges of tge solution space.

---

### Official Review · Reviewer_qZWw · 2026-03-11

**Soundness:** 3
**Presentation:** 2
**Significance:** 3
**Originality:** 3
**Overall Recommendation:** 4
**Confidence:** 2

**Summary:**

This paper develops a framework for analyzing distribution shift that may arise from both covariate and concept changes. The authors address a limitation of existing approaches when the supports of the source and target covariate distributions do not overlap, in which case conditional distributions used to define concept shift are not well-defined. To overcome this, they introduce a formulation based on optimal transport that aligns the covariate distributions before comparing conditional label distributions. Using this framework, the paper derives an upper bound on the target-domain generalization error in terms of the source-domain error, a covariate shift term, and an optimal-transport–based concept shift term. The authors further provide finite-sample guarantees for estimating these quantities and propose an algorithm for quantifying distribution shift and estimating the resulting error bound in practical settings.

**Compliance With Llm Reviewing Policy:**

Affirmed.

**Final Justification:**

In the rebuttal, the authors clarified several points, particularly their positioning relative to prior OT-based approaches and the role of their γ*-based concept shift. The additional synthetic comparison and sensitivity analyses help support their claim that the proposed formulation can yield tighter and more informative bounds than prior work in certain cases. These responses mostly addressed my main concerns and improved my understanding of the contribution. While, I believe that practical applicability is somewhat limited due to reliance on quantities such as Lipschitz constants and representation choices, and the interpretation of the bias term in the stochastic-label setting remains challenging, I believe that strengths outweigh the weaknesses.
Therefore, my overall assessment is that I have a more positive evaluation of this work, recommending weak acceptance.

**Key Questions For Authors:**

**Questions**
1. Can you clarify the exact role of $\beta$ across theory and experiments? In particular, what guarantee supports the debiased $X$-shift estimator when $\beta>0$, since Theorem 4.2 is stated for $\beta=0$, while the experiments again use $\beta > 0$ (e.g., $\beta = 0.2$)?
2. Can you compare against alternative shift metrics or prior theoretical bounds to show that the proposed quantities are more informative in practice?
3. Is there any way to estimate or upper bound the stochastic-labeling bias term $\Delta$ empirically?
4. Since the bound scales linearly with $L_h$, if $L_h$ is large, as is typical for deep networks, the bound may become vacuous. Can  the framework still remain informative in such settings?
5. How sensitive are $S_{Cov}, S_{Cpt}^\gamma$ and the bound in Theorem 3.13 to the choice of $\beta$, metric $\rho_X$ and representation space? Since the optimal coupling $\gamma^*$ depends on these choices it would be helpful to understand whether the resulting notion of concept shift is stable or canonical.

**Comments**

* The reference list appears to contain several citations that are not used in the main text. In particular, Arjovsky et al. (2019), Chen & Marchand (2023), Krueger et al. (2021), and Pei et al. (2018) do not seem to be cited anywhere in the paper. The authors may want to either incorporate discussion of these works in the related-work section or remove them to avoid vacuous references.

* The proof of Proposition D.5 appears to be a direct results from prior work (e.g., Fazlyab et al.). It would be helpful to explicitly state that the result is adapted from prior work when stating the proposition or briefly clarify how the argument applies in the present setting if it is not the exact same statement.

**Limitations:**

Yes

**Strengths And Weaknesses:**

**Strengths**

1. Addressing covariate support mismatch is an important problem, and the OT-based alignment idea is interesting and well-motivated.
2. The theoretical development appears coherent, with formal proofs and estimators supporting the proposed framework.
3. The work goes beyond population-level analysis and provides finite-sample statistical guarantees.
4. Experiments cover regression, binary classification, and multiclass classification, which supports the paper’s claims of generality of settings.

**Weaknesses**

1. The positioning relative to prior OT-based domain adaptation work could be clearer. In particular, it would help to better distinguish the proposed $ {\gamma^*}$  - conditioned formulation from existing joing-OT or conditional-shift approaches.
2. The exposition is quite dense. Namely, many lemmas appear in succession, and key concepts, such as estimator interpretation, the role of $ \beta $, and practical implications of Lipschitz constants and transport-based shifts are not explained very intuitively.
3. The empirical validation is mostly qualitative. The main evidence is that the estimated bound visually tracks test error in scatter plots. Quantitative evaluation, e.g., rank correlation, mean bound slack, or comparisons with simpler baselines, would make the results more convincing.
4. Practical applicability depends on quantities that can be difficult to estimate in modern ML settings, particularly Lipschitz constants and the geometry of the covariate space. While the paper partially addresses this by operating in learned representation spaces, the framework is not yet clearly applicable to large end-to-end neural models.
5. In the stochastic-label setting, the concept-shift estimator concentrates around $ {{S}^{\gamma}}_{Cpt}+\Delta $, rather than the target quantity itself, with $ \Delta $ controlled only through irreducible error. While theoretically justified, this reduces the interpretability of the estimated concept-shift term unless $ \Delta $ can be characterized or estimated in practice.

---

> ### Author Rebuttal · Authors · 2026-03-31
>
> Thank you for the careful reading and constructive comments. We are glad you value our OT-based Y|X shift idea and the completeness of our theory.
>
> ## W1.
> Since target labels are usually absent in DA, most prior OT-based bounds study only X shift (Redko et al., 2017; Shen et al., 2018). Courty et al. (2017) uses the learner to estimate the target concept, thus studying an approximate joint XY shift. The Y|X shift in these results is hidden in the joint-error λ. Our baseline, Zhao et al. (2019), showed that λ is loose and non-estimable, and proposed an explicit Y|X shift instead. We follow this line, improve Y|X shift via OT, propose a tighter, fully estimable, and general theory. We have also provided a more detailed table in our response to Reviewer 74cM’s Q1.
>
> ## Q1.
> The concentration result for the X shift estimator is proved only for $\beta=0$, while our experiments use small $\beta>0$. We add that $|W_\beta-W|\leq\beta C$, where $C$ depends only on the marginal entropies (Blondel et al., 2018). So the gap between our experiments and proved result is controlled by $\beta$, and the debiased X shift estimator can be guaranteed for small $\beta$. A sensitivity test is given below.
>
> ## Q2.
> We compare with our baseline Zhao et al. (2019), for binary classification and known to be tighter than prior bounds, estimable on synthetic tasks with known concepts. We use a 10-dim covariates: $D_X^S=U([0,1]^{10})$, source concept is logistic regression (LR) with $w_S=\frac1{\sqrt{10}}\mathbf{1}, b_S=0$. The target domain is a shift of $D_X^S$ by 0.5 in a random direction, with target concept another LR with $b_T=0$, where $w_T$ is formed by rotating $w_S$ by angle $\theta$ toward a random direction. The learner is also LR. We vary $\theta$, repeat each setting 20 times, and report the average of test error, the baseline, and our bound:
>
> |$\quad\quad\theta^{\circ}$|0|20|40|60|80|100|120|140|160|180|
> |:--:|:--:|:--:|:--:|:--:|:--:|:--:|:--:|:--:|:--:|:--:|
> |test error|0.000|0.057|0.114|0.168|0.218|0.262|0.298|0.325|0.344|0.350|
> |baseline bound|0.193|0.254|0.310|0.365|0.414|0.458|0.495|0.524|0.539|0.543|
> |our bound|0.196|0.217|0.253|0.296|0.338|0.377|0.409|0.434|0.451|0.455|
>
> When $\theta=0$ (X shift only), the two bound are almost the same. As $\theta$ grows, Y|X shift increases, and the baseline becomes looser than ours. This supports our point: the prior ill-defined Y|X shift term is loose, while our $\gamma^*$-Y|X shift is tighter and more informative.
>
> ## Q3.
> Yes. Proposition 4.7 already shows that $\Delta$ is bounded by the irreducible error under squared loss. Irreducible error is also bounded by generalization error of any model, and test error unbiasedly estimate the generalization error. Thus, for tasks covered by our theory, the squared-loss test error of any SOTA model gives a natural empirical upper bound for $\Delta$.
>
> ## Q4.
> Yes. A large $L_h$ can make the bound loose, while the X and Y|X shifts in our framework still stay informative. On PACS, the feature dimension is 2048, so $L_h$ is large and the bound is loose: over all points in Fig. 1(c), the Pearson correlation between the bound and test error is 56.9%. Since $L_h$ measures the model’s largest response to input change, we also replace it by $\frac14 L_h$ and $\frac18 L_h$ to reflect a more average response. The resulting scaled bounds raise the correlation to 86.7% and 91.4%. This suggests that the looseness mainly comes from $L_h$, while our X and Y|X shifts still retain much information about error.
>
> ## Q5.
> To keep the theory more general, we use an abstract metric $\rho_X$, at the cost that the measured shift may vary across metrics. In practice, we recommend Euclidean distance, as in this paper and most works. For $\beta$, we test the sensitivity of $S_{Cov}$ and $S_{Cpt}^{\gamma^{*}}$. We take the "-90%" domain of ColoredMNIST as target, use ERM, train with the best hyperparameters from DomainBed. At the last checkpoint, we sample 10,000 feature-label pairs from both source and target domains. Running DataShifts with different $\beta$ gives:
>
> |$\quad\beta$|0.001|0.002|0.005|0.01|0.02|0.05|0.1|0.2|
> |:--:|:--:|:--:|:--:|:--:|:--:|:--:|:--:|:--:|
> |$\hat S_{Cov}$|0.5074|0.5073|0.5071|0.5066|0.5048|0.5028|0.4985|0.4927|
> |$\hat S_{Cpt}$|1.4459|1.4461|1.4471|1.4489|1.4507|1.4582|1.4686|1.4903|
> |time (s)|15.27|14.13|12.75|11.64|10.66|9.12|8.05|6.75|
>
> Across magnitude changes in $\beta$, the coefficients of variation of $\hat S_{Cov}$ and $\hat S_{Cpt}$ are 0.983% and 0.998%, so both are stable for small $\beta$. Since entropic OT gets faster as $\beta$ grows, we use $\beta=0.2$ in the paper to balance speed and accuracy.
>
> **Thanks again for your support. We will add these tests and notes in the revision to further strengthen our theory. If these replies address your concerns, we would greatly appreciate a score increase and welcome further comments.**
>
> [1]. Blondel et al. Smooth and sparse optimal transport. AISTATS, 2018.

---

> > ### Author Rebuttal · Reviewer_qZWw · 2026-04-03
> >
> > I appreciate the authors' response and clarification, my concerns have been appropriately addressed. I increase my score to 4, recommending weak acceptance.

---

> > > ### Author Response · Authors · 2026-04-04
> > >
> > > Thank you very much for your careful reading of our work and for your constructive comments. We are glad to have been able to address your concerns in our rebuttal, and we are very grateful to your increased appreciation of our work. We wish you all the best!

---

### Decision · Program_Chairs · 2026-04-30

**Decision:**

Accept (regular)

**Comment:**

While the reviewers mentioned (1) the qualitative experimental results, (2) the looseness in the bound from the hypothesis Lipschitz constant, and (3) the limited discussion of the method's actionability, the reviewers overall found the proposed quantification of covariate and concept shift to be conceptually clean and appreciated the statistical estimation procedure. Given this contribution, I recommend acceptance.